# Efficient Federated Learning against Byzantine Attacks and Data Heterogeneity via Aggregating Normalized Gradients

**Shiyuan Zuo**[1]
zuoshiyuan@bit.edu.cn

**Xingrun Yan**[1]
ystarr_cs@163.com

**Rongfei Fan**[1,*]
fanrongfei@bit.edu.cn

**Li Shen**[2]
shenli6@mail.sysu.edu.cn

**Puning Zhao**[2]
zhaopn@mail.sysu.edu.cn

**Jie Xu**[3]
xujie@cuhk.edu.cn

**Han Hu**[1]
hhu@bit.edu.cn

[1]Beijing Institute of Technology, Beijing, China     [2]Sun Yat-Sen University, Guangzhou, China
[3]The Chinese University of Hong Kong (Shenzhen), Shenzhen, China

## Abstract

Federated Learning (FL) enables multiple clients to collaboratively train models without sharing raw data, but is vulnerable to Byzantine attacks and data heterogeneity, which can severely degrade performance. Existing Byzantine-robust approaches tackle data heterogeneity, but incur high computational overhead during gradient aggregation, thereby slowing down the training process. To address this issue, we propose a simple yet effective Federated Normalized Gradients Algorithm (Fed-NGA), which performs aggregation by merely computing the weighted mean of the normalized gradients from each client. This approach yields a favorable time complexity of $\mathcal{O}(pM)$, where $p$ is the model dimension and $M$ is the number of clients. We rigorously prove that Fed-NGA is robust to both Byzantine faults and data heterogeneity. For non-convex loss functions, Fed-NGA achieves convergence to a neighborhood of stationary points under general assumptions, and further attains zero optimality gap under some mild conditions, which is an outcome rarely achieved in existing literature. In both cases, the convergence rate is $\mathcal{O}(1/T^{\frac{1}{2}-\delta})$, where $T$ denotes the number of iterations and $\delta \in (0, 1/2)$. Experimental results on benchmark datasets confirm the superior time efficiency and convergence performance of Fed-NGA over existing methods.

## 1  Introduction

FL has recently emerged as a distributed paradigm that addresses challenges of large-scale data and privacy concerns by enabling multiple edge clients to collaboratively train a global model without sharing raw data (Zuo et al., 2024; Yin et al., 2018; Konečný et al., 2016; Li et al., 2020). Under the coordination of a central server, clients exchange model parameters or gradients rather than raw data. In each training round, the server aggregates the received messages to update the global model, which is then broadcast back to clients for local updates using their private data (Wang et al., 2019; Guo et al., 2023). This privacy-preserving mechanism, coupled with growing edge computing capabilities, makes FL increasingly attractive for modern learning scenarios (Dorfman et al., 2023; Zhao et al., 2024).

---

*Corresponding author.

Despite its advantages, FL faces robustness challenges due to the participation of multiple clients (Kairouz et al., 2021; Vempaty et al., 2013). Messages uploaded to the central server may deviate from expected values because of data corruption, device failures, or malicious behavior (Yang et al., 2020; Cao and Lai, 2019). Such anomalies are referred to as Byzantine attacks, and the responsible clients are termed Byzantine clients, while others are considered honest (So et al., 2020; Cao and Lai, 2019). The server neither knows the number nor the identities of Byzantine clients, who may adaptively craft and coordinate misleading updates (Chen et al., 2017). These attacks can severely degrade or even derail training, making robust aggregation strategies essential for secure FL.

In addition to robustness issues, data heterogeneity among participating clients presents a major challenge in FL. Unlike traditional distributed learning, FL clients gather training data from their local environments, resulting in non-independent and non-identically distributed (non-IID) data across clients (Zhao et al., 2018). This heterogeneity introduces biases in local optimal solutions, which can hinder the convergence of the global model. As a result, achieving reliable convergence under non-IID conditions remains a central focus in FL research (Xie and Song, 2023).

The literature on Byzantine-resilient FL aggregation spans both strongly convex settings (Pillutla et al., 2022; Wu et al., 2020; Zhu and Ling, 2023; Li et al., 2019; Karimireddy et al., 2021) and the more general and challenging case of non-convex loss functions (Yin et al., 2018; Turan et al., 2022; Blanchard et al., 2017; Luan et al., 2024). The robustness performance of these methods has been analyzed on IID (Xie et al., 2018; Yin et al., 2018; Turan et al., 2022; Blanchard et al., 2017; Wu et al., 2020; Zhu and Ling, 2023; Karimireddy et al., 2021) and non-IID datasets (Pillutla et al., 2022; Li et al., 2019; Luan et al., 2024), with the wish of achieving a stationary point. Most existing studies have established convergence to a neighborhood of stationary points, while none have thus far guaranteed convergence with zero optimality gap, that is, assured convergence to a true stationary point. It is worth noting that most prior work has primarily concentrated on designing sophisticated aggregation schemes to counter Byzantine attacks, often overlooking the need to reduce the computational burden of aggregation. As summarized in Table 1, the most robust aggregation algorithms typically impose significant computational overhead on the central server, with complexities dependent on parameters mainly including the model dimension $p$, the number of clients $M$, and the error tolerance of aggregation algorithm $\epsilon$. A more detailed survey of related work is provided in Appendix D. In this paper, we aim to fill this research gap, i.e., to reduce the cost of computational complexity for aggregation while preserving generality to data heterogeneity. One potential approach for achieving our goal is to normalize the uploaded vectors before aggregation. Using normalization to handle gradients is widely adopted in machine learning (Cutkosky and Mehta, 2020) as it helps stabilize training by ensuring that the loss remains low even under slight perturbations to model parameters (Dai et al., 2023), and by mitigating the impact of heavy-tailed noise during training (Jakovetić et al., 2023). In FL, normalization has also been used to address data heterogeneity, as demonstrated in Li et al. (2021) and Wang et al. (2020). While existing literature has not applied normalization specifically to counter Byzantine attacks, its ability to tackle heavy-tailed noise suggests potential for mitigating the influence of outliers, which are commonly observed under Byzantine attacks. Even in the absence of adversarial behavior, the operation of normalization can still alleviate data heterogeneity and perform well in a FL system. Last but not least, as shown in Table 1, normalization incurs minimal computational overhead, making it highly attractive option provided it can effectively mitigate the impact of Byzantine attacks. Motivated by above factors, we investigate the realization of Federated learning employing the Normalized Gradients Algorithm on non-IID datasets and propose a novel robust aggregation algorithm named Fed-NGA. Fed-NGA normalizes the vectors offloaded from participating clients and then calculates their weighted mean to update the global model parameters. Rigorous convergence analysis is conducted for the non-convex loss function.

**Contributions:** Our main contributions are summarized as follows:

- **Algorithmically**, we propose a new aggregation algorithm, Fed-NGA, which employs normalized gradients for aggregation to ensure robustness against Byzantine attacks and data heterogeneity. The aggregation complexity is as low as $\mathcal{O}(pM)$.

- **Theoretically**, we provide a rigorous convergence analysis of the proposed Fed-NGA algorithm under non-convex loss functions. Specifically, we prove that Fed-NGA converges in the presence of Byzantine attacks, as long as the fraction of corrupted data remains below one-half, despite the presence of data heterogeneity. With regard to the convergence, we

Table 1: List of references on the convergence of FL under Byzantine attacks.

| References | Algorithm | Loss Function | Data Heterogeneity | Time Complexity for Aggregation |
|---|---|---|---|---|
| Xie et al. (2018) | median | - | IID | $\mathcal{O}(pM \log(M))$ |
| Yin et al. (2018) | trimmed mean | non-convex strongly convex | IID | $\mathcal{O}(pM \log(M))$ |
| Turan et al. (2022) | RANGE | non-convex strongly convex | IID | $\mathcal{O}(pM \log(M))$ |
| Blanchard et al. (2017) | Krum | non-convex | IID | $\mathcal{O}(pM^2)$ |
| Pillutla et al. (2022) | RFA[1] | strongly convex | non-IID | $\mathcal{O}(pM \log^3(M\epsilon^{-1}))$ |
| Wu et al. (2020) | Byrd-SAGA | strongly convex | IID | $\mathcal{O}(pM \log^3(M\epsilon^{-1}))$ |
| Zhu and Ling (2023) | BROADCAST | strongly convex | IID | $\mathcal{O}(pM \log^3(M\epsilon^{-1}))$ |
| Li et al. (2019) | RSA[2] | strongly convex | non-IID | $\mathcal{O}(Mp^{3.5})$ |
| Karimireddy et al. (2021) | CClip[3] | non-convex | IID | $\mathcal{O}(\tau Mp)$ |
| Luan et al. (2024) | MCA[4] | strongly convex | non-IID | $\mathcal{O}(pM \log^3(M\epsilon^{-1}))$ |
| **This work** | Fed-NGA | non-convex | non-IID | $\mathcal{O}(pM)$ |

[1] The commonly used Weiszfeld algorithm to calculate the geometric median does not always converge. Therefore, we use the currently fastest algorithm provided in Cohen et al. (2016) to evaluate the time complexity.
[2] The aggregation rule of RSA requires solving $M$ convex problems involving nonlinear feasible regions.
[3] $\tau$ denotes the number of times centered clipping is carried out.
[4] The algorithmic flow of MCA is highly similar to that of the geometric median, allowing us to conclude that the two algorithms share the same time complexity.

establish a convergence rate of $\mathcal{O}(1/T^{1/2-\delta})$, where $T$ denotes the number of iterations and $\delta \in (0, 1/2)$. Furthermore, under general assumptions and with a properly designed learning rate schedule, we show that the optimality gap converges to a neighborhood of stationary points. Remarkably, under certain mild conditions, Fed-NGA achieves exact convergence to stationary points, i.e., zero optimality gap, representing a significant theoretical advancement over existing Byzantine-robust federated learning methods.

- **Numerically**, we conduct extensive experiments to compare Fed-NGA with baseline methods under various setups of data heterogeneity. The results confirm the superiority of our proposed Fed-NGA in terms of test accuracy, robustness, and running time.

## 2 Problems Statement

In this section, we present Fed-NGA. We first describe the problem setup under Byzantine attacks.

### 2.1 Problem Setup

**FL optimization problem:** Consider an FL system with one central server and $M$ clients, which form the set $\mathcal{M} \triangleq \{1, 2, 3, \cdots, M\}$. For any participating client, say the $m$th client, it has a local dataset $\mathcal{S}_m$ with $S_m$ elements. The $i$th element of $\mathcal{S}_m$ is a ground-truth label $s_{m,i} = \{x_{m,i}, y_{m,i}\}$. Here, $x_{m,i} \in \mathbb{R}^{in}$ represents the input vector, and $y_{m,i} \in \mathbb{R}^{out}$ denotes the output vector. Using the dataset $\mathcal{S}_m$ for $m = 1, 2, 3, \cdots, M$, the learning task is to train a $p$-dimensional model parameter $w \in \mathbb{R}^p$ to minimize the global loss function, denoted as $F(w)$. Specifically, we aim to solve the following optimization problem:

$$\min_{w \in \mathbb{R}^p} F(w). \tag{1}$$

In (1), the global loss function $F(w)$ is defined as

$$F(w) \triangleq \frac{1}{\sum_{m \in \mathcal{M}} S_m} \sum_{m \in \mathcal{M}} \sum_{s_{m,i} \in \mathcal{S}_m} f(w, s_{m,i}), \tag{2}$$

where $f(w, s_{m,i})$ denotes the loss function to evaluate the error for approximating $y_{m,i}$ with an input of $x_{m,i}$. For convenience, we define the local loss function of the $m$th client as

$$F_m(w) \triangleq \frac{1}{S_m} \sum_{s_{m,i} \in \mathcal{S}_m} f(w, s_{m,i}) \tag{3}$$

and the weight coefficient of the $m$th client as $\alpha_m = \frac{S_m}{\sum_{i \in \mathcal{M}} S_i}, m \in \mathcal{M}$. Then the global loss function $F(w)$ is rewritten as

$$F(w) = \sum_{m \in \mathcal{M}} \alpha_m F_m(w). \tag{4}$$

Accordingly, the gradient of the loss function of $F(w)$ and $F_m(w)$ with respect to the model parameter $w$ are written as $\nabla F(w)$ and $\nabla F_m(w)$, respectively.

In the conventional FL framework, iterative interactions between a central server and a group of $M$ clients are performed to update the global model parameter $w$ until convergence, under the assumption that all clients transmit reliable messages. However, in the presence of Byzantine attacks, the key challenge in solving the optimization problem (1) lies in the ability of Byzantine clients to collude and send arbitrarily malicious updates to the server, thereby distorting the learning process. This underscores the need for a federated training framework that is robust against such adversarial behavior.

**Byzantine attack:** Based on the above FL framework, assume there are $B$ Byzantine clients out of $M$ total clients, which form the set $\mathcal{B}$. Any Byzantine client can send an arbitrary vector $\star \in \mathbb{R}^p$ to the central server. Suppose $g_m^t$ is the actual vector uploaded by the $m$th client to the central server during the $t$th round of iteration, then we have

$$g_m^t = \star, m \in \mathcal{B}. \tag{5}$$

For ease of representing the ratio of Byzantine attacks, we denote the intensity level of the Byzantine attack $\bar{C}_\alpha$, with the weight coefficient of the $m$th client, as

$$\bar{C}_\alpha \triangleq \sum_{m \in \mathcal{B}} \alpha_m = \frac{1}{\sum_{i \in \mathcal{M}} S_i} \sum_{m \in \mathcal{B}} S_m. \tag{6}$$

Intuitively, we assume $\bar{C}_\alpha < 0.5$, a common assumption in many studies Xie et al. (2018); Yin et al. (2018); Pillutla et al. (2022); Wu et al. (2020); Zhu and Ling (2023); Li et al. (2019). Accordingly, the ratio of honest clients can be defined as

$$C_\alpha = 1 - \bar{C}_\alpha. \tag{7}$$

## 2.2 Algorithm Description

To achieve our design goal, we develop Fed-NGA. In each iteration, the honest clients upload their locally trained gradient vectors to the central server, while $B$ Byzantine clients may send arbitrary vectors to bias the FL learning process. After receiving the uploaded vectors from all $M$ clients, the central server normalizes each vector and uses the normalized vectors to update the global model parameter. Once the global model parameter is updated, the central server broadcasts it to all $M$ clients in preparation for the next iteration of training. Below, we provide a full description of Fed-NGA (see Algorithm 1 and Figure 1), with its crucial steps explained in detail as follows.

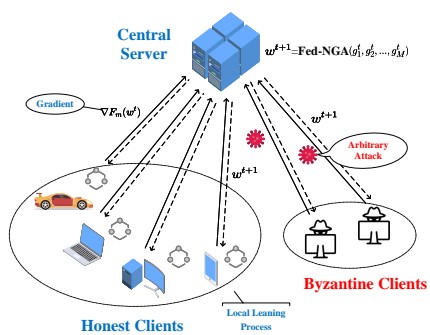

Figure 1: Illustration of the learning process of Fed-NGA on iteration number $t$.

**Local Updating:** In the $t$th round of iteration, after receiving the global model parameter $w^t$ broadcast by the central server, all honest clients $m$, where $m \in \mathcal{M} \setminus \mathcal{B}$, select a subdataset $\xi_m^t$ from their dataset $\mathcal{S}_m$ to calculate their local training gradient $\nabla F_m(w^t, \xi_m^t)$. Meanwhile, all Byzantine clients $m$, where $m \in \mathcal{B}$, may send arbitrary vectors or other malicious messages based on their dataset and the global model parameter $w^t$. Let $g_m^t$ represent the vector (either the local training gradient or a malicious message) uploaded to the central server by client $m$, and we have

$$g_m^t = \begin{cases} \nabla F_m(w^t, \xi_m^t), & m \in \mathcal{M} \setminus \mathcal{B} \\ \star, & m \in \mathcal{B} \end{cases} \tag{8}$$

---

**Algorithm 1:** Fed-NGA Algorithm

---

1: **Input:** Initial global model parameter $w^0$, clients set $\mathcal{M}$, and the number of iteration $T$.
2: **Output:** Updated global model parameter $w^T$.
3: % % **Initialization**
4: Every client $m$ establishes its own set $\mathcal{S}_m$ for $m \in \mathcal{M} \setminus \mathcal{B}$.
5: **for** $t = 0, 1, 2, \cdots, T-1$ **do**
6:     **for** every client $m \in \mathcal{M} \setminus \mathcal{B}$ in parallel **do**
7:         Receive the global model $w^t$. Select a sub-dataset $\xi_m^t$ from $\mathcal{S}_m$ to train local model and evaluate the local training gradient $\nabla F_m(w^t, \xi_m^t)$. Set $g_m^t = \nabla F_m(w^t, \xi_m^t)$ and upload $g_m^t$ to the central server.
8:     **end for**
9:     **for** every client $m \in \mathcal{B}$ in parallel **do**
10:         Receive the global model $w^t$. Generate an arbitrary vector or malicious vector $g_m^t$ based on $w^t$ and dataset $\mathcal{S}_m$. Upload this vector $g_m^t$ to the central server.
11:     **end for**
12:     Receive all uploaded vectors $g_m^t, m \in \mathcal{M}$. Normalize all uploaded vectors $g_m^t$ to generate $g^t$. Update the global model parameter $w^{t+1}$ by

$$w^{t+1} = w^t - \eta^t \cdot g^t. \tag{11}$$

13:     Broadcast the model parameter $w^{t+1}$ to all clients.
14: **end for**
15: Output the model parameter $w^T$.

---

**Aggregation and Broadcasting:** In the $t$th iteration, upon receiving the vectors $g_m^t$ from all clients $m \in \mathcal{M}$, the central server normalizes each vector by dividing its corresponding norm and updates the global model parameter $w^{t+1}$ using the learning rate $\eta^t$, as given by

$$w^{t+1} = w^t - \eta^t \sum_{m \in \mathcal{M}} \alpha_m \cdot \frac{g_m^t}{\|g_m^t\|}. \tag{9}$$

For convenience, we also define

$$g^t = \sum_{m \in \mathcal{M}} \alpha_m \cdot \frac{g_m^t}{\|g_m^t\|}, \tag{10}$$

which represents the weighted sum of the normalized local gradients from all clients in the $t$th iteration. Then, the central server broadcasts the global model parameter $w^{t+1}$ to all clients in preparation for the calculation in the $t + 1$th iteration.

## 3 Theoretical Results

In this section, we theoretically analyze the robustness and convergence performance of Fed-NGA on non-IID datasets. Below, we first present the necessary assumptions.

### 3.1 Assumption

First, we state some general assumptions for $m \in \mathcal{M} \setminus \mathcal{B}$, which are also assumed in Huang et al. (2023); Wu et al. (2023); Xiao and Ji (2023).

**Assumption 3.1** (Lipschitz Continuity). The loss function $f(w, s_{m,i})$ has $L$-Lipschitz continuity, i.e., for $\forall w_1, w_2 \in \mathbb{R}^p$, it follows that

$$f(w_1, s_{m,i}) - f(w_2, s_{m,i}) \leqslant \langle \nabla f(w_2, s_{m,i}), w_1 - w_2 \rangle + \frac{L}{2} \|w_1 - w_2\|^2. \tag{12}$$

Under Assumption 3.1, given that the local loss functions $F_m(w)$ are linear combinations of the loss function $f(w, s_{m,i})$ Huang et al. (2023), it can be rigorously deduced that they all exhibit Lipschitz

continuity. By an analogous line of reasoning, the global loss function can also be shown to satisfy Lipschitz continuity. A notable characteristic of Lipschitz continuous functions is that they can be non-convex, which provides flexibility in the analysis and optimization of these functions.

**Assumption 3.2** (Local Unbiased Gradient). For $\xi_m \subseteq \mathcal{S}_m$, the gradient of local training loss function $F_m(w, \xi_m)$ is unbiased, which implies that

$$\mathbb{E}\{\nabla F_m(w, \xi_m)\} = \nabla F_m(w) \tag{13}$$

This assumption is widely adopted in the literature Huang et al. (2023); Wu et al. (2023); Xiao and Ji (2023), and $\nabla F_m(w, \xi_m)$ reduces to the exact gradient $\nabla F_m(w)$ when $\xi_m = \mathcal{S}_m$.

**Assumption 3.3** (Bounded Inner Error: Type I). For $\forall w \in \mathbb{R}^p$, the inner error of gradients is uniformly bounded, i.e.,

$$\mathbb{E}\{\|\nabla F_m(w, \xi_m) - \nabla F_m(w)\|\} \leqslant \sigma. \tag{14}$$

This assumption is also assumed in Huang et al. (2023); Wu et al. (2023); Xiao and Ji (2023).

**Assumption 3.4** (Bounded Data Heterogeneity: Type I). We define a representation of data heterogeneity by inspecting gradient direction, i.e.,

$$\theta_m = \|\nabla F_m(w) - \nabla F(w)\|. \tag{15}$$

We also assume the data heterogeneity is bounded, which implies

$$\theta_m \leqslant \theta, \tag{16}$$

where $\theta$ is the heterogeneity upper bound.

This assumption is also assumed in Huang et al. (2023); Wu et al. (2023); Xiao and Ji (2023).

Building upon the aforementioned general assumptions, we introduce two refined variants, which are demonstrated to be reasonable in Appendix A. Their relationships with Assumptions 3.3 and 3.4 are also discussed therein.

**Assumption 3.5** (Bounded Inner Error: Type II). For $\forall w \in \mathbb{R}^p$, an alternative bound on the inner gradient error is presented as follows:

$$\mathbb{E}\left\{\left\|\frac{\nabla F_m(w, \xi_m)}{\|\nabla F_m(w, \xi_m)\|} - \frac{\nabla F_m(w)}{\|\nabla F_m(w)\|}\right\|\right\} \leqslant \sigma'. \tag{17}$$

**Assumption 3.6** (Bounded Data Heterogeneity: Type II). For $m \in \mathcal{M}$, we define another representation of data heterogeneity by inspecting gradient direction, i.e.,

$$\theta'_m = \left\|\frac{\nabla F_m(w)}{\|\nabla F_m(w)\|} - \frac{\nabla F(w)}{\|\nabla F(w)\|}\right\|. \tag{18}$$

Then the bounded data heterogeneity implies

$$\theta'_m \leqslant \theta', \tag{19}$$

where $\theta'$ is the heterogeneity upper bound.

### 3.2 Convergence Analysis

In this subsection, we present the convergence analysis of the proposed Fed-NGA algorithm for non-convex loss functions that satisfy Assumption 3.1. All the proofs are deferred to Appendix E and Appendix F.

#### 3.2.1 With Assumption 3.3 and 3.4 on the loss function

**Theorem 3.7.** *With Assumption 3.1, 3.2, 3.3, 3.4, and $\rho > 0$ such that $\frac{2\rho}{\rho+1}C_\alpha - 1 > 0$, we have*

$$\frac{1}{\sum_{t=0}^{T-1} \eta^t} \sum_{t=0}^{T-1} \left(\frac{2\rho}{\rho+1}C_\alpha - 1\right) \eta^t \|\nabla F(w^t)\| \leq \frac{\mathbb{E}\{F(w^0) - F(w^T)\}}{\sum_{t=0}^{T-1} \eta^t} + \frac{L\sum_{t=0}^{T-1}(\eta^t)^2}{2\sum_{t=0}^{T-1} \eta^t} + \frac{2\rho^2}{\rho+1}C_\alpha(\sigma + \theta) \tag{20}$$

*Proof.* Please refer to Appendix E. □

**Remark 3.8.** For the case with Assumption 3.3 and Assumption 3.4, the bound of optimality gap and convergence rate in Theorem 3.7 reveal the impact of learning rate $\eta^t$, the ratio of Byzantine attack $\bar{C}_\alpha$ (i.e., $1 - C_\alpha$), Type I inner error bound $\sigma$, and Type I data heterogeneity bound $\theta$ on the convergence performance for our proposed Fed-NGA. With the above results, to reduce the optimality gap and speed up the convergence rate, we need to first select an appropriate learning rate $\eta^t$. Then we analyze the robustness of Fed-NGA to Byzantine attacks on non-IID datasets under our selected learning rate. All the discussions are given as follows:

- With regard to the optimality gap, when $\lim_{T \to \infty} \sum_{t=1}^{T-1} \eta^t \to \infty$ and $\lim_{T \to \infty} \sum_{t=1}^{T-1} (\eta^t)^2 < \infty$, together with the prerequisite condition of Theorem 3.7, it can be derived that the average iteration error $\frac{1}{\sum_{t=0}^{T-1} \eta^t} \sum_{t=0}^{T-1} \left( \frac{2\rho}{\rho+1} C_\alpha - 1 \right) \eta^t \|\nabla F(w^t)\|$ converges to $\mathcal{O}(\sigma + \theta)$. Therefore, $\min_t \|\nabla F(w^t)\|$ will converge to $\mathcal{O}(\sigma + \theta)$.

- To further concretize the convergence rate, we set $\eta^t = \eta/(t+c)^{1/2+\delta}, c > 0$, where $\delta$ is an arbitrary value lying between $(0, 1/2)$. Then, the right-hand side of (20) is upper bounded by

$$\mathcal{O}\left( \frac{\mathbb{E}\{F(w^0) - F(w^T)\}}{\eta T^{\frac{1}{2}-\delta}} \right) + \mathcal{O}\left( \frac{L\eta(1 + 2\delta - 2\delta T^{-2\delta})}{2T^{\frac{1}{2}-\delta}} \right) + \mathcal{O}\left( \frac{2\rho^2}{\rho+1} C_\alpha(\sigma + \theta) \right), \quad (21)$$

  which still converges to $\mathcal{O}\left( \frac{2\rho^2}{\rho+1} C_\alpha(\sigma + \theta) \right)$ as $T \to \infty$, at a rate of $\mathcal{O}\left( 1/T^{\frac{1}{2}-\delta} \right)$.

- The constraint $\frac{2\rho}{\rho+1} C_\alpha - 1 > 0$ and the term $\frac{2\rho^2}{\rho+1} C_\alpha(\sigma + \theta)$, which characterizes the optimality gap, reveal a trade-off between the optimality gap and the tolerable ratio of Byzantine clients in the proposed Fed-NGA algorithm. In the extreme case where there are no Byzantine clients (i.e., $C_\alpha = 1$), the optimality gap can be reduced to as small as $\sigma + \theta$.

### 3.2.2 With Assumption 3.5 and 3.6 on the loss function

**Theorem 3.9.** *With Assumption 3.1, 3.5, 3.6, and $\left( 2 - (\theta')^2/2 - \sigma' \right) C_\alpha - 1 > 0$, we have*

$$\frac{1}{\sum_{t=0}^{T-1} \eta^t} \sum_{t=0}^{T-1} \eta^t \|\nabla F(w^t)\| \leqslant \frac{\mathbb{E}\{F(w^0) - F(w^T)\}}{\left( (2 - \frac{(\theta')^2}{2} - \sigma')C_\alpha - 1 \right) \sum_{t=0}^{T-1} \eta^t} + \frac{L \sum_{t=0}^{T-1} (\eta^t)^2}{2 \left( (2 - \frac{(\theta')^2}{2} - \sigma')C_\alpha - 1 \right) \sum_{t=0}^{T-1} \eta^t}.$$
$$(22)$$

*Proof.* Please refer to Appendix F. □

**Remark 3.10.** For the case based on Assumption 3.5 and 3.6, the bound of optimality gap and convergence rate in Theorem 3.9 reveal the impact of learning rate $\eta^t$, the ratio of Byzantine attack $\bar{C}_\alpha$, Type II inner error bound $\sigma'$, and Type II data heterogeneity bound $\theta'$ on the convergence performance for our proposed Fed-NGA. With the above results, to reduce the optimality gap and speed up the convergence rate, we first select an appropriate learning rate $\eta^t$, and then analyze the associated robustness of Fed-NGA to Byzantine attacks on non-IID datasets, which are given as follows:

- With regard to the optimality gap, when $\lim_{T \to \infty} \sum_{t=1}^{T-1} \eta^t \to \infty$ and $\lim_{T \to \infty} \sum_{t=1}^{T-1} (\eta^t)^2 < \infty$, together with the prerequisite condition of Theorem 3.9, it can be derived that the average iteration error $\left( 1/\sum_{t=0}^{T-1} \eta^t \right) \cdot \sum_{t=0}^{T-1} \eta^t \|\nabla F(w^t)\|$ converges to 0.

- To further concretize the convergence rate, we set $\eta^t = \eta/(t+c)^{1/2+\delta}, c > 0$, where $\delta$ is an arbitrary value lying between $(0, 1/2)$. Then, the right-hand side of (22) is upper bounded

Table 2: The maximum test accuracy (%) for Fed-NGA and baselines is evaluated under different types of Byzantine attacks, with the concentration parameter $\beta = 0.6$, with TinyImageNet ($\bar{C}_\alpha = 0.2$, MobileNetV3), CIFAR10 ($\bar{C}_\alpha = 0.3$, LeNet), and MNIST ($\bar{C}_\alpha = 0.4$, MLP) datasets.

| Dataset | TinyImageNet | | | | | | CIFAR10 | | | | | | MNIST | | | | | |
|---|---|---|---|---|---|---|---|---|---|---|---|---|---|---|---|---|---|---|
| Attack Name | No Attack | Gaussian | Same-value | Sign-flip | LIE | FoE | No Attack | Gaussian | Same-value | Sign-flip | LIE | FoE | No Attack | Gaussian | Same-value | Sign-flip | LIE | FoE |
| Fed-NGA | **56.95** | **55.77** | **49.31** | **45.38** | **55.82** | **45.23** | **54.48** | **52.07** | **29.16** | **51.16** | **51.82** | **51.16** | **96.72** | **94.98** | **83.66** | **94.71** | 94.92 | **94.71** |
| FedAvg | 55.67 | 46.27 | 0.54 | 0.50 | 48.16 | 0.50 | 49.11 | 10.40 | 10.00 | 10.00 | 10.00 | 10.00 | 95.32 | 16.70 | 11.35 | 9.82 | 78.43 | 9.82 |
| Median | 48.45 | 45.87 | 41.14 | 21.41 | 46.67 | 38.60 | 39.98 | 39.64 | 10.70 | 38.87 | 39.75 | 14.19 | 92.61 | 92.57 | 61.60 | 92.33 | 92.59 | 68.09 |
| Krum | 36.22 | 36.76 | 30.68 | 36.31 | 36.09 | 0.50 | 40.76 | 39.55 | 10.11 | 39.67 | 39.96 | 14.02 | 92.60 | 92.73 | 60.63 | 92.28 | 92.71 | 67.50 |
| GM | 55.68 | 54.22 | 48.98 | 43.14 | 54.32 | 42.88 | 48.41 | 48.08 | 28.81 | 47.89 | 47.90 | 10.47 | 94.67 | 94.46 | 70.11 | 94.31 | 94.33 | 11.12 |
| MCA | 55.62 | 54.22 | 1.70 | 0.50 | 54.34 | 0.50 | 48.67 | 48.53 | 10.00 | 10.00 | 48.28 | 10.00 | 95.10 | 94.82 | 11.35 | 11.35 | 94.84 | 11.35 |
| CClip | 50.16 | 45.99 | 43.51 | 38.38 | 45.48 | 37.25 | 48.59 | 43.42 | 10.00 | 10.00 | 47.79 | 10.00 | 94.99 | 94.87 | 10.09 | 9.80 | **94.93** | 9.80 |

by

$$\mathcal{O}\left(\frac{\mathbb{E}\left\{F(w^0) - F(w^T)\right\}}{\left(\left(2 - \frac{(\theta')^2}{2} - \sigma'\right)C_\alpha - 1\right)\eta T^{\frac{1}{2}-\delta}}\right) + \mathcal{O}\left(\frac{L\eta(1 + 2\delta - 2\delta T^{-2\delta})}{2\left(\left(2 - \frac{(\theta')^2}{2} - \sigma'\right)C_\alpha - 1\right)T^{\frac{1}{2}-\delta}}\right), \quad (23)$$

which still converges to 0 as $T \to \infty$, at a rate of $\mathcal{O}\left(1/T^{\frac{1}{2}-\delta}\right)$.

- As discussed in Appendix A, in the ideal scenario where both the inner error and data heterogeneity remain uniformly small throughout the entire training process, or in the case of full-batch training on IID datasets (i.e., $\sigma' = 0$ and $\theta' = 0$), such that Assumptions 3.5 and 3.6 hold at all training iterations, convergence to a stationary point with zero optimality gap can be rigorously guaranteed. In more general scenarios, Assumptions 3.5 and 3.6, as well as Assumptions 3.3 and 3.4, can be adopted in an alternating manner. Specifically, during the early stages of training when gradient norms are typically large, Assumptions 3.5 and 3.6 are more likely to be satisfied, thereby guiding the learning process toward convergence with zero optimality gap. As the model approaches convergence and gradient norms become smaller, Assumptions 3.3 and 3.4 become more applicable and continue to ensure convergence.

## 4 Experiments

### 4.1 Setup

**Datasets, Models and Hyperparameters:** Our experiments are conducted on the TinyImageNet, CIFAR10, and MNIST datasets, utilizing the MobileNetV3 (Howard et al., 2019), LeNet (LeCun et al., 1998), and Multilayer Perceptron (MLP) (Yue et al., 2022) models. For the non-IID settings, we adopt the Dirichlet ($\beta$) distribution, where the label distribution on each device follows a Dirichlet distribution with $\beta > 0$ as the concentration parameter. Additionally, we consider five types of Byzantine attacks to bias the FL training process. Further details are provided in Appendix B.

**Baselines:** The convergence performance of six methods (Fed-NGA, FedAvg, Median, Krum, Geometric Median (GM), MCA (Luan et al., 2024)), and CClip (Karimireddy et al., 2021) is compared. Among these, FedAvg is renowned in traditional FL and can be used as a performance metric with no Byzantine attacks. Median, Krum, GM, MCA, and CClip utilize coordinate-wise median, Krum, geometric median, maximum correntropy aggregation, and centered clipping, respectively, to update the global model parameters over the uploaded messages.

**Metrics:** To evaluate the performance of Fed-NGA, we compare it with other baselines by measuring both the test accuracy and the running time of the parameter aggregation process (excluding model training time at local, see Appendix B for implementation specifics). Higher test accuracy indicates better performance, while a lower running time reflects greater efficiency in the aggregation process.

### 4.2 Results for Convergence Performance

We begin with Table 2 and Table 3, which summarize the maximum test accuracy and running time (excluding model training time at local) of Fed-NGA and baselines under five types of Byzantine attacks for three learning models. Figure 2 shows the maximum test accuracy with $\beta = 0.6$ on CIFAR10 dataset and LeNet model. Figure 3 presents the running time of Fed-NGA and the baselines

Table 3: The running time (in seconds) of Fed-NGA and robust baselines is evaluated under different types of Byzantine attacks, with the concentration parameter of $\beta = 0.6$, with TinyImageNet ($\bar{C}_\alpha = 0.2$, MobileNetV3), CIFAR10 ($\bar{C}_\alpha = 0.3$, LeNet), and MNIST ($\bar{C}_\alpha = 0.4$, MLP) datasets.

| Dataset | TinyImageNet | | | | | | CIFAR10 | | | | | | MNIST | | | | | |
|---|---|---|---|---|---|---|---|---|---|---|---|---|---|---|---|---|---|---|
| Attack Name | No Attack | Gaussian | Same-value | Sign-flip | LIE | FoE | No Attack | Gaussian | Same-value | Sign-flip | LIE | FoE | No Attack | Gaussian | Same-value | Sign-flip | LIE | FoE |
| **Fed-NGA** | **0.2260** | **0.1942** | **0.1951** | **0.2011** | **0.1912** | **0.3367** | **0.0558** | **0.0491** | **0.0454** | **0.0557** | **0.0422** | **0.0557** | **0.0484** | **0.0497** | **0.0443** | **0.0421** | **0.0408** | **0.0421** |
| Median | 0.3333 | 0.2927 | 0.2929 | 0.2974 | 0.2893 | 0.3732 | 22.163 | 31.063 | 32.196 | 30.416 | 31.061 | 30.357 | 4.5015 | 5.2404 | 5.4980 | 5.3578 | 5.1514 | 4.6999 |
| Krum | 50.902 | 49.322 | 49.567 | 50.199 | 49.830 | 50.391 | 21.773 | 31.079 | 32.516 | 31.037 | 31.052 | 30.923 | 4.4100 | 5.4015 | 7.7471 | 5.2540 | 4.9461 | 4.4994 |
| GM | 5.1880 | 4.9234 | 5.4841 | 6.9124 | 5.8732 | 11.497 | 7.8912 | 15.172 | 26.252 | 18.994 | 14.140 | 36.934 | 1.6423 | 3.5852 | 27.2812 | 4.0717 | 2.7088 | 13.8719 |
| MCA | 3.5246 | 2.9496 | 5.2586 | 6.6065 | 4.0824 | 4.6278 | 6.3659 | 7.3452 | 25.098 | 1369.4 | 7.3201 | 1369.4 | 1.2252 | 1.4916 | 12.4749 | 324.53 | 1.2387 | 324.53 |
| CClip | 2.5082 | 2.2738 | 2.2174 | 3.5504 | 3.1020 | 3.5263 | 5.6009 | 4.6567 | 16.016 | 1006.7 | 4.6273 | 1006.7 | 1.5097 | 1.2726 | 4.7536 | 794.37 | 1.2541 | 794.37 |

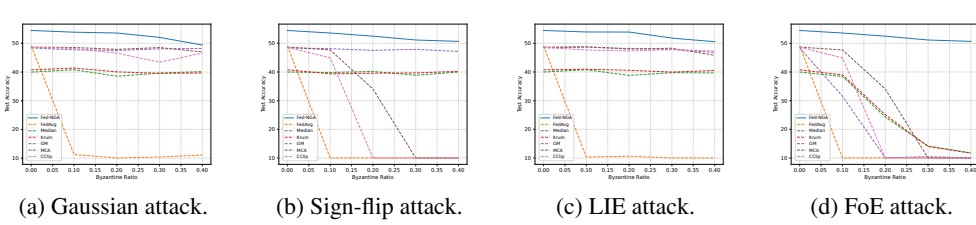

(a) Gaussian attack.     (b) Sign-flip attack.     (c) LIE attack.     (d) FoE attack.

Figure 2: The maximum test accuracy (%) for Fed-NGA and baselines with $\beta = 0.6$ on CIFAR10 dataset and LeNet model.

under Same-value attacks on TinyImageNet and MNIST datasets. Finally, Figure 4 demonstrates the convergence of Fed-NGA on the TinyImageNet dataset across three distinct data heterogeneity concentration parameters. Additional experimental results and detailed performance analysis are provided in Appendix C.

**Byzantine Attack Robustness Analysis:** Table 2 quantifies Fed-NGA's superior robustness on TinyImageNet across pristine (no-attack) and five Byzantine attack scenarios. Fed-NGA achieves minimum accuracy improvements of 1.27% in no-attack environments and peak gains of 2.35% under Byzantine attacks, demonstrating remarkable resistance to Sign-flip and FoE attacks. On CIFAR10 with $\bar{C}_\alpha = 0.3$ and $\beta = 0.6$, Fed-NGA attains SoTA with accuracy enhancements ranging from 0.35% to 5.37%. For MNIST, while showing marginal parity (-0.01%) with baselines under LIE attacks, Fed-NGA delivers substantial improvements of 13.55% and 26.62% against Same-value and FoE attacks respectively. Figure 2 further validates these findings, illustrating Fed-NGA's consistent defensive efficacy across varying attack intensities.

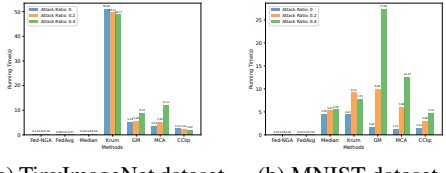

(a) TinyImageNet dataset.     (b) MNIST dataset.

Figure 3: The running time (s) for Fed-NGA and baselines is evaluated on Same-value attack and $\beta = 0.6$ across three different Byzantine ratios.

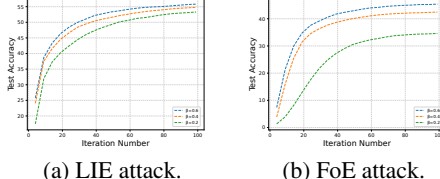

(a) LIE attack.     (b) FoE attack.

Figure 4: The maximum test accuracy (%) for Fed-NGA is evaluated on TinyImageNet dataset and $\bar{C}_\alpha = 0.2$ across three different data heterogeneity concentration parameters.

**Running Time:** From Table 3, Fed-NGA demonstrates the shortest parameter aggregation time among robust baselines across all datasets and model architectures. For the TinyImageNet dataset, Fed-NGA establishes state-of-the-art (SoTA) performance, achieving the lowest aggregation latency under all Byzantine attack scenarios. The efficiency gap becomes even more pronounced on CIFAR10, where baseline methods require at least 100× longer aggregation time than Fed-NGA for LeNet. When applied to simpler datasets like MNIST, other robust baselines exhibit aggregation times at least 20× longer than those of Fed-NGA. As shown in Figure 3, Fed-NGA requires only 3× the aggregation time of FedAvg while maintaining Byzantine resilience. These results collectively

demonstrate Fed-NGA's significant advantages in computational efficiency compared to existing robust aggregation algorithms.

**Data Heterogeneity:** Figure 4 illustrates the relationship between data heterogeneity and Fed-NGA's classification performance on TinyImageNet with $\bar{C}_\alpha = 0.2$. The results show an inverse correlation between the concentration parameter $\beta$ and test accuracy: As $\beta$ decreases (indicating greater data heterogeneity), Fed-NGA's performance gradually declines. Furthermore, the results exhibit differential sensitivity to data heterogeneity depending on attack type.

## 5   Conclusion

In this paper, we proposed Fed-NGA, a simple yet effective algorithm that addresses the dual challenges of Byzantine robustness and data heterogeneity in FL. By leveraging normalized gradients and a lightweight aggregation rule, Fed-NGA significantly reduces computational overhead while maintaining strong theoretical guarantees. Our analysis establishes convergence to a neighborhood of stationary points under general assumptions, and to a true stationary point under some other mild conditions, with a provable convergence rate of $\mathcal{O}(1/T^{1/2-\delta})$. Empirical evaluations on standard benchmarks validate the practical benefits of Fed-NGA in terms of both time efficiency and robustness. With regard to limitation, more general assumptions may be required to guarantee a zero optimality gap, although this work is the first to prove it for Byzantine-resistant FL algorithms but under some mild conditions.

## 6   Acknowledgment

This work was supported by the National Natural Science Foundation of China Nos.62171034, U2336211, 62471424, 62576364, 92267202, the Guangdong Provincial Key Laboratory of Future Networks of Intelligence No.2022B1212010001.

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

# A The Reasonableness of Assumptions 3.5 and 3.6

To justify Assumptions 3.5 and 3.6, we observe that the direction of the gradient conveys more critical information than its norm, as the impact of the norm can be modulated by the learning rate $\eta^t$. Motivated by this insight, we redefine the notions of inner error bound and data heterogeneity bound purely from the perspective of gradient direction, leading to the concepts of Type II inner error and Type II data heterogeneity, as formalized in Assumptions 3.5 and 3.6.

Regarding the relationship between Assumptions 3.3 and 3.5 (i.e., Type I and Type II inner error bound), as well as Assumptions 3.4 and 3.6 (i.e., Type I and Type II data heterogeneity bound), we offer the following insights. As illustrated in Figure 5, when the norms of both vectors $x_1$ and $x_2$ are greater than 1, it can be observed that $\|x_1 - x_2\|$ exceeds $\left\|\frac{x_1}{\|x_1\|} - \frac{x_2}{\|x_2\|}\right\|$. Therefore, if $\|x_1 - x_2\| \leq \sigma_0$, then the normalized distance is also bounded by $\sigma_0$. Conversely, when $\|x_1\| < 1$ and $\|x_2\| < 1$, the $\left\|\frac{x_1}{\|x_1\|} - \frac{x_2}{\|x_2\|}\right\|$ dominates and hence provides a

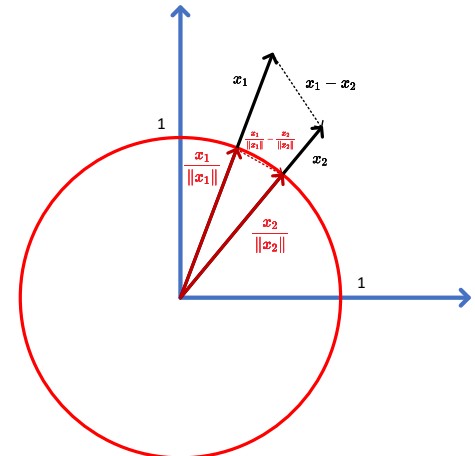

Figure 5: The illustration depicts the vectors $x_1$, $x_2$, $x_1 - x_2$, $\frac{x_1}{\|x_1\|}$, $\frac{x_2}{\|x_2\|}$, and $\frac{x_1}{\|x_1\|} - \frac{x_2}{\|x_2\|}$, with the radius of the circle set to 1.

tighter bound on $\|x_1 - x_2\|$. Motivated by this observation, we map the gradient vectors $\nabla F(w)$, $\nabla F_m(w)$, and $\nabla F_m(w, \xi_m)$ to $x_1$ and $x_2$, and draw the following implications: When the norms of the concerned gradients are relatively large (e.g., greater than 1), Assumptions 3.3 and 3.4 are more restrictive. In contrast, when the norms of concerned gradients are small, Assumptions 3.5 and 3.6 become more restrictive. In practice, the selection of the appropriate assumption set should be guided by the behavior and scale of gradient norms during different stages of training.

Reviewing the training process, gradient norms are typically large during the early stages and gradually decrease to below 1 as training progresses. Consequently, in the initial phase, Assumptions 3.5 and 3.6, being more relaxed, are more likely to hold, thereby guiding the training toward convergence with zero optimality gap. As the model nears convergence and the gradient norms diminish, Assumptions 3.3 and 3.4 become more suitable and also ensure convergence. Notably, if the inner error and data heterogeneity remain consistently small throughout the entire training process, i.e., Assumptions 3.5 and 3.6 hold at all iterations, then convergence to a stationary point with zero optimality gap can be rigorously guaranteed.

# B Experimental Setups in Detail

To carry out our experiments, we set up a machine learning environment in PyTorch 2.2.2 on Ubuntu 20.04, powered by four RTX A6000 GPUs and one AMD 7702 CPU. Firstly, we describe the datasets as below:

**Datasets:**

- **MNIST:** MNIST dataset includes a training set and a test set. The training set contains 60000 samples and the test set contains 10000 samples, each sample of which is a $28 \times 28$ pixel grayscale image.

- **CIFAR10:** The CIFAR10 dataset includes a training set and a test set. The training set contains 50,000 samples, and the test set contains 10,000 samples, each of which is a $32 \times 32$ pixel color image.

- **TinyImageNet:** The TinyImageNet dataset consists of a training set, a validation set, and a test set. The training set includes 100,000 samples, while both the validation set and the test set contain 10,000 samples each. Each sample is a 64 × 64 pixel color image.

We split the above three datasets into $M$ non-IID training sets, which is realized by letting the label of data samples to conform to Dirichlet distribution. The extent of non-IID can be adjusted by tuning the concentration parameter $\beta$ of Dirichlet distribution.

**Models:** We adopt LeNet, Multilayer Perceptron (MLP), and MobileNetV3 models, respectively. The introduction of these three models is as follows:

- **LeNet:** The LeNet model is one of the earliest published convolutional neural networks. For the experiments, like LeCun et al. (1998), we are going to train a LeNet model with two convolutional layers (both kernel sizes are 5 and the out channel of first one is 6 and second one is 16), two pooling layers (both kernel size are 2 and stride are 2), and three fully connected layer (the first one is (16 * 4 * 4, 120), the second is (120, 60) and the last is (60, 10)) for MNIST dataset. And for CIFAR10 dataset, we are going to train a LeNet model with two convolutional layers (both kernel size are 5 and the out channels are 64), two pooling layers (both kernel size are 2 and stride are 2), and three fully connected layers (the first one is (64 * 5 * 5, 384), the second is (384, 192) and the last is (192, 10)). Cross-entropy function is taken as the training loss.

- **MLP:** The MLP model is a machine learning model based on Feedforward Neural Network that can achieve high-level feature extraction and classification. We configure the MLP model to be with three connected layers (the first one is (28 * 28, 200), the second is (200, 100) and the last is (100, 10)). And for CIFAR10 dataset, we configure the MLP model to be with three connected layers (the first one is (3* 32 * 32, 200), the second is (200, 200) and the last is (200, 10)) like Yue et al. (2022). And also cross-entropy function is taken as the training loss.

- **MobileNetV3:** MobileNetV3 is a lightweight convolutional neural network (CNN) meticulously optimized for mobile and embedded devices. It integrates depthwise separable convolutions with Neural Architecture Search (NAS) to enable efficient feature extraction and classification under strict computational constraints, with its detailed architectural design documented in Howard et al. (2019). For specific dataset adaptability, we conducted fine-tuning to optimize its performance on the TinyImageNet dataset, ensuring robust feature learning across its 200-class image corpus.

**Hyperparameters:** We set $M = 50$ and fix the batch size at 32 across all experiments. For numerically computing the GM and MCA, the error tolerance is defined as $\epsilon = 1 \times 10^{-5}$. The concentration parameter $\beta$ takes values 0.6, 0.4, and 0.2. The number of iterations is configured as 500 for the MNIST and CIFAR10 datasets, and 100 for the TinyImageNet dataset.

Regarding learning rates:

- On MNIST dataset, Fed-NGA employs $\eta^t = \frac{1}{2\sqrt{0.002t+1}}$, while baseline methods use $\eta^t = \frac{1}{2\sqrt{0.198t+1}}$.

- For CIFAR10 dataset, Fed-NGA uses $\eta^t = \frac{1}{\sqrt{0.002t+1}}$, with baselines adopting $\eta^t = \frac{1}{2\sqrt{0.198t+1}}$.

- On TinyImageNet dataset, Fed-NGA uses $\eta^t = \frac{1.2}{\sqrt{0.01t+1}}$, whereas baselines use $\eta^t = \frac{1}{100\sqrt{0.01t+1}}$.

These divergent learning rate schedules stem from Fed-NGA's gradient normalization mechanism, which introduces variations in gradient magnitudes across iterations. Using Fed-NGA's learning rates for baseline methods leads to substantially degraded performance, necessitating tailored adjustments to maintain stability and convergence. For time complexity benchmarking, we implement distinct numerical computation frameworks tailored to time complexity: **TensorFlow tensor operations** accelerate MobileNetV3's execution profiling due to its deep architecture, while **NumPy array computations** handle the lightweight LeNet and MLP models.

**Byzantine Attacks:** The ratio of Byzantine attacks, $\bar{C}_\alpha$, is set to 0, 0.1, 0.2, 0.3, and 0.4. We select five types of Byzantine attacks, which are introduced as follows,

- **Gaussian attack:** All Byzantine attacks are selected as the Gaussian attack, which obeys $\mathcal{N}(0, 81)$.

- **Same-value attack:** Each dimension of the Byzantine clients' uploaded vector is set to 1.

- **Sign-flip attack:** All Byzantine clients upload $-3 * \sum_{m \in \mathcal{M} \setminus \mathcal{B}} g_m^t$ to the central server on iteration number $t$.

- **LIE attack (Baruch et al., 2019):** LIE attack adds small amounts of noise to each dimension of the benign gradients. The noise is controlled by a coefficient $c$, which enables the attack to evade detection by robust aggregation methods while negatively impacting the global model. Specifically, the attacker calculates the mean $\rho$ and standard deviation $\nu$ of the parameters submitted by honest users, calculates the coefficient $c$ based on the total number of honest and malicious clients, and finally computes the malicious update as $\rho + c\nu$. We set $c$ to 0.7.

- **FoE attack (Xie et al., 2020):** The FoE attack enables Byzantine clients to upload $\frac{q}{M-B} \sum_{\mathcal{M} \setminus \mathbf{B}} g_m^t$ to disrupt the FL training process. The coefficient $q$ is configured differently based on the specific attack and algorithm. For FedAvg, we set $q = -3 * (M - B)$. For Median, Krum, and GM, we set $q = -0.1$. For MCA and CClip, we also set $q = -3 * (M - B)$. In the case of our proposed Fed-NGA, since the uploaded vectors are normalized, the value of $q$ does not influence the actual aggregation. Nonetheless, we set $q = -3 * (M - B)$ for consistency.

## C    Results for Convergence Performance in Detail

### C.1    Training Performance for Different Byzantine Attacks

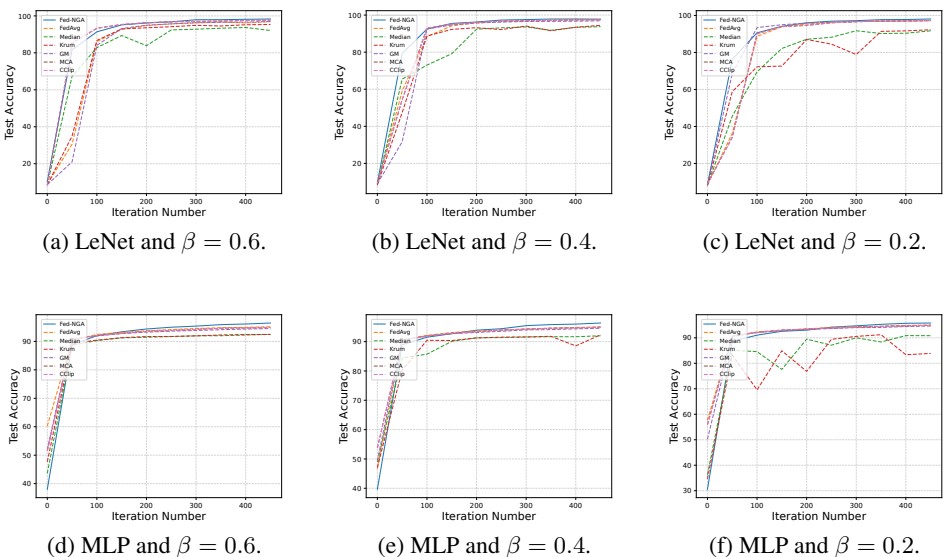

(a) LeNet and $\beta = 0.6$.  (b) LeNet and $\beta = 0.4$.  (c) LeNet and $\beta = 0.2$.

(d) MLP and $\beta = 0.6$.  (e) MLP and $\beta = 0.4$.  (f) MLP and $\beta = 0.2$.

Figure 6: The test accuracy (%) over 500 iterations for Fed-NGA and baselines is evaluated without Byzantine attacks on MNIST dataset.

In this section, we analyze the training performance under various Byzantine attacks, referencing Table 4, Table 5 and Table 6, as well as Figure 6, Figure 7, Figure 8, Figure 9, Figure 10, and Figure 11. Tables 4, Table 5, and Table 6 summarize the maximum test accuracy for Fed-NGA and baselines under four Byzantine ratios, three data heterogeneity concentration parameter setups, five types of Byzantine attacks, and three learning models, applied to the MNIST, CIFAR10, and TinyImageNet

Table 4: The maximum test accuracy (%) of 500 iterations of Fed-NGA and baselines with different types, ratios of Byzantine attack, and concentration parameter on MNIST dataset and two learning models.

| Model | β | Algorithm ($\bar{C}_\alpha$) | No Attack 0 | Gaussian Attack 0.1 | 0.2 | 0.3 | 0.4 | Same-value Attack 0.1 | 0.2 | 0.3 | 0.4 | Sign-flip Attack 0.1 | 0.2 | 0.3 | 0.4 | LIE Attack 0.1 | 0.2 | 0.3 | 0.4 | FoE Attack 0.1 | 0.2 | 0.3 | 0.4 |
|---|---|---|---|---|---|---|---|---|---|---|---|---|---|---|---|---|---|---|---|---|---|---|---|
| LeNet | 0.6 | **Fed-NGA** | **98.50** | **98.39** | **98.17** | **98.02** | **97.48** | 97.00 | 93.96 | 88.27 | **81.24** | **98.24** | **98.16** | **98.02** | **97.52** | **98.37** | **98.18** | **98.00** | 97.64 | **98.24** | **98.16** | **98.02** | **97.52** |
| | | FedAvg | 97.25 | 11.35 | 11.40 | 12.23 | 11.35 | 11.35 | 11.35 | 11.35 | 11.35 | 11.01 | 11.35 | 9.82 | 11.95 | 95.27 | 93.89 | 89.38 | 87.33 | 9.80 | 11.35 | 9.82 | 11.95 |
| | | Median | 94.04 | 95.42 | 95.12 | 95.22 | 95.55 | 95.16 | 91.51 | 73.17 | 43.97 | 93.86 | 94.82 | 94.71 | 94.66 | 95.50 | 95.07 | 95.47 | 95.52 | 93.48 | 89.55 | 73.53 | 8.92 |
| | | Krum | 95.72 | 95.04 | 95.37 | 95.49 | 95.82 | 93.75 | 90.32 | 76.43 | 39.65 | 94.44 | 94.97 | 95.00 | 95.09 | 94.97 | 94.89 | 94.98 | 94.86 | 94.21 | 90.44 | 78.37 | 8.92 |
| | | GM | 97.04 | 97.12 | 97.26 | 97.11 | 97.48 | 96.22 | **94.52** | **91.84** | 78.76 | 97.19 | 96.82 | 96.81 | 97.28 | 96.90 | 97.40 | 97.17 | 97.47 | 94.77 | 7.61 | 8.88 | 8.92 |
| | | MCA | 97.69 | 96.10 | 97.78 | 97.61 | 93.92 | **97.87** | 11.35 | 11.35 | 11.35 | 97.68 | 11.86 | 9.80 | 10.09 | 98.07 | 97.82 | 97.66 | 97.56 | 97.68 | 11.86 | 9.80 | 10.09 |
| | | CClip | 97.54 | 93.79 | 97.71 | 97.20 | 96.91 | 97.26 | 11.35 | 11.35 | 11.35 | 97.33 | 96.60 | 9.80 | 9.80 | 95.80 | 97.68 | 97.74 | **97.67** | 97.33 | 96.60 | 9.80 | 9.80 |
| | 0.2 | **Fed-NGA** | **98.27** | **98.16** | **97.87** | **97.39** | **97.19** | 96.59 | 92.05 | 84.59 | **79.30** | **98.08** | **97.87** | **97.43** | **96.81** | **98.16** | **97.89** | **97.49** | 97.22 | **98.08** | **97.87** | **97.43** | **96.81** |
| | | FedAvg | 97.42 | 11.35 | 11.35 | 11.35 | 11.35 | 11.35 | 11.35 | 11.35 | 10.09 | 30.19 | 11.35 | 10.32 | 10.09 | 95.95 | 94.27 | 91.16 | 88.63 | 30.19 | 11.35 | 10.32 | 10.09 |
| | | Median | 92.95 | 93.37 | 93.97 | 94.30 | 93.70 | 90.78 | 82.59 | 56.53 | 10.43 | 91.90 | 93.18 | 93.65 | 92.20 | 93.24 | 93.19 | 93.79 | 94.43 | 89.77 | 86.72 | 66.14 | 13.27 |
| | | Krum | 93.21 | 92.83 | 93.57 | 89.86 | 93.75 | 90.51 | 81.15 | 68.33 | 10.42 | 93.25 | 93.03 | 93.10 | 91.89 | 92.36 | 93.62 | 93.56 | 94.22 | 90.22 | 83.32 | 64.41 | 13.09 |
| | | GM | 97.28 | 96.80 | 96.48 | 97.27 | 96.90 | 95.89 | **94.04** | **87.43** | 15.11 | 96.92 | 97.18 | 96.29 | 96.87 | 97.31 | 97.15 | 97.36 | 96.66 | 92.17 | 9.58 | 9.58 | 9.58 |
| | | MCA | 97.53 | 97.37 | 97.71 | 97.13 | 96.34 | **97.65** | 11.35 | 11.35 | 10.09 | 97.19 | 13.14 | 11.35 | 9.80 | 97.88 | 94.73 | 97.38 | **97.44** | 97.19 | 13.14 | 11.35 | 9.80 |
| | | CClip | 97.45 | 97.32 | 97.48 | 97.35 | 96.57 | 97.07 | 9.82 | 10.32 | 11.35 | 97.41 | 9.80 | 9.80 | 9.80 | 97.58 | 97.45 | 97.27 | 97.37 | 97.41 | 9.80 | 9.80 | 9.80 |
| MLP | 0.6 | **Fed-NGA** | **96.72** | **96.40** | **96.07** | **95.38** | **94.98** | 93.06 | 89.79 | **86.32** | **83.66** | **96.21** | **95.90** | **95.34** | **94.71** | **96.47** | **95.99** | **95.35** | 94.92 | **96.21** | **95.90** | **94.93** | **94.71** |
| | | FedAvg | 95.32 | 15.25 | 12.30 | 15.50 | 16.70 | 10.28 | 11.35 | 11.35 | 11.35 | 10.32 | 10.28 | 11.35 | 9.82 | 91.75 | 87.77 | 82.98 | 78.43 | 10.32 | 10.28 | 11.35 | 9.82 |
| | | Median | 92.61 | 92.67 | 92.58 | 92.44 | 92.57 | 92.24 | 90.18 | 83.54 | 61.60 | 92.52 | 92.46 | 92.24 | 92.33 | 92.72 | 92.77 | 92.61 | 92.59 | 91.88 | 90.06 | 77.76 | 68.09 |
| | | Krum | 92.60 | 92.71 | 92.66 | 92.32 | 92.73 | 92.20 | 90.16 | 83.18 | 60.63 | 92.69 | 92.54 | 91.98 | 92.28 | 92.69 | 92.72 | 92.48 | 92.71 | 91.77 | 89.93 | 77.94 | 67.50 |
| | | GM | 94.67 | 94.78 | 94.76 | 94.46 | 94.46 | 93.70 | **90.87** | 86.28 | 70.11 | 94.44 | 94.65 | 93.98 | 94.31 | 94.68 | 94.69 | 94.35 | 94.33 | 91.13 | 80.95 | 11.46 | 11.12 |
| | | MCA | 95.10 | 95.00 | 95.10 | 94.86 | 94.82 | **95.15** | 13.10 | 11.35 | 11.35 | 94.64 | 91.51 | 11.91 | 11.35 | 95.13 | 95.20 | 94.83 | 94.84 | 94.64 | 91.51 | 11.91 | 11.35 |
| | | CClip | 94.99 | 95.23 | 94.98 | 94.96 | 94.87 | 95.02 | 11.35 | 11.35 | 10.09 | 94.74 | 40.64 | 9.80 | 9.80 | 95.17 | 95.00 | 94.93 | **94.93** | 94.74 | 40.64 | 9.80 | 9.80 |
| | 0.2 | **Fed-NGA** | **96.34** | **95.95** | **95.45** | **95.15** | 94.64 | 92.54 | 88.79 | **86.59** | **80.71** | **95.72** | **95.45** | **94.55** | **94.38** | **95.85** | **95.51** | **95.18** | 94.46 | **95.72** | **95.45** | **94.55** | **94.38** |
| | | FedAvg | 95.17 | 12.03 | 17.96 | 13.68 | 14.37 | 11.35 | 10.28 | 10.10 | 10.10 | 24.05 | 10.10 | 11.35 | 10.09 | 91.47 | 87.83 | 83.27 | 79.32 | 24.05 | 10.10 | 11.35 | 10.09 |
| | | Median | 91.36 | 91.44 | 90.82 | 91.21 | 91.05 | 90.92 | 88.07 | 78.97 | 47.12 | 90.95 | 90.27 | 90.86 | 89.96 | 91.50 | 90.93 | 91.50 | 91.06 | 90.89 | 88.16 | 79.49 | 51.04 |
| | | Krum | 91.31 | 91.36 | 91.00 | 91.29 | 90.95 | 90.82 | 88.17 | 78.53 | 48.28 | 91.07 | 90.26 | 90.66 | 89.82 | 91.31 | 91.16 | 91.28 | 91.33 | 90.70 | 88.48 | 80.12 | 49.72 |
| | | GM | 94.78 | 94.34 | 94.37 | 94.36 | 94.10 | 92.99 | **89.91** | 78.15 | 66.44 | 94.18 | 93.60 | 94.25 | 92.76 | 94.41 | 94.30 | 94.30 | 94.11 | 90.11 | 74.78 | 25.28 | 14.71 |
| | | MCA | 95.08 | 95.10 | 95.03 | 95.04 | **94.79** | **94.95** | 12.60 | 10.10 | 10.10 | 94.81 | 94.01 | 12.48 | 10.32 | 95.07 | 95.03 | 95.01 | **94.91** | 94.81 | 94.01 | 12.48 | 10.32 |
| | | CClip | 95.03 | 94.94 | 94.98 | 94.75 | 94.76 | 94.95 | 10.32 | 10.10 | 10.10 | 94.57 | 14.47 | 9.80 | 9.80 | 95.06 | 95.03 | 94.86 | 94.66 | 94.57 | 14.47 | 9.80 | 9.80 |

(a) Gaussian attack.

(b) Same-value attack.

(c) Sign-flip attack.

(d) LIE attack.

(e) FoE attack.

Figure 7: The maximum test accuracy (%) over 500 iterations for Fed-NGA and baselines is evaluated on the MNIST dataset and MLP model wiht $\beta = 0.2$ across five different Byzantine ratios.

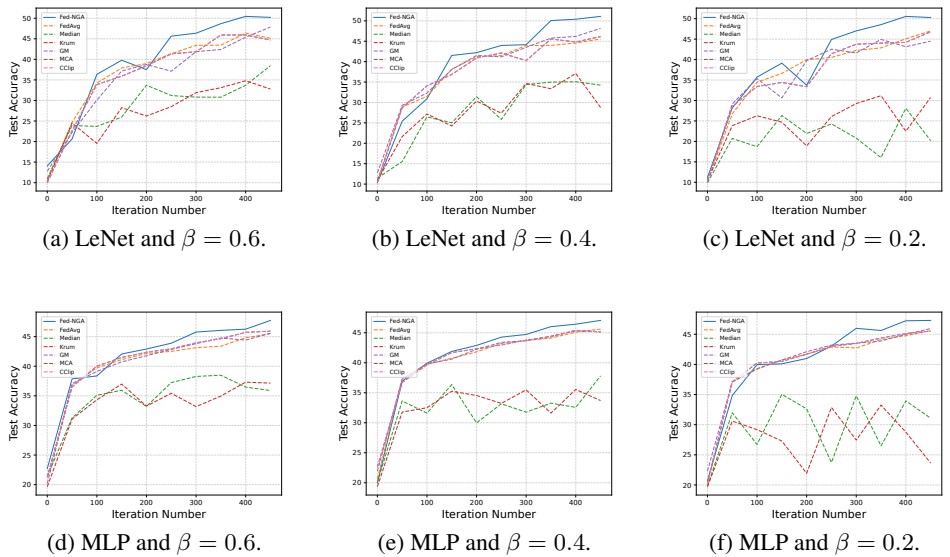

Figure 8: The test accuracy (%) over 500 iterations for Fed-NGA and baselines is evaluated on CIFAR10 dataset without Byzantine attacks.

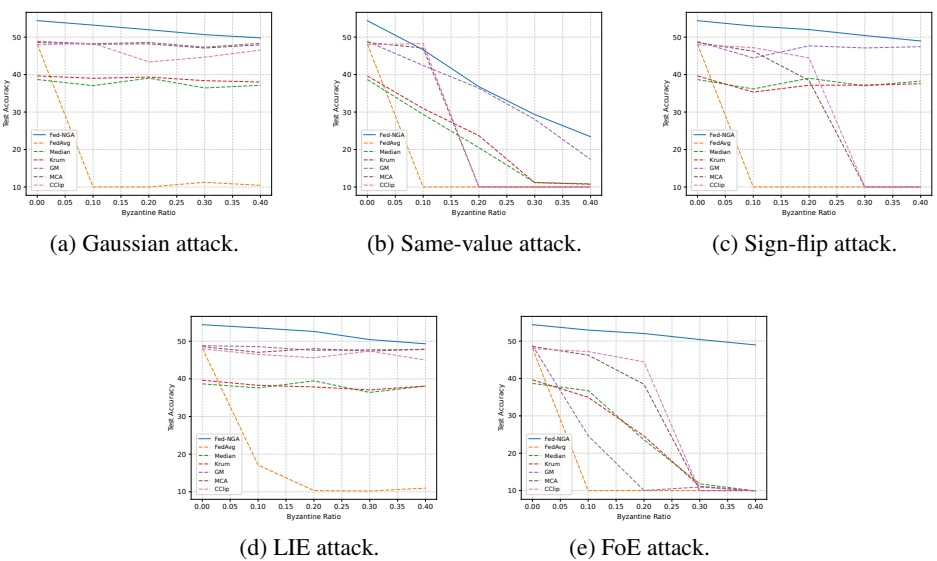

Figure 9: The maximum test accuracy (%) over 500 iterations for Fed-NGA and baselines is evaluated on the CIFAR10 dataset and LeNet model with $\beta = 0.4$ across five different Byzantine ratios.

datasets, respectively. Figure 6, Figure 8, and Figure 10 illustrate the test accuracy trends for Fed-NGA and baselines under three data heterogeneity concentration parameters and three learning models in the absence of Byzantine attacks, for the MNIST, CIFAR10, and TinyImageNet datasets, respectively. Figure 7, Figure 9, and Figure 11 present the maximum test accuracy for Fed-NGA and baselines across four Byzantine ratios, under three data heterogeneity concentration parameters, for three learning models on the MNIST, CIFAR10, and TinyImageNet datasets, respectively. In the subsequent analysis, we evaluate the performance of Fed-NGA and baseline methods according to the type of Byzantine attack.

Table 5: The maximum test accuracy (%) of 500 iterations of Fed-NGA and baselines with different types, ratios of Byzantine attack, and concentration parameter on CIFAR10 dataset and two learning models.

| Model | β | Algorithm | No Attack 0 | Gaussian Attack 0.1 | 0.2 | 0.3 | 0.4 | Same-value Attack 0.1 | 0.2 | 0.3 | 0.4 | Sign-flip Attack 0.1 | 0.2 | 0.3 | 0.4 | LIE Attack 0.1 | 0.2 | 0.3 | 0.4 | FoE Attack 0.1 | 0.2 | 0.3 | 0.4 |
|---|---|---|---|---|---|---|---|---|---|---|---|---|---|---|---|---|---|---|---|---|---|---|---|
| LeNet | 0.6 | **Fed-NGA** | 54.48 | 53.88 | 53.60 | 52.07 | 50.62 | 46.10 | **38.80** | 29.16 | 25.88 | 53.60 | 52.49 | 51.16 | 50.68 | 53.96 | 53.93 | 51.82 | 50.56 | 53.60 | 52.49 | 51.16 | 50.68 |
| | | FedAvg | 49.11 | 11.28 | 10.06 | 10.40 | 11.09 | 10.00 | 10.00 | 10.00 | 10.00 | 10.00 | 10.03 | 10.00 | 10.00 | 10.31 | 10.63 | 10.00 | 10.00 | 10.00 | 10.03 | 10.00 | 10.00 |
| | | Median | 39.98 | 40.80 | 38.55 | 39.64 | 40.15 | 31.79 | 24.21 | 10.70 | 10.82 | 39.81 | 40.19 | 38.87 | 39.98 | 40.80 | 38.80 | 39.75 | 39.67 | 38.43 | 24.26 | 14.19 | 11.81 |
| | | Krum | 40.76 | 41.40 | 40.06 | 39.55 | 39.66 | 32.41 | 22.54 | 10.11 | 10.46 | 39.34 | 39.58 | 39.67 | 40.22 | 40.96 | 40.59 | 39.96 | 40.50 | 38.99 | 25.10 | 14.02 | 11.65 |
| | | GM | 48.41 | 47.73 | 47.53 | 48.08 | 48.23 | 42.96 | 36.39 | 28.81 | 21.15 | 48.13 | 47.55 | 47.89 | 47.16 | 48.68 | 48.18 | 47.90 | 46.80 | 31.53 | 10.06 | 10.47 | 10.04 |
| | | MCA | 48.67 | 48.56 | 47.90 | 48.53 | 47.00 | **48.34** | 10.00 | 10.00 | 10.00 | 47.69 | 34.07 | 10.00 | 10.00 | 48.78 | 48.02 | 48.28 | 45.82 | 47.69 | 34.07 | 10.00 | 10.00 |
| | | CClip | 48.59 | 48.22 | 46.59 | 43.42 | 46.61 | 48.34 | 10.00 | 10.00 | 10.00 | 44.94 | 10.00 | 10.00 | 10.00 | 47.68 | 47.33 | 47.79 | 47.30 | 44.94 | 10.00 | 10.00 | 10.00 |
| | 0.4 | **Fed-NGA** | 54.39 | 53.21 | 51.96 | 50.66 | 49.79 | 46.63 | 36.76 | 29.34 | 23.42 | 52.95 | 52.02 | 50.42 | 48.97 | 53.52 | 52.60 | 50.43 | 49.31 | 52.95 | 52.02 | 50.42 | 48.97 |
| | | FedAvg | 48.14 | 10.00 | 10.00 | 11.25 | 10.43 | 10.00 | 10.00 | 10.00 | 10.00 | 10.00 | 10.00 | 10.00 | 10.00 | 17.12 | 10.34 | 10.22 | 10.98 | 10.00 | 10.00 | 10.00 | 10.00 |
| | | Median | 38.68 | 37.05 | 39.05 | 36.43 | 37.14 | 29.36 | 20.50 | 11.20 | 10.83 | 36.17 | 39.03 | 37.00 | 38.26 | 37.60 | 39.47 | 36.40 | 38.08 | 36.73 | 23.56 | 11.82 | 9.89 |
| | | Krum | 39.64 | 39.01 | 39.33 | 38.37 | 38.02 | 30.95 | 23.68 | 11.16 | 10.73 | 35.32 | 37.16 | 37.19 | 37.56 | 38.28 | 37.85 | 37.06 | 38.10 | 34.96 | 24.59 | 11.15 | 9.91 |
| | | GM | 48.84 | 48.24 | 48.58 | 47.31 | 48.35 | 42.44 | 36.36 | 28.05 | 17.37 | 44.44 | 47.67 | 47.11 | 47.45 | 48.56 | 47.56 | 47.74 | 47.76 | 24.67 | 10.03 | 10.95 | 9.92 |
| | | MCA | 48.56 | 48.10 | 48.21 | 47.09 | 47.94 | 47.09 | 10.02 | 10.00 | 10.00 | 46.25 | 38.43 | 10.00 | 10.00 | 47.06 | 48.00 | 47.37 | 47.91 | 46.25 | 38.43 | 10.00 | 10.00 |
| | | CClip | 47.97 | 48.33 | 43.37 | 44.65 | 46.53 | **48.27** | 10.00 | 10.00 | 10.00 | 47.20 | 44.44 | 10.00 | 10.00 | 46.50 | 45.61 | 47.38 | 45.01 | 47.20 | 44.44 | 10.00 | 10.00 |
| MLP | 0.6 | **Fed-NGA** | 48.26 | 47.78 | 47.82 | 46.53 | 46.19 | 30.57 | 25.93 | 20.54 | **19.13** | 47.90 | 47.26 | 46.48 | 45.72 | 48.10 | 47.40 | 46.49 | 46.10 | 47.90 | 47.26 | 46.48 | 45.72 |
| | | FedAvg | 46.12 | 13.86 | 12.86 | 10.82 | 13.01 | 10.00 | 10.00 | 10.00 | 10.00 | 14.01 | 12.14 | 10.00 | 10.00 | 28.95 | 18.12 | 14.94 | 14.43 | 14.01 | 12.14 | 10.00 | 10.00 |
| | | Median | 40.01 | 39.60 | 39.83 | 39.81 | 38.50 | 32.41 | **27.94** | 19.60 | 17.57 | 39.68 | 39.84 | 39.72 | 38.20 | 39.83 | 40.20 | 40.16 | 39.35 | 38.71 | 34.61 | 21.54 | 17.38 |
| | | Krum | 40.34 | 39.92 | 39.94 | 39.86 | 39.06 | 32.95 | 27.27 | 19.32 | 17.61 | 39.81 | 39.65 | 39.80 | 38.60 | 40.35 | 40.24 | 40.62 | 38.88 | 38.37 | 34.40 | 21.94 | 17.04 |
| | | GM | 46.04 | 45.60 | 46.03 | 45.02 | 45.51 | 31.13 | 23.53 | **20.97** | 15.56 | 45.43 | 46.19 | 44.69 | 44.71 | 46.00 | 46.06 | 44.99 | 44.84 | 36.26 | 18.22 | 9.53 | 10.82 |
| | | MCA | 46.61 | 45.86 | 46.23 | 45.50 | 44.63 | **46.21** | 12.69 | 10.00 | 10.00 | 45.52 | 20.35 | 13.91 | 10.00 | 46.40 | 46.29 | 44.96 | 44.90 | 45.52 | 20.35 | 13.91 | 10.00 |
| | | CClip | 45.89 | 45.41 | 45.81 | 45.50 | 45.37 | 45.69 | 10.00 | 10.00 | 10.00 | 45.17 | 10.00 | 10.00 | 10.00 | 45.56 | 45.36 | 45.44 | 44.52 | 45.17 | 10.00 | 10.00 | 10.00 |
| | 0.4 | **Fed-NGA** | 48.39 | 48.27 | 47.20 | 46.37 | 46.21 | 30.27 | 25.13 | 18.81 | 18.27 | 47.57 | 47.52 | 46.26 | 45.55 | 48.00 | 47.41 | 46.24 | 46.17 | 47.57 | 47.52 | 46.26 | 45.55 |
| | | FedAvg | 46.44 | 11.94 | 13.67 | 13.43 | 12.43 | 10.00 | 10.00 | 10.00 | 10.00 | 12.91 | 10.00 | 11.76 | 10.08 | 29.15 | 18.48 | 12.80 | 14.90 | 12.91 | 10.00 | 11.76 | 10.08 |
| | | Median | 39.15 | 38.42 | 39.47 | 38.72 | 39.95 | 33.17 | **26.63** | 17.51 | 16.25 | 37.83 | 38.16 | 38.36 | 39.32 | 38.95 | 39.54 | 38.76 | 39.76 | 37.53 | 34.95 | 22.89 | 12.55 |
| | | Krum | 39.14 | 38.79 | 39.41 | 39.23 | 39.83 | 33.06 | 26.15 | 17.60 | 16.72 | 38.42 | 38.41 | 38.39 | 39.75 | 38.92 | 40.01 | 39.67 | 39.10 | 37.89 | 35.14 | 22.75 | 12.60 |
| | | GM | 46.72 | 45.92 | 45.79 | 45.71 | 45.09 | 30.11 | 22.72 | 18.55 | 15.48 | 45.58 | 46.03 | 45.13 | 44.85 | 45.91 | 45.79 | 45.58 | 45.30 | 35.71 | 15.72 | 14.46 | 10.00 |
| | | MCA | 46.41 | 46.10 | 46.08 | 45.78 | 45.53 | **46.08** | 10.21 | 10.00 | 10.00 | 45.44 | 39.81 | 13.03 | 10.00 | 45.85 | 45.99 | 45.79 | 45.11 | 45.44 | 39.81 | 13.03 | 10.00 |
| | | CClip | 46.18 | 46.48 | 45.28 | 45.03 | 45.32 | 45.89 | 10.00 | 10.00 | 10.00 | 44.10 | 10.00 | 10.00 | 10.00 | 45.07 | 45.60 | 45.86 | 44.68 | 44.10 | 10.00 | 10.00 | 10.00 |

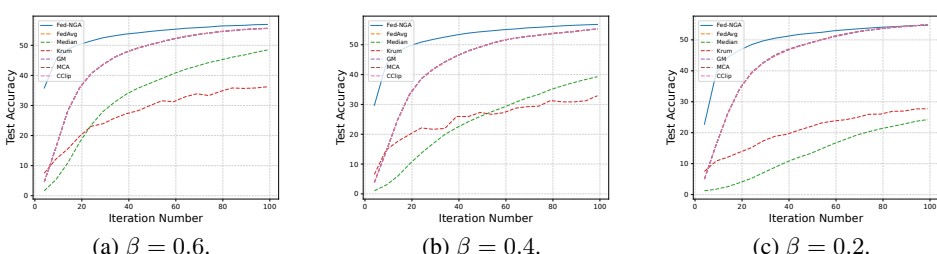

(a) $\beta = 0.6$.     (b) $\beta = 0.4$.     (c) $\beta = 0.2$.

Figure 10: The test accuracy (%) over 100 iterations for Fed-NGA and baselines is evaluated on TinyImageNet dataset and MobileNetV3 model without Byzantine attacks.

**No Attack:** First, for the simpler dataset (MNIST), Table 4 and Figure 6 demonstrate that Fed-NGA consistently outperforms all other baselines across different data heterogeneity concentration parameter configurations and two learning models in the absence of Byzantine attacks. Specifically, compared to the baselines, Fed-NGA improves test accuracy for the LeNet model by 0.81%, and 0.74% for data heterogeneity concentration parameters $\beta = 0.6$, and 0.2, respectively. Similarly, for the MLP model, Fed-NGA achieves test accuracy gains of 1.4%, and 1.17% for data heterogeneity concentration parameters $\beta = 0.6$, and 0.2, respectively. Moreover, as shown in Figure 6, the convergence speed of Fed-NGA is comparable to that of other baselines and exhibits superior performance under different data heterogeneity conditions. In contrast, the Median and Krum methods demonstrate poor performance irrespective of the learning model used. For the more complex CIFAR10 dataset, Table 5 and Figure 8 also highlight the advantages of Fed-NGA over

Table 6: The maximum test accuracy (%) of 100 iterations of Fed-NGA and baselines with different types, ratios of Byzantine attack, and concentration parameter on TinyImageNet dataset and MobileNetV3 model.

| Attack Name | | | No Attack | Gaussian Attack | | | | Same-value Attack | | | | Sign-flip Attack | | | | LIE Attack | | | | FoE Attack | | | |
|---|---|---|---|---|---|---|---|---|---|---|---|---|---|---|---|---|---|---|---|---|---|---|---|
| Model | β | $\bar{C}_\alpha$ / Algorithm | 0 | 0.1 | 0.2 | 0.3 | 0.4 | 0.1 | 0.2 | 0.3 | 0.4 | 0.1 | 0.2 | 0.3 | 0.4 | 0.1 | 0.2 | 0.3 | 0.4 | 0.1 | 0.2 | 0.3 | 0.4 |
| MobileNetV3 | 0.6 | **Fed-NGA** | **56.95** | **56.54** | **55.77** | **54.80** | 53.12 | **55.40** | **49.31** | 22.68 | 2.20 | 52.65 | **45.38** | 13.44 | 0.50 | **56.29** | **55.82** | **54.95** | **53.87** | 52.19 | **45.23** | 12.92 | 0.01 |
| | | FedAvg | 55.67 | 49.75 | 46.27 | 40.12 | 34.26 | 0.72 | 0.54 | 0.51 | 0.50 | 0.50 | 0.50 | 0.50 | 0.50 | 51.37 | 48.16 | 45.38 | 42.63 | 0.50 | 0.50 | 0.50 | 0.50 |
| | | Median | 48.45 | 46.90 | 45.87 | 44.90 | 45.82 | 42.21 | 41.14 | **46.03** | 50.36 | 36.19 | 21.41 | 4.34 | 9.39 | 46.06 | 46.67 | 45.23 | 46.08 | 43.01 | 38.60 | **22.60** | 3.22 |
| | | Krum | 36.22 | 35.89 | 36.76 | 34.04 | 32.85 | 36.62 | 30.68 | 33.03 | 31.62 | 34.25 | 36.31 | 36.65 | **31.63** | 37.18 | 36.09 | 36.30 | 33.75 | 0.51 | 0.50 | 0.49 | 0.47 |
| | | GM | 55.68 | 54.47 | 54.22 | 53.82 | 53.50 | 52.38 | 48.98 | 27.57 | 1.39 | 50.71 | 43.14 | 4.87 | 0.02 | 54.56 | 54.32 | 53.69 | 53.55 | 50.12 | 42.88 | 4.62 | 0.04 |
| | | MCA | 55.62 | 54.64 | 54.22 | 53.82 | **53.66** | 54.88 | 1.70 | 0.53 | 0.51 | **54.50** | 0.50 | 0.50 | 0.50 | 54.69 | 54.34 | 53.72 | 53.72 | **54.50** | 0.50 | 0.50 | 0.50 |
| | | CClip | 50.16 | 47.51 | 45.99 | 42.30 | 40.00 | 45.22 | 43.51 | 35.94 | 18.31 | 44.91 | 38.38 | 1.98 | 0.01 | 47.85 | 45.48 | 43.56 | 39.16 | 44.26 | 37.25 | 1.66 | 0.19 |
| | 0.4 | **Fed-NGA** | **56.82** | **56.00** | **54.85** | **53.74** | 51.95 | **54.64** | 47.12 | 10.39 | 1.69 | 51.17 | **42.45** | 0.11 | 0.01 | **56.04** | **54.79** | **54.25** | 51.97 | 51.60 | **43.54** | 0.10 | 0.02 |
| | | FedAvg | 55.29 | 50.03 | 45.94 | 39.57 | 33.83 | 0.76 | 0.60 | 0.53 | 0.51 | 0.50 | 0.50 | 0.50 | 0.50 | | | | | 0.50 | 0.50 | 0.50 | 0.50 |
| | | Median | 39.30 | 39.28 | 37.69 | 39.33 | 39.42 | 34.77 | 40.00 | **45.66** | **48.38** | 25.38 | 10.78 | 3.64 | 10.76 | 38.54 | 40.44 | 40.51 | 38.67 | 37.20 | 30.19 | **14.93** | **2.68** |
| | | Krum | 32.92 | 26.82 | 32.12 | 28.97 | 31.43 | 35.87 | 30.39 | 29.99 | 31.15 | 31.31 | 32.69 | **34.66** | **29.97** | 32.47 | 30.77 | 32.60 | 33.36 | 0.46 | 0.46 | 0.46 | 0.45 |
| | | GM | 55.32 | 54.96 | 54.01 | 53.65 | 53.14 | 52.41 | **47.61** | 8.48 | 0.61 | 50.09 | 40.67 | 0.17 | 0.01 | 54.94 | 54.59 | 54.19 | 53.61 | 50.49 | 39.61 | 0.11 | 0.09 |
| | | MCA | 55.32 | 54.98 | 53.99 | 53.73 | **53.35** | 54.60 | 0.90 | 0.57 | 0.51 | **55.05** | 0.50 | 0.50 | 0.50 | 54.93 | 54.75 | 54.19 | **53.75** | **55.05** | 0.50 | 0.50 | 0.50 |
| | | CClip | 48.60 | 46.92 | 43.87 | 40.81 | 35.35 | 45.89 | 43.17 | 34.73 | 15.86 | 43.73 | 31.14 | 0.22 | 0.02 | 46.25 | 45.69 | 41.83 | 39.41 | 43.74 | 29.97 | 0.25 | 0.11 |

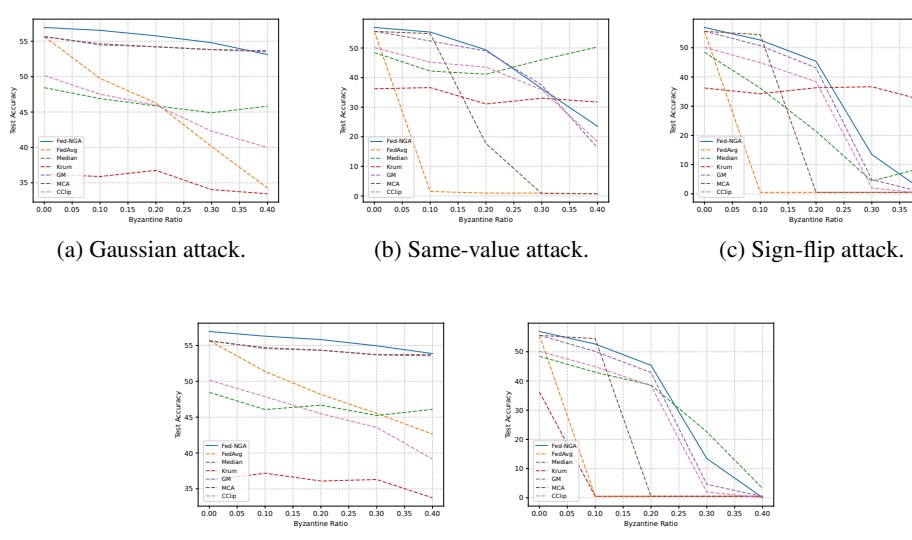

(a) Gaussian attack.   (b) Same-value attack.   (c) Sign-flip attack.

(d) LIE attack.   (e) FoE attack.

Figure 11: The maximum test accuracy (%) over 100 iterations for Fed-NGA and baselines is evaluated on the TinyImageNet dataset and MobileNetV3 model with $\beta = 0.6$ across five different Byzantine ratios.

other baselines across different data heterogeneity concentration parameter configurations and two learning models without Byzantine attacks. Compared to the MNIST dataset, Fed-NGA achieves even more pronounced improvements on the CIFAR10 dataset, improving test accuracy for the LeNet model by 5.37%, and 5.55% for data heterogeneity concentration parameters $\beta = 0.6$, and 0.4, respectively. For the MLP model, Fed-NGA achieves test accuracy gains of 1.65%, and 1.67% for data heterogeneity concentration parameters $\beta = 0.6$, and 0.4, respectively. Furthermore, Figure 8 indicates that Fed-NGA demonstrates a convergence speed comparable to other baselines for both learning models, while maintaining greater stability in the presence of high data heterogeneity. Finally for TinyImageNet dataset, Table 6 and Figure 10 show Fed-NGA outperformed baselines across different data heterogeneity concentration parameter configurations on MobileNetV3 model without Byzantine attack. Fed-NGA improves the test accuracy by 1.27% and 1.5% for $\beta = 0.6$, and 0.4, respectively. Also, as shown in Figure 10, we can easily see the advantages of Fed-NGA compared with baselines.

Finally, on the TinyImageNet dataset, Table 6 and Figure 10 demonstrate Fed-NGA's consistent superiority over baseline methods under varying data heterogeneity settings ($\beta$ values) when implemented on MobileNetV3 models in non-Byzantine scenarios. Specifically, Fed-NGA yields test accuracy improvements of 1.27% and 1.5% for concentration parameters $\beta = 0.6$ and $0.4$, respectively. Furthermore, the visual comparison in Figure 10 clearly reveals Fed-NGA's accelerated convergence rate and enhanced stability throughout the training process compared to conventional approaches.

**Gaussian Attack:** First, we assess the training performance of Fed - NGA and the baselines on the MNIST dataset. As presented in Table 4, among all the baselines, Fed - NGA attains the highest test accuracy in most instances (with the exception of $\beta = 0.2$ on the MLP model) across four different Byzantine ratios.For the LeNet model, when considering the data heterogeneity concentration parameters, Fed-NGA enhances the test accuracy by 0% to 1.27% for $\beta = 0.6$ and by 0.04% to 0.79% for $\beta = 0.2$ across the four different Byzantine ratios. Similarly, on the MLP model, Fed-NGA boosts the test accuracy by 0.11% to 1.4% for $\beta = 0.6$ and by - 0.15% to 1.17% for $\beta = 0.2$. Furthermore, as depicted in Figure 7, it is evident that Fed-NGA is robust against all types of Byzantine attacks and maintains excellent performance across the four different Byzantine ratios. Turning to the CIFAR10 dataset, Table 5 shows that Fed-NGA outperforms all baselines, achieving the highest test accuracy. Specifically, Fed-NGA improves test accuracy on the LeNet model by 2.39% to 5.32%, and 1.44% to 4.88% across four different Byzantine ratios for data heterogeneity concentration parameters $\beta = 0.6$, and $0.4$, respectively. Similarly, for the MLP model, Fed-NGA achieves improvements of 0.68% to 1.92%, and 0.59% to 1.79% across four different Byzantine ratios for data heterogeneity concentration parameters $\beta = 0.6$, and $0.4$, respectively. Moreover, Figure 9 verifies Fed-NGA's sustained robustness to Gaussian attacks, maintaining the highest accuracy across four Byzantine ratios. For the TinyImageNet dataset, Table 6 demonstrates Fed-NGA's statistically significant superiority over baseline methods across data heterogeneity settings $\bar{C}_\alpha = 0.1$, $0.2$, and $0.3$, achieving peak test accuracy while showing minor limitations at $\bar{C}_\alpha = 0.4$. When implemented on the MobileNetV3 model under Byzantine attack scenarios, Fed-NGA achieves accuracy gains ranging from -0.54% to 1.9% and -1.4% to 1.5% across four Byzantine client ratios, corresponding to data heterogeneity parameters $\beta = 0.6$ and $0.4$ respectively. The complementary analysis in Figure 11 further confirms Fed-NGA's consistent stability and accuracy advantages of experimental configurations.

**Same-value Attack:** Referencing the Same-value attack, the performance of Fed-NGA is not always the best among all baselines, but it is undoubtedly the most stable. For the MNIST dataset, as shown in Table 4, Fed-NGA improves test accuracy by 2.48% to 64.19% under high Byzantine ratios ($\bar{C}_\alpha = 0.4$) on the LeNet model. A similar phenomenon is observed for the MLP model with $\bar{C}_\alpha = 0.3$ and $0.4$, where the accuracy improvement ranges from 0.04% to 14.27%. From Figure 7, all robust methods experience performance degradation as the Byzantine ratio increases. Even under high data heterogeneity, where other baselines may fail to converge, Fed-NGA demonstrates the least performance degradation, highlighting its stability in the face of high data heterogeneity and Byzantine ratios. On the CIFAR10 dataset, Fed-NGA exhibits a consistent performance trend with the MNIST results, achieving test accuracy improvements of 0.24%–11.63% under high Byzantine ratios ($\bar{C}_\alpha = 0.3, 0.4$). As shown in Figure 9, Fed-NGA maintains accuracy advantages across most attack scenarios while remaining marginally below baselines at $\bar{C}_\alpha = 0.1$. However, the TinyImageNet dataset exhibits distinct behavioral patterns compared to the other two benchmark datasets. Intriguingly, we observe an unexpected positive correlation between Byzantine client ratios and test accuracy in the Median method, which the set of Same-value attack is not reasonable. Notably, Fed-NGA demonstrates measurable improvements of 0.53% and 0.04% test accuracy at $\bar{C}_\alpha = 0.1$, while maintaining competitive performance at high Byzantine ratio – surpassing all baseline methods except the Median and Krum approaches. These findings are corroborated by the comparative analysis in Figure 11, which further validates Fed-NGA's robustness across heterogeneous experimental conditions.

**Sign-flip Attack:** For the MNIST dataset, Table 4 illustrates that Fed-NGA achieves higher test accuracy than all baselines in most cases—with the exception of $\beta = 0.2$ and $\bar{C}_\alpha = 0.4$ on the LeNet model—across four Byzantine ratios. Specifically, under the data heterogeneity parameter $\beta = 0.6$, Fed-NGA improves test accuracy on the LeNet model by 0.24% to 1.34% across all Byzantine ratios. When $\beta = 0.2$, a minor performance gap of approximately 0.06% arises only at $\bar{C}_\alpha = 0.4$; in all other scenarios, accuracy improvements reach up to 1.14%, highlighting its robustness to varying data distributions. On the MLP model, Fed-NGA yields 0.4% to 1.47% accuracy

gains for $\beta = 0.6$ and 0.3% to 1.62% for $\beta = 0.2$ across the four Byzantine ratios, consistently outperforming baseline methods. Moreover, Figure 7 visually demonstrates the superiority of Fed-NGA: the framework maintains a notable accuracy lead over baselines, showcasing resilience across all evaluated Byzantine attack scenarios. For the CIFAR10 dataset, using the LeNet model, Fed-NGA achieves performance improvements of 3.27% to 5.47%, and 1.52% to 5.75% across four Byzantine ratios for data heterogeneity concentration parameters $\beta = 0.6$, and 0.4, respectively. For the MLP model, Fed-NGA achieves similar improvements, ranging from 1.01% to 2.38%, and 0.7% to 1.99%, across the same conditions. Moreover, as shown in Figure 9, Fed-NGA consistently outperforms under the Sign-flip attack regardless of the Byzantine ratio. Only the GM method demonstrates comparable performance under scenarios with high Byzantine ratios and significant data heterogeneity. Turning to the TinyImageNet dataset, Fed-NGA demonstrates significant superiority at $\bar{C}_\alpha = 0.2$, achieving test accuracy improvements of 2.24% and 1.78% under data heterogeneity parameters $\beta = 0.6$ and $\beta = 0.4$ respectively. Notably, we observe an inverse performance pattern under high Byzantine ratios where all comparative methods underperform Krum - a divergence from results observed in smaller-scale datasets. This phenomenon may stem from MobileNetV3's enhanced architectural complexity and expanded parameter space (relative to baseline models), which potentially aligns better with Krum's robust aggregation paradigm. These findings are validated through the comparative analysis in Figure 11.

**LIE Attack:** We first evaluate the training performance of Fed-NGA and baselines on the MNIST dataset. As shown in Table 4, Fed-NGA achieves the highest test accuracy among all baselines except for the case of $\bar{C}_\alpha = 0.4$. For the LeNet model, under data heterogeneity parameters $\beta = 0.6$ and $\beta = 0.2$, test accuracy changes range from a slight decrease of 0.03% to an increase of 0.36%, and from a 0.22% decrease to a 0.44% increase, respectively. For the MLP model, corresponding accuracy variations span from a 0.01% decrease to a 0.79% increase for $\beta = 0.6$, and from a 0.45% decrease to a 0.78% increase for $\beta = 0.2$, reflecting Fed-NGA's competitive performance across most configurations. Additionally, Figure 7 demonstrates that Fed-NGA maintains stable test accuracy when facing attacks of different proportions, outperforming baselines in most scenarios. This consistency highlights the framework's ability to withstand both data heterogeneity and Byzantine attacks during training. For the CIFAR10 dataset, Fed-NGA consistently outperforms all baselines under all experimental setups. Test accuracy increases by 3.26% to 5.75%, and 1.4% to 4.96% for the LeNet model, and by 1.11% to 1.7%, and 0.38% to 2.09% for the MLP model, across four Byzantine ratios for data heterogeneity concentration parameters $\beta = 0.6$, and 0.4, respectively. As further evidenced by Figure 9, Fed-NGA maintains robust performance against LIE attack, achieving the highest accuracy among comparative baselines. Finally, we evaluate Fed-NGA's performance on the TinyImageNet dataset under varying Byzantine attack scenarios. While Fed-NGA does not universally dominate across all four Byzantine ratios and two data heterogeneity parameters ($\beta = 0.6, 0.4$), it demonstrates significant superiority in most experimental configurations. The framework achieves marginal improvements ranging from 0.04% to 1.6% in test accuracy across these settings, with only a single outlier instance showing a 1.78% degradation at $\bar{C}_\alpha = 0.4$ combined with $\beta = 0.4$. Crucially, as evidenced in Figure 11, Fed-NGA exhibits remarkable resilience against LIE attack, outperforming all baseline methods by maintaining superior decision boundaries in high-dimensional feature spaces.

**FoE Attack:** Against the newly proposed Byzantine attack, the FoE attack, Fed-NGA exhibits superior defense performance compared to all baseline methods, regardless of the dataset, learning model, or data heterogeneity level. For the MNIST dataset, as shown in Table 4, Fed-NGA improves test accuracy by 0.56% to 85.57%, and 0.67% to 83.54% for data heterogeneity concentration parameters $\beta = 0.6$, and 0.2, respectively, across four different Byzantine ratios. For the MLP model, Fed-NGA also enhances test accuracy by 1.67% to 26.62%, and 0.91% to 43.34% for data heterogeneity concentration parameters $\beta = 0.6$, and 0.2, respectively. Figure 6 conclusively demonstrates that Fed-NGA maintains the highest test accuracy across all four Byzantine client ratios under FoE attacks. Similarly, on the CIFAR10 dataset, Fed-NGA achieves test accuracy improvements of 5.91% to 38.87%, and 5.75% to 38.97% for data heterogeneity concentration parameters $\beta = 0.6$, and 0.4, respectively, across four different Byzantine ratios. For the MLP model, Fed-NGA provides additional improvements of 2.38% to 28.34%, and 2.13% to 32.95% for data heterogeneity concentration parameters $\beta = 0.6$, and 0.4, respectively. Moreover, Figure 9 demonstrates that although the training performance of Fed-NGA decreases as the Byzantine ratio increases, its convergence remains robust and intact. In contrast, the baselines fail to converge under high Byzantine attack rates underscoring the robustness and superiority of Fed-NGA. On

Table 7: The running time (in seconds) for 500 iterations of Fed-NGA and robust baselines is evaluated under varying types and ratios of Byzantine attacks, as well as different concentration parameters, on the MNIST dataset using two learning models. Notably, only the running time of the aggregation algorithms is considered, excluding the model training time.

| Model | $\beta$ | $\bar{C}_\alpha$ / Algorithm | No Attack 0 | Gaussian Attack 0.1 | 0.2 | 0.3 | 0.4 | Same-value Attack 0.1 | 0.2 | 0.3 | 0.4 | Sign-flip Attack 0.1 | 0.2 | 0.3 | 0.4 | LIE Attack 0.1 | 0.2 | 0.3 | 0.4 | FoE Attack 0.1 | 0.2 | 0.3 | 0.4 |
|---|---|---|---|---|---|---|---|---|---|---|---|---|---|---|---|---|---|---|---|---|---|---|---|
| LeNet | 0.6 | Fed-NGA | 0.0495 | 0.0494 | 0.0509 | 0.0499 | 0.0479 | 0.0441 | 0.0435 | 0.0432 | 0.0438 | 0.0424 | 0.0417 | 0.0529 | 0.0427 | 0.0417 | 0.0413 | 0.0407 | 0.0488 | 0.0424 | 0.0417 | 0.0529 | 0.0427 |
| | | Median | 1.0961 | 1.4079 | 1.4794 | 1.4009 | 1.4417 | 1.3547 | 1.4668 | 1.5285 | 1.5046 | 1.4461 | 1.4202 | 1.4232 | 1.4633 | 1.3550 | 1.5062 | 1.4550 | 1.4660 | 1.3165 | 1.4099 | 0.0529 | 1.3912 |
| | | Krum | 1.0428 | 1.3709 | 1.3887 | 1.3568 | 1.4820 | 1.2106 | 1.5283 | 1.5549 | 1.5008 | 1.7902 | 1.5460 | 1.4541 | 1.4158 | 1.4033 | 1.4109 | 1.4756 | 1.4118 | 1.3667 | 1.4338 | 1.3354 | 1.4015 |
| | | GM | 1.4404 | 2.6604 | 2.7069 | 3.0213 | 3.2632 | 2.4557 | 3.0724 | 4.3619 | 7.8069 | 6.7954 | 8.5500 | 8.3508 | 7.3220 | 2.5941 | 2.8214 | 2.8121 | 3.0466 | 5.6698 | 7.9838 | 3.2985 | 5.3879 |
| | | MCA | 1.3869 | 1.5871 | 1.5197 | 1.6443 | 1.7110 | 1.6746 | 3.2483 | 4.6444 | 4.1924 | 4.0908 | 573.71 | 771.85 | 866.42 | 1.5965 | 1.6401 | 1.6593 | 1.6177 | 4.0908 | 573.71 | 771.85 | 866.42 |
| | | CClip | 1.4687 | 1.3463 | 1.3091 | 1.2905 | 1.2207 | 1.3100 | 2.8976 | 3.9070 | 3.6089 | 2.1143 | 4.3782 | 609.35 | 645.87 | 2.4963 | 2.3101 | 1.1877 | 1.2209 | 2.1143 | 4.3782 | 609.35 | 645.87 |
| | 0.2 | Fed-NGA | 0.0491 | 0.0471 | 0.0474 | 0.0460 | 0.0527 | 0.0442 | 0.0431 | 0.0467 | 0.0437 | 0.0419 | 0.0411 | 0.0444 | 0.0497 | 0.0424 | 0.0405 | 0.0408 | 0.0501 | 0.0419 | 0.0411 | 0.0444 | 0.0497 |
| | | Median | 1.1376 | 1.0380 | 1.0200 | 1.0539 | 1.0395 | 1.0114 | 1.0270 | 1.0274 | 1.0381 | 1.2884 | 1.4038 | 1.0945 | 1.2521 | 1.3964 | 1.4313 | 1.4007 | 1.4090 | 1.4558 | 1.4054 | 1.4209 | 1.4096 |
| | | Krum | 1.0990 | 1.0344 | 1.0261 | 1.0314 | 1.0147 | 1.0248 | 1.0394 | 1.0340 | 1.0447 | 1.3822 | 1.4826 | 1.4203 | 1.3488 | 1.4650 | 1.3890 | 1.4101 | 1.4067 | 1.5175 | 1.4327 | 1.3869 | 1.3980 |
| | | GM | 1.4693 | 1.5985 | 1.6613 | 1.8224 | 1.8735 | 1.5991 | 1.7919 | 2.3361 | 4.6157 | 3.0367 | 3.7362 | 3.7513 | 4.1834 | 2.8287 | 2.9844 | 3.1595 | 3.2337 | 6.1995 | 5.5147 | 5.2773 | 7.1418 |
| | | MCA | 1.2041 | 0.8844 | 0.8396 | 0.8614 | 0.8996 | 0.8721 | 1.9088 | 2.9640 | 2.6172 | 2.4331 | 294.17 | 292.48 | 300.33 | 1.6248 | 1.6204 | 0.8395 | 0.7814 | 2.4331 | 294.17 | 292.48 | 300.33 |
| | | CClip | 1.7319 | 1.3196 | 1.2437 | 1.2700 | 1.2825 | 1.2185 | 2.8768 | 3.9593 | 3.3750 | 1.7452 | 220.82 | 546.44 | 740.75 | 2.5752 | 1.4126 | 1.3398 | 1.2669 | 1.7452 | 220.82 | 546.44 | 740.75 |
| MLP | 0.6 | Fed-NGA | 0.0484 | 0.0474 | 0.0481 | 0.0472 | 0.0497 | 0.0447 | 0.0446 | 0.0438 | 0.0443 | 0.0421 | 0.0428 | 0.0527 | 0.0421 | 0.0413 | 0.0414 | 0.0409 | 0.0408 | 0.0421 | 0.0428 | 0.0527 | 0.0421 |
| | | Median | 4.5015 | 5.0526 | 5.1602 | 5.1421 | 5.2404 | 5.1285 | 5.2423 | 5.1293 | 5.4980 | 5.3448 | 5.4025 | 5.1455 | 5.3578 | 5.1993 | 5.5216 | 5.2909 | 5.1514 | 4.5143 | 4.6986 | 4.5896 | 4.6999 |
| | | Krum | 4.4100 | 5.1084 | 5.1810 | 5.2744 | 5.4015 | 8.8255 | 9.1504 | 7.5758 | 7.7471 | 5.0697 | 5.2123 | 5.0784 | 5.2540 | 4.7379 | 4.8549 | 4.8822 | 4.9461 | 4.5738 | 4.6408 | 4.5050 | 4.4994 |
| | | GM | 1.6423 | 2.8772 | 3.0643 | 3.3867 | 3.5852 | 7.6170 | 9.9575 | 13.2752 | 27.2812 | 3.1658 | 4.1455 | 3.6401 | 4.0717 | 2.1465 | 2.2603 | 2.4946 | 2.7088 | 7.8570 | 9.8331 | 15.6687 | 13.8719 |
| | | MCA | 1.2252 | 1.1696 | 1.1852 | 1.4670 | 1.4916 | 2.4586 | 5.9877 | 13.0282 | 12.4749 | 3.1312 | 3.7039 | 314.31 | 324.53 | 1.2502 | 1.1470 | 1.2437 | 1.2387 | 3.1312 | 3.7039 | 314.31 | 324.53 |
| | | CClip | 1.5097 | 1.1214 | 1.0005 | 1.2100 | 1.2726 | 1.2982 | 2.9847 | 4.3713 | 4.7536 | 4.1957 | 5.5971 | 775.70 | 794.37 | 1.2090 | 1.1351 | 1.2477 | 1.2541 | 4.1957 | 5.5971 | 775.70 | 794.37 |
| | 0.2 | Fed-NGA | 0.0479 | 0.0496 | 0.0510 | 0.0525 | 0.0541 | 0.0445 | 0.0438 | 0.0453 | 0.0448 | 0.0428 | 0.0507 | 0.0418 | 0.0503 | 0.0423 | 0.0411 | 0.0510 | 0.0502 | 0.0428 | 0.0507 | 0.0418 | 0.0503 |
| | | Median | 5.6327 | 8.4563 | 8.9068 | 8.3310 | 9.4235 | 6.6534 | 6.4367 | 6.6297 | 6.8857 | 6.6302 | 6.5910 | 6.9369 | 7.0503 | 4.5795 | 4.5821 | 4.5783 | 4.5563 | 4.5556 | 4.6016 | 4.5392 | 4.5346 |
| | | Krum | 5.4715 | 6.6928 | 6.4730 | 6.7230 | 6.7225 | 6.5784 | 6.6319 | 6.7175 | 6.7881 | 6.7623 | 6.5456 | 6.6883 | 6.9612 | 4.5270 | 4.5950 | 4.5649 | 4.6092 | 4.6184 | 4.5467 | 4.6258 | 4.6109 |
| | | GM | 2.4400 | 3.8002 | 4.3986 | 4.7389 | 4.9582 | 3.8191 | 4.9185 | 6.8835 | 12.6822 | 3.1201 | 4.4071 | 5.1005 | 5.9135 | 2.2716 | 2.4778 | 2.6678 | 2.8905 | 7.4475 | 11.0187 | 14.7150 | 8.7599 |
| | | MCA | 1.5746 | 1.6207 | 1.7081 | 1.1802 | 1.0556 | 1.9305 | 4.5181 | 6.6776 | 6.9574 | 2.3722 | 3.1110 | 306.37 | 342.84 | 1.2306 | 1.2382 | 1.3130 | 1.1409 | 2.3722 | 3.1110 | 306.37 | 342.84 |
| | | CClip | 1.5441 | 1.0945 | 0.9906 | 1.2122 | 1.2319 | 1.3254 | 2.8811 | 4.8414 | 5.4114 | 3.8493 | 5.9270 | 752.94 | 692.51 | 1.2173 | 1.1969 | 1.2293 | 1.1951 | 3.8493 | 5.9270 | 752.94 | 692.51 |

the TinyImageNet dataset, Fed-NGA demonstrates consistent superiority over all baseline methods at $\bar{C}_\alpha = 0.2$, achieving statistically significant accuracy gains of 2.35% and 3.93% under data heterogeneity parameters $\beta = 0.6$ and $\beta = 0.4$ respectively. Notably, we observe a paradigm shift in high Byzantine scenarios where all methodologies exhibit degraded performance - a marked contrast to patterns seen in smaller-scale datasets, as validated in Figure 11. Crucially, under low Byzantine ratios (0.1 and 0.2), Fed-NGA maintains competitive advantages across evaluation metrics.

## C.2 Aggregation Running Time Analysis

In this section, we analyze the running time under different Byzantine attacks, referencing Table 7, Table 8, Table 9, Figure 12, Figure 13 and Figure 14. Table 7, Table 8, and Table 9 provide the running time for Fed-NGA and robust baselines under four Byzantine ratios, three data heterogeneity concentration parameter setups, five types of Byzantine attacks, and three learning models, applied to the MNIST, CIFAR10 and TinyImageNet datasets, respectively. Figure 12, Figure 13, and Figure 14 depict the running time for Fed-NGA and baselines under three Byzantine ratios and three learning models, for the MNIST dataset with $\beta = 0.2$, the CIFAR10 dataset with $\beta = 0.4$, and the TinyImageNet dataset with $\beta = 0.6$, respectively. In the following analysis, we assess the performance of Fed-NGA and baselines in relation to the type of dataset.

**MNIST Dataset:** From Table 7, it is evident that Fed-NGA outperforms the baselines in terms of aggregated running time. In the absence of Byzantine attacks, the aggregated running time of some baseline methods is up to 20 times, and in some cases over 100 times, that of Fed-NGA. This demonstrates that the proposed algorithm significantly reduces computational complexity. Under Byzantine attacks, the aggregated running time of baseline methods increases markedly with higher Byzantine ratios, whereas Fed-NGA remains unaffected due to its straightforward normalization of uploaded vectors. Specifically, the aggregated running time of baselines is at least 20 times, and in some instances up to 200 times, the running time of our algorithm Fed-NGA. Furthermore, as shown in Figure 12, while Fed-NGA requires only 2 to 2.5 times the aggregation cost of FedAvg, it achieves robustness against Byzantine attacks with a time cost that is merely 1/20 of other robust algorithms. This highlights the fast and efficient nature of Fed-NGA, aligning with our theoretical proofs and underscoring its effectiveness in experimental settings.

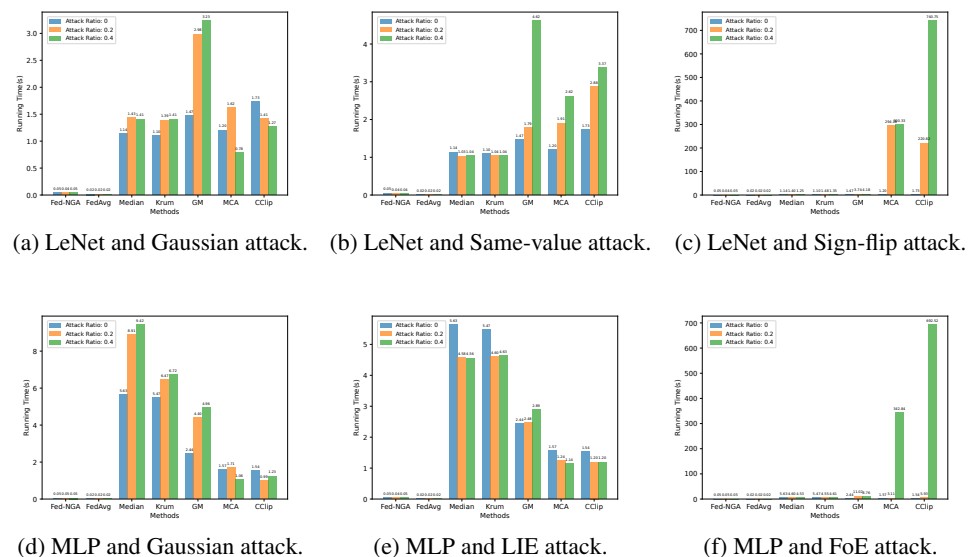

(a) LeNet and Gaussian attack.    (b) LeNet and Same-value attack.    (c) LeNet and Sign-flip attack.

(d) MLP and Gaussian attack.    (e) MLP and LIE attack.    (f) MLP and FoE attack.

Figure 12: The running time (s) over 500 iterations for Fed-NGA and baselines is evaluated on the MNIST dataset and $\beta = 0.2$ across three different Byzantine ratios.

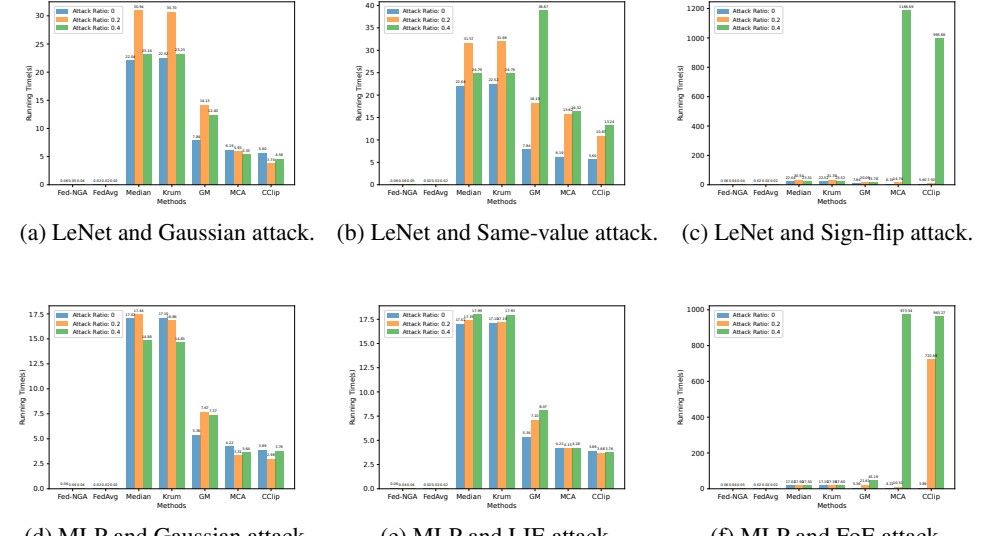

(a) LeNet and Gaussian attack.    (b) LeNet and Same-value attack.    (c) LeNet and Sign-flip attack.

(d) MLP and Gaussian attack.    (e) MLP and LIE attack.    (f) MLP and FoE attack.

Figure 13: The running time (s) over 500 iterations for Fed-NGA and baselines is evaluated on the CIFAR10 dataset and $\beta = 0.4$ across three different Byzantine ratios.

**CIFAR10 Dataset:** Similarly to the case of the MNIST dataset, Fed-NGA outperforms the baselines in terms of aggregated running time. Due to the increased complexity of the dataset and learning models, the aggregation time cost for Fed-NGA and the baselines is significantly higher than that observed for the MNIST dataset. As shown in Table 8, the aggregation time cost for the robust baselines is at least 60 times, and in some cases up to 500 times, that of our proposed Fed-NGA algorithm. Moreover, consistent with the MNIST results, Figure 13 illustrates that while Fed-NGA requires only 2 to 2.5 times the aggregation cost of FedAvg, it achieves robustness against Byzantine attacks with a time cost that is merely 1/60 of other robust algorithms. These results reaffirm the fast and efficient nature of Fed-NGA.

Table 8: The running time (in seconds) for 500 iterations of Fed-NGA and robust baselines is evaluated under varying types and ratios of Byzantine attacks, as well as different concentration parameters, on the CIFAR10 dataset using two learning models. Notably, only the running time of the aggregation algorithms is considered, excluding the model training time.

| Model | $\beta$ | $\bar{C}_{rs}$ / Algorithm | No Attack | Gaussian Attack | | | | Same-value Attack | | | | Sign-flip Attack | | | | LIE Attack | | | | FoE Attack | | | |
|---|---|---|---|---|---|---|---|---|---|---|---|---|---|---|---|---|---|---|---|---|---|---|---|
| | | | 0 | 0.1 | 0.2 | 0.3 | 0.4 | 0.1 | 0.2 | 0.3 | 0.4 | 0.1 | 0.2 | 0.3 | 0.4 | 0.1 | 0.2 | 0.3 | 0.4 | 0.1 | 0.2 | 0.3 | 0.4 |
| LeNet | 0.6 | Fed-NGA | 0.0558 | 0.0491 | 0.0507 | 0.0491 | 0.0585 | 0.0451 | 0.0460 | 0.0454 | 0.0454 | 0.0452 | 0.0446 | 0.0557 | 0.0430 | 0.0450 | 0.0438 | 0.0422 | 0.0424 | 0.0452 | 0.0446 | 0.0557 | 0.0430 |
| | | Median | 22.163 | 30.729 | 31.249 | 31.063 | 23.112 | 31.095 | 31.958 | 32.196 | 24.563 | 30.753 | 31.058 | 30.416 | 23.116 | 30.317 | 30.008 | 31.061 | 23.246 | 30.504 | 30.798 | 30.357 | 23.203 |
| | | Krum | 21.773 | 30.630 | 30.716 | 31.079 | 22.927 | 30.988 | 31.989 | 32.516 | 24.626 | 32.290 | 30.370 | 31.037 | 22.877 | 30.889 | 30.478 | 31.052 | 23.316 | 30.715 | 30.604 | 30.923 | 23.059 |
| | | GM | 7.8912 | 13.707 | 14.761 | 15.172 | 12.563 | 14.559 | 18.281 | 26.252 | 37.983 | 16.459 | 18.627 | 18.994 | 15.464 | 12.923 | 13.942 | 14.140 | 11.839 | 48.862 | 19.998 | 36.934 | 36.097 |
| | | MCA | 6.3659 | 6.1991 | 6.074 | 7.3452 | 5.3612 | 7.6208 | 14.926 | 25.098 | 16.313 | 12.393 | 14.667 | 1369.4 | 1211.5 | 7.5085 | 7.3273 | 7.3201 | 5.4143 | 12.393 | 14.667 | 1369.4 | 1211.5 |
| | | CClip | 5.6009 | 3.6275 | 3.6749 | 4.6567 | 4.6492 | 4.6610 | 10.3312 | 16.016 | 14.418 | 6.3801 | 977.80 | 1006.7 | 980.22 | 4.5891 | 4.4772 | 4.6273 | 4.6075 | 6.3801 | 977.80 | 1006.7 | 980.22 |
| | 0.4 | Fed-NGA | 0.0557 | 0.0456 | 0.0454 | 0.0445 | 0.0450 | 0.0462 | 0.0557 | 0.0469 | 0.0452 | 0.0461 | 0.0447 | 0.0433 | 0.0429 | 0.0457 | 0.0436 | 0.0424 | 0.0425 | 0.0461 | 0.0447 | 0.0433 | 0.0429 |
| | | Median | 22.039 | 30.774 | 30.944 | 31.192 | 23.162 | 31.465 | 31.573 | 32.464 | 24.790 | 30.886 | 30.539 | 30.209 | 23.506 | 31.222 | 30.917 | 31.118 | 23.513 | 30.724 | 30.512 | 30.343 | 23.103 |
| | | Krum | 22.516 | 30.753 | 30.695 | 31.162 | 23.232 | 31.479 | 31.956 | 31.896 | 24.786 | 34.600 | 31.390 | 31.226 | 23.524 | 30.683 | 31.140 | 31.203 | 23.335 | 31.020 | 30.768 | 31.105 | 20.798 |
| | | GM | 7.8409 | 13.287 | 14.127 | 15.230 | 12.402 | 14.506 | 18.187 | 26.443 | 38.870 | 23.413 | 20.085 | 18.022 | 15.782 | 12.751 | 13.689 | 14.221 | 11.597 | 46.245 | 28.602 | 45.415 | 12.222 |
| | | MCA | 6.1858 | 6.1785 | 5.9279 | 7.5666 | 5.3525 | 7.592 | 15.816 | 26.134 | 16.317 | 11.111 | 14.738 | 1372.1 | 1186.6 | 7.5115 | 7.3110 | 7.3616 | 5.3096 | 11.111 | 14.738 | 1372.1 | 1186.6 |
| | | CClip | 5.6043 | 3.6822 | 3.7389 | 4.6671 | 4.5781 | 4.6254 | 10.866 | 16.583 | 13.239 | 6.5340 | 7.5022 | 1004.9 | 995.68 | 4.5509 | 4.5971 | 4.5906 | 4.6077 | 6.5340 | 7.5022 | 1004.9 | 995.68 |
| MLP | 0.6 | Fed-NGA | 0.0548 | 0.0476 | 0.0471 | 0.0463 | 0.0462 | 0.0460 | 0.0541 | 0.0565 | 0.0440 | 0.0556 | 0.0531 | 0.0426 | 0.0425 | 0.0454 | 0.0427 | 0.0422 | 0.0422 | 0.0556 | 0.0531 | 0.0426 | 0.0425 |
| | | Median | 14.848 | 22.815 | 20.519 | 17.324 | 17.100 | 21.613 | 19.356 | 16.380 | 16.651 | 22.223 | 21.085 | 17.383 | 17.693 | 22.763 | 20.880 | 17.384 | 16.578 | 16.982 | 18.442 | 15.850 | 15.898 |
| | | Krum | 14.723 | 22.458 | 21.266 | 17.284 | 16.907 | 22.164 | 21.124 | 15.996 | 15.551 | 22.304 | 20.747 | 17.508 | 16.853 | 22.219 | 21.242 | 17.263 | 17.175 | 15.302 | 14.829 | 16.149 | 15.390 |
| | | GM | 4.7015 | 8.9260 | 9.3729 | 9.9517 | 8.5036 | 10.499 | 12.349 | 15.369 | 26.609 | 7.6535 | 9.4907 | 11.405 | 10.420 | 8.1844 | 8.6524 | 7.1553 | 7.9380 | 26.384 | 33.353 | 17.151 | 72.127 |
| | | MCA | 3.7877 | 4.1849 | 4.1655 | 4.1178 | 4.1442 | 5.4874 | 10.919 | 15.374 | 12.494 | 6.4467 | 7.6997 | 1103.8 | 1117.5 | 5.0190 | 4.9915 | 4.0936 | 4.1376 | 6.4467 | 7.6997 | 1103.8 | 1117.5 |
| | | CClip | 3.9597 | 3.0945 | 3.0215 | 3.8030 | 3.7894 | 3.7987 | 8.2872 | 13.733 | 11.526 | 5.4316 | 791.55 | 956.93 | 968.22 | 3.7747 | 3.6873 | 3.7801 | 3.7786 | 5.4316 | 791.55 | 956.93 | 968.22 |
| | 0.4 | Fed-NGA | 0.0551 | 0.0453 | 0.0446 | 0.0444 | 0.0441 | 0.0461 | 0.0454 | 0.0443 | 0.0441 | 0.0449 | 0.0428 | 0.0531 | 0.0511 | 0.0450 | 0.0417 | 0.0420 | 0.0415 | 0.0449 | 0.0428 | 0.0531 | 0.0511 |
| | | Median | 17.021 | 18.7756 | 17.436 | 18.219 | 14.857 | 18.691 | 17.748 | 18.330 | 17.925 | 18.874 | 18.065 | 18.399 | 17.781 | 18.900 | 17.352 | 18.215 | 17.988 | 18.678 | 17.898 | 18.048 | 17.502 |
| | | Krum | 17.103 | 18.8529 | 16.856 | 18.051 | 14.654 | 18.854 | 17.229 | 18.185 | 18.033 | 20.405 | 17.520 | 18.139 | 18.021 | 18.827 | 17.187 | 18.294 | 17.929 | 19.224 | 17.361 | 18.100 | 17.601 |
| | | GM | 5.3557 | 7.3474 | 7.6709 | 8.1666 | 7.3699 | 9.0445 | 10.698 | 15.807 | 28.996 | 10.100 | 11.287 | 9.9554 | 10.764 | 6.8215 | 7.1002 | 7.5882 | 8.0680 | 35.066 | 21.614 | 39.320 | 45.192 |
| | | MCA | 4.2199 | 3.3581 | 3.3128 | 4.2743 | 3.6447 | 4.4169 | 10.012 | 14.732 | 12.680 | 8.4247 | 10.512 | 963.14 | 973.54 | 4.2964 | 4.1473 | 4.3099 | 4.1839 | 8.4247 | 10.512 | 963.14 | 973.54 |
| | | CClip | 3.8866 | 3.0917 | 2.9789 | 3.7529 | 3.7627 | 3.7550 | 8.7779 | 12.988 | 11.422 | 5.3783 | 722.88 | 958.34 | 963.27 | 3.7574 | 3.6842 | 3.7557 | 3.7600 | 5.3783 | 722.88 | 958.34 | 963.27 |

Table 9: The running time (in seconds) for 100 iterations of Fed-NGA and robust baselines is evaluated under varying types and ratios of Byzantine attacks, as well as different concentration parameters, on the TinyImageNet dataset using MobileNetV3 model. Notably, only the running time of the aggregation algorithms is considered, excluding the model training time.

| Model | $\beta$ | $\bar{C}_{rs}$ / Algorithm | No Attack | Gaussian Attack | | | | Same-value Attack | | | | Sign-flip Attack | | | | LIE Attack | | | | FoE Attack | | | |
|---|---|---|---|---|---|---|---|---|---|---|---|---|---|---|---|---|---|---|---|---|---|---|---|
| | | | 0 | 0.1 | 0.2 | 0.3 | 0.4 | 0.1 | 0.2 | 0.3 | 0.4 | 0.1 | 0.2 | 0.3 | 0.4 | 0.1 | 0.2 | 0.3 | 0.4 | 0.1 | 0.2 | 0.3 | 0.4 |
| MobileNetV3 | 0.6 | Fed-NGA | 0.2260 | 0.1945 | 0.1942 | 0.1943 | 0.1946 | 0.1946 | 0.1951 | 0.1946 | 0.1946 | 0.2004 | 0.2011 | 0.3729 | 0.2393 | 0.1914 | 0.1912 | 0.1912 | 0.1912 | 0.2934 | 0.3367 | 0.2374 | 0.2079 |
| | | Median | 0.3333 | 0.2981 | 0.2927 | 0.2939 | 0.2941 | 0.2917 | 0.2929 | 0.2950 | 0.2944 | 0.2976 | 0.2974 | 0.4640 | 0.3310 | 0.2887 | 0.2893 | 0.2894 | 0.2898 | 0.2977 | 0.3732 | 0.3345 | 0.2990 |
| | | Krum | 50.902 | 49.488 | 49.322 | 49.107 | 48.917 | 49.431 | 49.567 | 49.089 | 48.766 | 49.715 | 50.199 | 50.445 | 50.386 | 49.548 | 49.830 | 49.643 | 49.559 | 49.857 | 50.391 | 50.140 | 50.546 |
| | | GM | 5.1880 | 5.0464 | 4.9234 | 4.5961 | 4.6844 | 5.1322 | 5.4841 | 6.3364 | 8.6997 | 6.0003 | 6.9124 | 7.8189 | 10.258 | 5.7212 | 5.8732 | 5.9651 | 5.9841 | 8.4599 | 11.497 | 29.273 | 18.405 |
| | | MCA | 3.5246 | 3.1642 | 2.9496 | 2.9277 | 2.6893 | 3.8002 | 5.2586 | 9.9156 | 12.110 | 4.4433 | 6.6065 | 281.55 | 284.46 | 3.8997 | 4.0824 | 4.1356 | 4.0803 | 4.0875 | 4.6278 | 4.7569 | 4.8875 |
| | | CClip | 2.5082 | 2.5023 | 2.2738 | 2.0861 | 1.8945 | 2.4371 | 2.2174 | 2.0012 | 1.8296 | 3.1602 | 3.5504 | 3.8123 | 3.7472 | 2.9279 | 3.1020 | 3.1577 | 3.0882 | 3.1429 | 3.5263 | 3.5499 | 3.5924 |
| | 0.4 | Fed-NGA | 0.2318 | 0.2108 | 0.1894 | 0.1894 | 0.1884 | 0.1899 | 0.1893 | 0.1892 | 0.1882 | 0.1970 | 0.1947 | 0.3776 | 0.2364 | 0.1855 | 0.1851 | 0.1858 | 0.1851 | 0.1960 | 0.3059 | 0.2388 | 0.2058 |
| | | Median | 0.3336 | 0.3084 | 0.2890 | 0.2882 | 0.2903 | 0.2880 | 0.2877 | 0.2879 | 0.2871 | 0.2945 | 0.2945 | 0.4779 | 0.3354 | 0.2845 | 0.2856 | 0.2854 | 0.2850 | 0.2974 | 0.3932 | 0.3387 | 0.3000 |
| | | Krum | 49.820 | 47.644 | 48.032 | 47.665 | 47.388 | 47.981 | 48.014 | 47.437 | 47.259 | 48.295 | 48.795 | 49.187 | 49.021 | 48.053 | 48.185 | 48.269 | 48.295 | 48.351 | 48.676 | 48.749 | 48.489 |
| | | GM | 5.1997 | 5.1261 | 4.8875 | 4.8087 | 4.6901 | 5.1526 | 5.3955 | 6.2347 | 8.6650 | 5.7837 | 6.8220 | 7.7338 | 10.474 | 5.5812 | 5.7961 | 5.9185 | 5.9249 | 8.7960 | 12.340 | 74.143 | 15.158 |
| | | MCA | 3.4985 | 3.3428 | 3.1068 | 3.0029 | 2.8139 | 3.8375 | 5.2622 | 9.9946 | 21.244 | 4.4326 | 213.20 | 281.49 | 287.46 | 3.8653 | 4.0282 | 4.0924 | 4.0250 | 4.0809 | 4.5840 | 4.7030 | 4.8649 |
| | | CClip | 2.5634 | 2.4380 | 2.2277 | 2.0405 | 1.8496 | 2.4447 | 2.2495 | 2.0578 | 1.8540 | 3.1309 | 3.5369 | 3.8138 | 3.7155 | 2.8951 | 3.0692 | 3.1245 | 3.0553 | 3.1218 | 3.4861 | 3.5293 | 3.5962 |

**TinyImageNet Dataset:** Fed-NGA maintains its computational efficiency advantage over robust baselines on the TinyImageNet benchmark. As evidenced in Table 9, our method achieves state-of-the-art (SoTA) time efficiency across all robust aggregation benchmarks. The baseline methods demonstrate 1.5× higher aggregation time costs compared to Fed-NGA's optimized workflow. Notably, Figure 14 reveals that while Fed-NGA marginally trails FedAvg in absolute speed - an expected trade-off given FedAvg's vulnerability to Byzantine attacks - it achieves superior robustness-efficiency balance. These empirical results conclusively validate Fed-NGA's time complexity advantages in large-scale FL scenarios.

## C.3 Convergence Performance of Data Heterogeneity

In this section, we analyze the training performance under different data heterogeneity concentration parameter setups, as illustrated in Figure 15, Figure 16, and Figure 17. These figures depict the

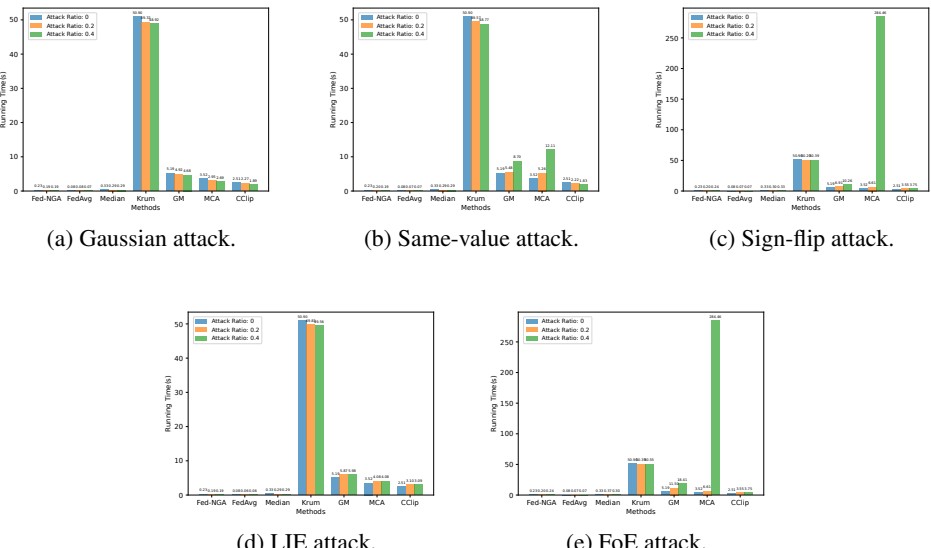

(a) Gaussian attack.  (b) Same-value attack.  (c) Sign-flip attack.

(d) LIE attack.  (e) FoE attack.

Figure 14: The running time (s) over 100 iterations for Fed-NGA and baselines is evaluated on the TinyImageNet dataset, MobileNetV3 model and $\beta = 0.6$ across three different Byzantine ratios.

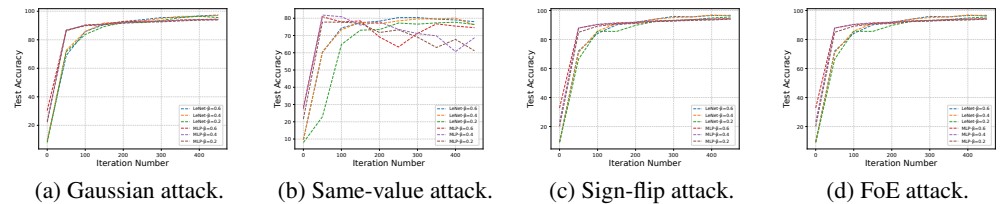

(a) Gaussian attack.  (b) Same-value attack.  (c) Sign-flip attack.  (d) FoE attack.

Figure 15: The test accuracy (%) over 500 iterations for Fed-NGA is evaluated on the MNIST dataset and $\bar{C}_\alpha = 0.4$ across three different data heterogeneity concentration parameters.

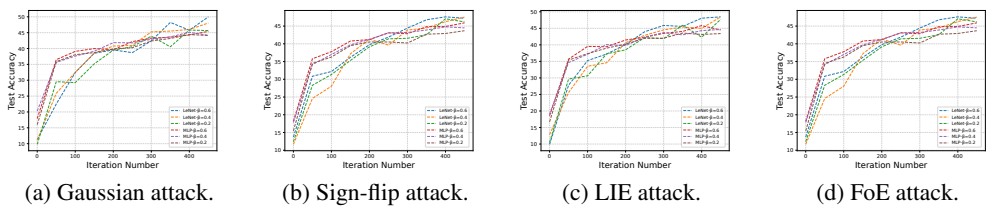

(a) Gaussian attack.  (b) Sign-flip attack.  (c) LIE attack.  (d) FoE attack.

Figure 16: The test accuracy (%) over 500 iterations for Fed-NGA is evaluated on the CIFAR10 dataset and $\bar{C}_\alpha = 0.3$ across three different data heterogeneity concentration parameters.

test accuracy trends for Fed-NGA under three data heterogeneity concentration parameter setups, five types of Byzantine attacks, and three learning models, applied to the MNIST, CIFAR10, and TinyImageNet datasets, respectively. In the subsequent analysis, we evaluate the performance of Fed-NGA and baseline methods in relation to the dataset type.

**MNIST Dataset:** From Figure 15, it is evident that as data heterogeneity increases, indicated by smaller concentration parameter values, the training performance of Fed-NGA declines. However, the performance degradation remains within an acceptable range, typically resulting in only a 1% to 2% gap. This observation demonstrates the strong adaptability of Fed-NGA to varying levels of data heterogeneity.

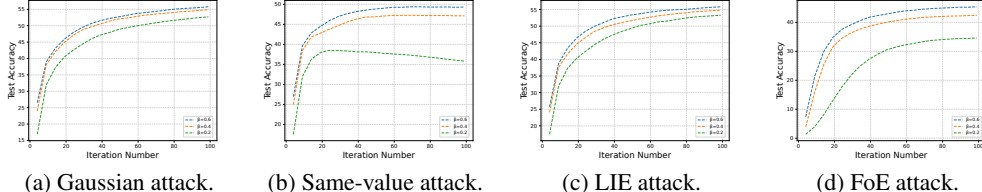

|            |                 |           |            |
|:----------:|:---------------:|:---------:|:----------:|
| (a) Gaussian attack. | (b) Same-value attack. | (c) LIE attack. | (d) FoE attack. |

Figure 17: The test accuracy (%) over 100 iterations for Fed-NGA is evaluated on the TinyImageNet dataset, MobileNetV3 model, and $\bar{C}_\alpha = 0.2$ across three different data heterogeneity concentration parameters.

**CIFAR10 Dataset:** As shown in Figure 16, the test accuracy trend of Fed-NGA is consistent with the observations for the MNIST dataset. When comparing different data heterogeneity concentration parameter setups, it is evident that the choice of learning model has a more significant impact on performance. Notably, the MLP model demonstrates greater stability across varying data heterogeneity concentration parameter setups compared to the LeNet model, although the maximum test accuracy achieved by the LeNet model is higher than that of the MLP model. Nonetheless, the performance degradation remains within an acceptable range, typically resulting in only a 2% to 3% gap.

**TinyImageNet Dataset:** Figure 17 reveals Fed-NGA's accuracy dependence on data heterogeneity levels under varying Byzantine attack patterns. Specifically, the framework exhibits heightened sensitivity to data distribution skews under Same-value and FoE attacks, where heterogeneity-induced accuracy degradation exceeds in extreme cases. Conversely, for Gaussian attack and LIE attack, Fed-NGA demonstrates relative stability with marginal accuracy losses at low heterogeneity levels. This divergence from MNIST/CIFAR10 benchmarks likely stems from MobileNetV3's enhanced parameter space and TinyImageNet's complex feature hierarchies, which amplify both attack surface vulnerabilities and model resilience characteristics.

# D   Related Works in Detail

FL is firstly proposed in (McMahan et al., 2017). The security issue in FL has been focused on since the emergence of FL (Vempaty et al., 2013; Yang et al., 2020). Byzantine attack, as a common mean of distributed attacking, has also caught considerable research attention recently. In related works dealing with Byzantine attacks, various assumptions are made for loss function's convexity (strongly convex or non-convex), and data heterogeneity (IID dataset or non-IID dataset). Furthermore, time complexity for aggregation is also a concerned performance metric as pointed out in Section 1. A detailed introduction of them is given in the following subsections. Also a summary of these related works' assumptions and performance is listed in Table 1.

## D.1   Aggregation Strategy of Byzantine-resilient FL Algorithms

As for aggregation strategies in related literature, the measures to tolerate Byzantine attacks can be categorized as median, Krum, geometric median, etc. For the methods based on **Median:** Xie et al. (2018) takes the coordinate-wise median of the uploaded vectors as the aggregated one, while Yin et al. (2018) aggregates the uploaded vectors by calculating their coordinate-wise median or coordinate-wise trimmed mean. Differently, the Robust Aggregating Normalized Gradient Method (RANGE) (Turan et al., 2022) takes the median of uploaded vectors from each client as the aggregated one, and each uploaded vector is supposed to be the median of the local gradients from multiple steps of local updating. For the methods based on **Krum:** Blanchard et al. (2017) selects a stochastic gradient as the global one, which has the shortest Euclidean distance from a group of stochastic gradients that are most closely distributed with the total number of Byzantine clients given. For the methods based on **Geometric median:** all existing methods, including Robust Federated Aggregation (RFA) (Pillutla et al., 2022), Byzantine attack resilient distributed Stochastic Average Gradient Algorithm (Byrd-SAGA) (Wu et al., 2020), and Byzantine-RObust Aggregation with gradient Difference Compression And STochastic variance reduction (BROADCAST) (Zhu and Ling, 2023), leverage the geometric median to aggregate upload vectors. Differently, RFA selects the tail-average of local model parameter

over multiple local updating as the upload vector, both Byrd-SAGA and BROADCAST utilize the SAGA method (Defazio et al., 2014) to generate the upload message while the latter one allows uploading compressed message. For **Other methods:** Robust Stochastic Aggregation (RSA) (Li et al., 2019) penalizes the difference between local model parameters and global model parameters by $l_d$ norm to isolate Byzantine clients. And Maximum Correntropy Aggregation (MCA) (Luan et al., 2024) leverages maximum correntropy to aggregate the aggregate upload vectors. Centered Clipping (CClip) (Karimireddy et al., 2021) employs the previously aggregated gradient as a reference, clips the modulus of the gradient vector, and then performs aggregation.

### D.2 Assumptions and Performance of Byzantine-resilient FL Algorithms

For the methods based on **Median:** the aggregation complexity of median methods is all $\mathcal{O}(pM \log(M))$. To see the difference, Xie et al. (2018) verifies the robustness of Byzantine attacks on IID datasets only by experiments, while both trimmed mean (Yin et al., 2018) and RANGE (Turan et al., 2022) offer theoretical analysis for non-convex and strongly convex loss functions on IID datasets. For the methods based on **Krum:** the aggregation complexity of Blanchard et al. (2017) is $\mathcal{O}(pM^2)$ and convergence analysis is established on non-convex loss function and IID datasets. For the methods based on **Geometric median:** the time complexity for aggregation is $\mathcal{O}(pM \log^3(M\epsilon^{-1}))$ (Cohen et al., 2016) for all the three methods (Pillutla et al., 2022; Wu et al., 2020; Zhu and Ling, 2023), which exhibit the robustness to Byzantine attacks under strongly convex loss function. Only RFA (Pillutla et al., 2022) reveals the proof on non-IID datasets, while both Byrd-SAGA (Wu et al., 2020) and BROADCAST (Zhu and Ling, 2023) are established on IID datasets. For **Other methods:** RSA (Li et al., 2019) exhibits an aggregation complexity of $\mathcal{O}(Mp^{3.5})$, with its convergence performance analyzed on heterogeneous datasets for strongly convex loss functions. Similarly, MCA (Luan et al., 2024), which has an aggregation complexity of $\mathcal{O}(pM \log^3(M\epsilon^{-1}))$, has its convergence performance evaluated on heterogeneous datasets for strongly convex loss function. The CClip algorithm (Karimireddy et al., 2021), tested under IID dataset with non-convex function, exhibits $\mathcal{O}(\tau Mp)$ aggregation complexity. Last but not least, the optimality gap of all the above algorithms is non-zero.

In contrast to the aforementioned studies, we focus on non-convex objectives under non-IID data distributions. Moreover, the proposed Fed-NGA algorithm exhibits a computational complexity of only $\mathcal{O}(pM)$, which is strictly lower than that of all existing Byzantine-robust methods. Despite its simplicity, Fed-NGA achieves convergence and even zero optimality gap under mild conditions, which have not been theoretically established by previous Byzantine-robust approaches.

## E   Proof of Theorem 3.7

Under Assumption 3.1, and recalling (9) and (10), we have

$$
\begin{aligned}
&\mathbb{E}\left\{F(w^{t+1}) - F(w^t)\right\} \\
&\leqslant \mathbb{E}\left\{\left\langle \nabla F(w^t), w^{t+1} - w^t \right\rangle + \frac{L}{2}\left\|w^{t+1} - w^t\right\|^2\right\} \\
&= \eta^t \underbrace{\left\langle -\nabla F(w^t), \mathbb{E}\left\{g^t\right\}\right\rangle}_{A_1} + \frac{L}{2}(\eta^t)^2 \underbrace{\mathbb{E}\left\{\left\|g^t\right\|^2\right\}}_{A_2}.
\end{aligned}
\tag{24}
$$

$A_2$ can be easily bounded because of Jensen's inequality $f\left(\sum_{m\in\mathcal{M}} \alpha_m x_m\right) \leqslant \sum_{m\in\mathcal{M}} \alpha_m f(x_m)$ with $\sum_{m\in\mathcal{M}} \alpha_m = 1, \alpha_m > 0$ for $m \in \mathcal{M}$ and $f(x) = x^2$, which implies that

$$A_2 = \mathbb{E}\left\{\left\|g^t\right\|^2\right\}$$

$$= \mathbb{E}\left\{\left\|\sum_{m \in \mathcal{M}} \alpha_m \cdot \frac{g_m^t}{\|g_m^t\|}\right\|^2\right\}$$

$$\leqslant \mathbb{E}\left\{\sum_{m \in \mathcal{M}} \alpha_m \left\|\frac{g_m^t}{\|g_m^t\|}\right\|^2\right\}$$

$$= 1. \tag{25}$$

Next, we aim to bound $A_1$. According to (10), there is

$$A_1 = \left\langle -\nabla F(w^t), \mathbb{E}\left\{g^t\right\}\right\rangle$$

$$= -\left\langle \nabla F(w)^t, \mathbb{E}\left\{\sum_{m \in \mathcal{M}} \alpha_m \cdot \frac{g_m^t}{\|g_m^t\|}\right\}\right\rangle$$

$$= -\sum_{m \in \mathcal{M}} \alpha_m \left\langle \nabla F(w^t), \mathbb{E}\left\{\frac{g_m^t}{\|g_m^t\|}\right\}\right\rangle$$

$$= \underbrace{-\sum_{m \in \mathcal{M}\setminus\mathcal{B}} \alpha_m \left\langle \nabla F(w^t), \mathbb{E}\left\{\frac{g_m^t}{\|g_m^t\|}\right\}\right\rangle}_{B_1} \underbrace{-\sum_{m \in \mathcal{B}} \alpha_m \mathbb{E}\left\{\left\langle \nabla F(w^t), \frac{g_m^t}{\|g_m^t\|}\right\rangle\right\}}_{B_2}. \tag{26}$$

$B_2$ is easier to be handled, so we first consider the bound of $B_2$. Because of $-\langle x, y\rangle \leqslant \|x\| \cdot \|y\|$, we have

$$B_2 = -\sum_{m \in \mathcal{B}} \alpha_m \left\langle \nabla F(w^t), \mathbb{E}\left\{\frac{g_m^t}{\|g_m^t\|}\right\}\right\rangle$$

$$\leqslant \sum_{m \in \mathcal{B}} \alpha_m \left\|\nabla F(w^t)\right\| \cdot \left\|\frac{g_m^t}{\|g_m^t\|}\right\|$$

$$\overset{①}{=} \sum_{m \in \mathcal{B}} \alpha_m \left\|\nabla F(w^t)\right\|$$

$$\overset{②}{=} (1 - C_\alpha) \left\|\nabla F(w^t)\right\|, \tag{27}$$

where ① follows from $\left\|\dfrac{g_m^t}{\|g_m^t\|}\right\| = 1$, and ② comes from the definition of $C_\alpha$ in (7).

Then we aim to bound $B_1$, which implies that

$$B_1 = -\sum_{m \in \mathcal{M}\setminus\mathcal{B}} \alpha_m \mathbb{E}\left\{\left\langle \nabla F(w^t), \frac{g_m^t}{\|g_m^t\|}\right\rangle\right\}$$

$$= -\sum_{m \in \mathcal{M}\setminus\mathcal{B}} \alpha_m \mathbb{E}\left\{\left\langle \nabla F(w^t), \frac{\nabla F_m(w^t, \xi_m^t)}{\|\nabla F_m(w^t, \xi_m^t)\|}\right\rangle\right\}$$

$$= -\sum_{m \in \mathcal{M}\setminus\mathcal{B}} \alpha_m \mathbb{E}\left\{\frac{\|\nabla F(w^t)\|^2 + \langle \nabla F(w^t), \nabla F_m(w^t, \xi_m^t) - \nabla F(w^t)\rangle}{\|\nabla F_m(w^t, \xi_m^t)\|}\right\} \tag{28}$$

We first assume that $\left\|\nabla F(w^t)\right\| \geq \rho\left\|\nabla F_m(w^t, \xi_m^t) - \nabla F(w^t)\right\|$, there is

$$-\mathbb{E}\left\{\frac{\left\|\nabla F(w^t)\right\|^2 + \left\langle\nabla F(w^t), \nabla F_m(w^t, \xi_m^t) - \nabla F(w^t)\right\rangle}{\left\|\nabla F_m(w^t, \xi_m^t)\right\|}\right\}$$

$$\overset{\text{①}}{\leqslant} -\mathbb{E}\left\{\frac{\left\|\nabla F(w^t)\right\|^2 - \left\|\nabla F(w^t)\right\|\left\|\nabla F_m(w^t, \xi_m^t) - \nabla F(w^t)\right\|}{\left\|\nabla F_m(w^t, \xi_m^t)\right\|}\right\}$$

$$\overset{\text{②}}{\leqslant} -\mathbb{E}\left\{\frac{(\rho - 1)\left\|\nabla F(w^t)\right\|^2}{\rho\left\|\nabla F(w^t) + \nabla F_m(w^t, \xi_m^t) - \nabla F(w^t)\right\|}\right\}$$

$$\overset{\text{③}}{\leqslant} -\mathbb{E}\left\{\frac{(\rho - 1)\left\|\nabla F(w^t)\right\|^2}{(\rho + 1)\left\|\nabla F(w^t)\right\|}\right\}$$

$$= -\frac{\rho - 1}{\rho + 1}\left\|\nabla F(w^t)\right\| \tag{29}$$

where ① comes from $-\left\langle a, b\right\rangle \leq \|a\|\|b\|$, ② comes from $\left\|\nabla F(w^t)\right\| \geq \rho\left\|\nabla F_m(w^t, \xi_m^t) - \nabla F(w^t)\right\|$, and ③ comes from $\|a + b\| \leq \|a\| + \|b\|$ and $\left\|\nabla F(w^t)\right\| \geq \rho\left\|\nabla F_m(w^t, \xi_m^t) - \nabla F(w^t)\right\|$.

Then considering $\left\|\nabla F(w^t)\right\| \leq \rho\left\|\nabla F_m(w^t, \xi_m^t) - \nabla F(w^t)\right\|$, we have,

$$-\mathbb{E}\left\{\left\langle\nabla F(w^t), \frac{\nabla F_m(w^t, \xi_m^t)}{\left\|\nabla F_m(w^t, \xi_m^t)\right\|}\right\rangle\right\} \leq \left\|\nabla F(w^t)\right\|$$

$$= -\frac{\rho - 1}{\rho + 1}\left\|\nabla F(w^t)\right\| + \frac{2\rho}{\rho + 1}\left\|\nabla F(w^t)\right\|$$

$$= -\frac{\rho - 1}{\rho + 1}\left\|\nabla F(w^t)\right\| + \frac{2\rho^2}{\rho + 1}\mathbb{E}\left\|\nabla F_m(w^t, \xi_m^t) - \nabla F(w^t)\right\| \tag{30}$$

Due to $\left\|\nabla F_m(w^t, \xi_m^t) - \nabla F(w^t)\right\| > 0$, Assumption 3.2, Assumption 3.3, and Assumption 3.4, there is

$$B_1 \leq -\sum_{m \in \mathcal{M}\backslash\mathcal{B}} \alpha_m\left(\frac{\rho - 1}{\rho + 1}\left\|\nabla F(w^t)\right\| + \frac{2\rho^2}{\rho + 1}\mathbb{E}\left\|\nabla F_m(w^t, \xi_m^t) - \nabla F(w^t)\right\|\right)$$

$$= -\sum_{m \in \mathcal{M}\backslash\mathcal{B}} \alpha_m\left(\frac{\rho - 1}{\rho + 1}\left\|\nabla F(w^t)\right\| + \frac{2\rho^2}{\rho + 1}\mathbb{E}\left\|\nabla F_m(w^t, \xi_m^t) - F_m(w^t) + F_m(w^t) - \nabla F(w^t)\right\|\right)$$

$$\leq -\frac{\rho - 1}{\rho + 1}C_\alpha\left\|\nabla F(w^t)\right\| + \frac{2\rho^2}{\rho + 1}C_\alpha(\sigma + \theta) \tag{31}$$

Then $A_1$ can be bounded as follow,

$$A_1 \leq -\left(\frac{2\rho}{\rho + 1}C_\alpha - 1\right)\left\|\nabla F(w^t)\right\| + \frac{2\rho^2}{\rho + 1}C_\alpha(\sigma + \theta) \tag{32}$$

Then we have

$$\mathbb{E}\left\{F(w^{t+1}) - F(w^t)\right\} \leq -\eta^t\left(\frac{2\rho}{\rho + 1}C_\alpha - 1\right)\left\|\nabla F(w^t)\right\| + \frac{2\rho^2}{\rho + 1}C_\alpha\eta^t(\sigma + \theta) + \frac{L}{2}(\eta^t)^2 \tag{33}$$

Let $\frac{2\rho}{\rho + 1}C_\alpha - 1 > 0$, we have

$$\frac{1}{\sum_{t=0}^{T-1}\eta^t}\sum_{t=0}^{T-1}\left(\frac{2\rho}{\rho + 1}C_\alpha - 1\right)\eta^t\left\|\nabla F(w^t)\right\| \leq \frac{\mathbb{E}\left\{F(w^0) - F(w^T)\right\}}{\sum_{t=0}^{T-1}\eta^t} + \frac{L\sum_{t=0}^{T-1}(\eta^t)^2}{2\sum_{t=0}^{T-1}\eta^t} + \frac{2\rho^2}{\rho + 1}C_\alpha(\sigma + \theta) \tag{34}$$

This completes the proof of Theorem 3.7.

# F Proof of Theorem 3.9

Regarding the proof of Theorem 3.9, only the proof of $B_1$ is different. Then we aim to bound $B_1$, which implies that

$$
\begin{aligned}
B_1 &= -\sum_{m\in\mathcal{M}\backslash\mathcal{B}} \alpha_m \mathbb{E}\left\{\left\langle \nabla F(w^t), \frac{g_m^t}{\|g_m^t\|}\right\rangle\right\} \\
&= -\sum_{m\in\mathcal{M}\backslash\mathcal{B}} \alpha_m \mathbb{E}\left\{\left\langle \nabla F(w^t), \frac{\nabla F_m(w^t,\xi_m^t)}{\|\nabla F_m(w^t,\xi_m^t)\|}\right\rangle\right\} \\
&= -\sum_{m\in\mathcal{M}\backslash\mathcal{B}} \alpha_m \left(\left\langle \nabla F(w^t), \frac{\nabla F(w^t)}{\|\nabla F(w^t)\|}\right\rangle + \left\langle \nabla F(w^t), \frac{\nabla F_m(w^t)}{\|\nabla F_m(w^t)\|} - \frac{\nabla F(w^t)}{\|\nabla F(w^t)\|}\right\rangle\right. \\
&\quad \left. + \mathbb{E}\left\{\left\langle \nabla F(w^t), \frac{\nabla F_m(w^t,\xi_m^t)}{\|\nabla F_m(w^t,\xi_m^t)\|} - \frac{\nabla F_m(w^t)}{\|\nabla F_m(w^t)\|}\right\rangle\right\}\right).
\end{aligned}
\tag{35}
$$

Because of $\langle x,x\rangle = \|x\|^2$ and the definition of $C_\alpha$ in (7), there is,

$$
\begin{aligned}
B_1 &= -\sum_{m\in\mathcal{M}\backslash\mathcal{B}} \alpha_m \left(\left\langle \nabla F(w^t), \frac{\nabla F_m(w^t)}{\|\nabla F_m(w^t)\|} - \frac{\nabla F(w^t)}{\|\nabla F(w^t)\|}\right\rangle + \mathbb{E}\left\{\left\langle \nabla F(w^t), \frac{\nabla F_m(w^t,\xi_m^t)}{\|\nabla F_m(w^t,\xi_m^t)\|} - \frac{\nabla F_m(w^t)}{\|\nabla F_m(w^t)\|}\right\rangle\right\}\right) \\
&\quad - C_\alpha \|\nabla F(w^t)\| \\
&\overset{①}{\leqslant} -\sum_{m\in\mathcal{M}\backslash\mathcal{B}} \alpha_m \left\langle \nabla F(w^t), \frac{\nabla F_m(w^t)}{\|\nabla F_m(w^t)\|} - \frac{\nabla F(w^t)}{\|\nabla F(w^t)\|}\right\rangle + \sigma' C_\alpha \|\nabla F(w^t)\| - C_\alpha \|\nabla F(w^t)\| \\
&\overset{②}{=} \sum_{m\in\mathcal{M}\backslash\mathcal{B}} \frac{\alpha_m \|\nabla F(w^t)\|}{2}\left(\left\|\frac{\nabla F(w^t)}{\|\nabla F(w^t)\|}\right\|^2 + \left\|\frac{\nabla F_m(w^t)}{\|\nabla F_m(w^t)\|} - \frac{\nabla F(w^t)}{\|\nabla F(w^t)\|}\right\|^2 - \left\|\frac{\nabla F_m(w^t)}{\|\nabla F_m(w^t)\|}\right\|^2\right) \\
&\quad - (1-\sigma')C_\alpha \|\nabla F(w^t)\| \\
&= \sum_{m\in\mathcal{M}\backslash\mathcal{B}} \frac{\alpha_m \|\nabla F(w^t)\|}{2}\left(1 + (\theta_m^t{}')^2 - 1\right) - (1-\sigma')C_\alpha \|\nabla F(w^t)\| \\
&\overset{③}{\leqslant} \frac{1}{2}C_\alpha (\theta')^2 \|\nabla F(w^t)\| - (1-\sigma')C_\alpha \|\nabla F(w^t)\| \\
&= -\left(1 - \frac{(\theta')^2}{2} - \sigma'\right)\cdot C_\alpha \|\nabla F(w^t)\|,
\end{aligned}
\tag{36}
$$

where ① comes from $-\langle x,y\rangle \leqslant \|x\|\|y\|$ and Assumption 3.5, ② can be derived from $\langle x,y\rangle = \frac{1}{2}\|x\|^2 + \frac{1}{2}\|y\|^2 - \frac{1}{2}\|x-y\|^2$, and ③ is bounded by Assumption 3.6.

Combining $B_1$ and $B_2$, we have

$$
\begin{aligned}
A_1 &= B_1 + B_2 \\
&\leqslant -\left(1 - \frac{(\theta')^2}{2} - \sigma'\right)\cdot C_\alpha \|\nabla F(w^t)\| + (1-C_\alpha)\|\nabla F(w^t)\| \\
&= -\left(\left(2 - \frac{(\theta')^2}{2} - \sigma'\right)C_\alpha - 1\right)\|\nabla F(w^t)\|.
\end{aligned}
\tag{37}
$$

Then combining $A_1$ and $A_2$, there is

$$
\begin{aligned}
&\mathbb{E}\left\{F(w^{t+1}) - F(w^t)\right\} \\
&\leqslant \eta^t \cdot A_1 + \frac{L}{2}(\eta^t)^2 \cdot A_2 \\
&\leqslant -\eta^t\left(\left(2 - \frac{(\theta')^2}{2} - \sigma'\right)C_\alpha - 1\right)\|\nabla F(w^t)\| + \frac{L}{2}(\eta^t)^2.
\end{aligned}
\tag{38}
$$

Summing up the inequality in (38) for $t = 0, 1, \cdots, T - 1$, we obtain

$$
\mathbb{E}\left\{ F(w^T) - F(w^0) \right\}
$$

$$
\leqslant - \left( \left( 2 - \frac{(\theta')^2}{2} - \sigma' \right) C_\alpha - 1 \right) \sum_{t=0}^{T-1} \eta^t \left\| \nabla F(w^t) \right\| + \frac{L}{2} \sum_{t=0}^{T-1} (\eta^t)^2, \tag{39}
$$

When $\left( 2 - \frac{(\theta')^2}{2} - \sigma' \right) C_\alpha - 1 > 0$, dividing the inequality in (39) by $\left( \left( 2 - \frac{(\theta')^2}{2} - \sigma' \right) C_\alpha - 1 \right) \sum_{t=0}^{T-1} \eta^t$
on both sides, we have

$$
\frac{1}{\sum_{t=0}^{T-1} \eta^t} \sum_{t=0}^{T-1} \eta^t \left\| \nabla F(w^t) \right\| \leqslant \frac{\mathbb{E}\left\{ F(w^0) - F(w^T) \right\}}{\left( (2 - \frac{(\theta')^2}{2} - \sigma') C_\alpha - 1 \right) \sum_{t=0}^{T-1} \eta^t} + \frac{L \sum_{t=0}^{T-1} (\eta^t)^2}{2 \left( (2 - \frac{(\theta')^2}{2} - \sigma') C_\alpha - 1 \right) \sum_{t=0}^{T-1} \eta^t}. \tag{40}
$$

This completes the proof of Theorem 3.9.

