# OpenReview forum: "Efficient Federated Learning against Byzantine Attacks and Data Heterogeneity via Aggregating Normalized Gradients"
_NeurIPS.cc/2025/Conference — NeurIPS 2025 poster_

### Official Review · Reviewer_Zgfg · 2025-06-23

**Clarity:** 2
**Significance:** 2
**Originality:** 3
**Rating:** 2
**Confidence:** 4

**Summary:**

This paper addresses the issue of high computational overhead when defending against Byzantine attacks in federated learning. The authors propose a Federated Normalized Gradients Algorithm (Fed-NGA), which achieves a favorable time complexity of O(pM). Extensive experiments are conducted to compare Fed-NGA with several baseline methods under various settings of data heterogeneity and Byzantine attacks.

**Questions:**

1. In line 194, what does $\delta$ represent? Is it related to the proportion of Byzantine clients, or how should it be interpreted?

**Ethical Concerns:**

["NO or VERY MINOR ethics concerns only"]

**Final Justification:**

Currently, the paper lacks a clear explanation and discussion of normalization techniques. Without such clarification, the use of normalization may appear insufficiently motivated, even if it brings theoretical or empirical gains. Given the significance of this issue,  I believe it warrants further revision and possibly another round of review. At this stage, I do not believe the paper is strong enough to warrant acceptance.

**Quality:**

2

**Strengths And Weaknesses:**

**Strengths**
1. The related work and background sections are thorough and well-documented, demonstrating a significant amount of effort.
2. The experimental section is comprehensive, including extensive comparisons across multiple datasets and attack scenarios.

**Weaknesses**
1. The algorithmic explanation lacks clarity. Specifically, the motivation behind using normalization is not well justified. Why is normalization effective in improving time complexity for aggregation under Byzantine attacks?
2. The idealized scenario of achieving a zero optimality gap discussed in Remark 3.10 relies on a very strong assumption. Therefore, it should not be presented as a main contribution of the paper.
3. As shown in Table 1, some baseline methods are analyzed under different loss function settings (e.g., strongly convex vs. non-convex). Using them in experimental comparisons without discussion may not be appropriate.
4. Although the authors justify the divergence in learning rates (Line 490) by referring to gradient magnitudes, the overall learning rate schedule remains unclear. It is not well-explained how the learning rates for baseline methods were selected, raising concerns about whether faster convergence is simply due to different learning rates.

---

> ### Author Rebuttal · Authors · 2025-07-29
>
> 1. **Regarding Weakness 1**
>
> With regard to Weakness 1, the motivation behind normalization stems from its low computational complexity compared to existing robust algorithms, as justified between Line 50 and Line 58 and detailed in Table 1. Specifically, Fed-NGA benefits from reduced aggregation time complexity through normalization: normalizing and aggregating vectors at the central server incurs a complexity of $\mathcal{O}(pM)$. This is substantially lower than that of other robust aggregation methods, such as $\mathcal{O}(pM \log(M))$ for Median and $\mathcal{O}(pM^2)$ for Krum, as reported in Table 1 on Page 2. Empirical results in Table 3 on Page 8 and Figure 3 on Page 9 further validate the algorithm’s efficiency gains in aggregation speed.
>
> 2. **Regarding Weakness 2**
>
> With regard to Weakness 2, to the best of our knowledge, no existing work has achieved a zero optimality gap, which remains the ultimate goal in convergence analysis, as also noted on Line 75 and Line 799. For the first time, we provide a theoretical condition under which a zero optimality gap can be guaranteed. Moreover, this condition is practically attainable, for example, in scenarios involving full-batch training on IID datasets, or when the inner error and data heterogeneity, as measured by gradient direction, are sufficiently constrained during training, as detailed in Appendix A.
>
> Last but not least, we would like to emphasize that this result is only one of the three theoretical contributions presented in our work, beyond which we also have algorithmic and empirical contributions. As such, this particular theoretical result is not intended to be the primary contribution of the paper.
>
> 3. **Regarding Weakness 3**
>
> With regard to Weakness 3, we would like to clarify that it is common practice in the literature to establish theoretical analysis under either the strongly convex or non-convex setting and then evaluate the proposed aggregation algorithms using standard and widely adopted training models, regardless of the specific assumptions made in the theoretical analysis. This practice is prevalent and generally accepted in our research community.
>
> For example, CClip [1], whose convergence guarantee is established under the non-convex setting, is empirically compared with RFA, which is only theoretically justified under the strongly convex setting. Conversely, MCA [2], with theoretical support under the strongly convex setting, is experimentally compared with Krum, whose analysis assumes the non-convex setting. Therefore, our evaluation protocol follows the standard experimental practices adopted in this field.
>
> [1] Karimireddy, Sai Praneeth, Lie He, and Martin Jaggi. "Learning from history for byzantine robust optimization." International conference on machine learning. PMLR, 2021.
>
> [2] Luan, Zhirong, et al. "Robust federated learning: Maximum correntropy aggregation against Byzantine attacks." IEEE Transactions on Neural Networks and Learning Systems 36.1 (2024): 62-75.
>
> 4. **Regarding Weakness 4**
>
> With regard to Weakness 4, we would like to emphasize that the primary advantage and contribution of our work lie in the final test accuracy achieved after convergence under various types and intensities of Byzantine attacks, rather than in the convergence speed.
>
> In terms of the convergence performance of our Fed-NGA, we would like to clarify that, theoretically, as stated in both the Abstract and the summary of contributions (Line 71), we only specify that Fed-NGA achieves a convergence rate of $\mathcal{O}(1/T^{1/2-\delta})$, but we do not present this as an advantage over the baselines in terms of convergence speed. Experimentally, we also observe that sometimes Fed-NGA tends to converge slowly than some baselines.
>
> Regarding the conceptual distinction between convergence speed and final test accuracy, we would like to emphasize that convergence rate is not necessarily indicative of the final accuracy. This phenomenon is illustrated in Figure 6 (Page 14) and Table 4 (Page 15). For instance, when training the LeNet model, Fed-NGA demonstrates slightly faster convergence than most baselines and ultimately achieves superior final accuracy. In contrast, when training the MLP model, Fed-NGA converges marginally slower but still outperforms the baselines in terms of final accuracy.
>
> On the other hand, as explained in the paper starting from Line 490, we intentionally adopt a favorable learning rate setup for the baselines to improve their convergence speed compared to using our learning rate directly, in the interest of fairness.
> In summary, we did not tune the learning rate in a biased manner to gain an unfair advantage over the baselines.
>
> 5. **Regarding Question 1**
>
> With regard to Question 1, we explain the definition of $\delta$ in Line 194 of our paper as “$\delta$ is an arbitrary value lying between $(0, 1/2)$.” This actually implies the $\delta$ serves solely as a tunable parameter for setting the learning rate and is not related to the proportion of Byzantine clients or any other physical parameters. Conceptually, $\delta$ can be viewed as a parameter that influences the convergence rate. Its presence indicates that convergence is guaranteed as long as the learning rate for given $t$ falls within a certain range, rather than requiring a specific fixed value.

---

> > ### Comment · Reviewer_Zgfg · 2025-08-04
> >
> > Thank you to the authors for their response. I appreciate the clarifications provided. However, I still have some follow-up concerns.
> >
> > 1. My major concern is the motivation behind key algorithmic choices.
> >
> > The authors’ response between lines 50 and 58 mainly discusses the limitations of existing studies—such as guaranteed convergence only to a non-zero optimality gap and high computational overhead. However, it remains unclear why normalization was chosen as a core component of the proposed algorithm. The use of normalization itself is not a novel idea in algorithm design, so the rationale behind selecting this particular technique needs to be explicitly addressed.
> >
> > In my view, this reflects a gap between identifying limitations in prior work and the justification for adopting normalization as a solution. Clarifying this connection is essential for understanding the motivation and contribution of the proposed approach.
> >
> > 2. Clarification on the derivation of $\delta$.
> >
> > I understand that "$\delta$ is an arbitrary value in the interval $(0, 1/2)$." However, it is still unclear where this choice arises from in the proof. Could the authors point to the specific equation or step in the analysis that imposes this bound?

---

> ### Author Response · Authors · 2025-08-05
>
> Thanks for the discussion.
>
> 1. **Regarding the motivation**
>
> Firstly, we would like to reiterate that our motivation is solely grounded in addressing the limitations of prior works, particularly their high computational complexity. As stated in Line 57: “In this paper, we aim to fill this research gap, i.e., to reduce the cost of computational complexity for aggregation while preserving generality to data heterogeneity.”
>
> Secondly, regarding the rationale for why normalization could serve as a viable or even effective operator against Byzantine attacks, we acknowledge that normalized gradient has been applied in machine learning without attack, such as in [1]. However, to the best of our knowledge, no prior work has utilized normalized aggregation specifically as a defense mechanism against Byzantine attacks in FL. Therefore, how normalization behaves under Byzantine threats remains unclear, and existing literature does not provide sufficient motivation to adopt it for Byzantine robustness.
>
> Thirdly, regarding our initial motivation to choose normalization as the core aggregator, it comes from the intuition that normalization can reduce the influence of vector magnitude in FL training, particularly for outlier vectors, which is important for achieving robustness against Byzantine attacks. In addition, normalization preserves the directional information of vectors, which supports convergence in the absence of attacks, as discussed in [1].
> However, we acknowledge that this intuition alone is not sufficient to serve as a solid motivation. It is only through our rigorous theoretical analysis and extensive empirical validation, both conducted for the first time in this work, that the effectiveness of normalization under Byzantine settings is confirmed. Therefore, at the stage of motivation, it would not be appropriate to present this intuition without theoretical or experimental support.
>
> In summary, the only reason we adopted normalization at the motivation stage was its low computational complexity, without relying on any presumed benefits beyond that.
>
> [1] Cutkosky, Ashok, and Harsh Mehta. "Momentum improves normalized sgd." International conference on machine learning. PMLR, 2020.

---

> ### Author Response · Authors · 2025-08-05
>
> 2. **Regarding the selection of region $(0, 1/2)$ for  $\delta$**
>
> The choice of $\delta$ is determined by the convergence condition of Fed-NGA, which is specified in Line 190 as “$\displaystyle \lim_{T \rightarrow \infty} \sum_{t=1}^{T-1} \eta^t \rightarrow \infty$ and $\displaystyle \lim_{T \rightarrow \infty} \sum_{t=1}^{T-1} (\eta^t)^2 < \infty$”. This condition can be inferred from Equation (20).
> To satisfy this convergence condition, we define $\displaystyle \eta^t = {\eta}/{(t+c)^{1/2+\delta}}$ with $c > 0$ and $\delta \in (0, 1/2)$, with the convergence rate given by $\mathcal{O} (\frac{1}{T^{1/2-\delta}})$.
>
> **On the other hand, for other intervals or selected values of  $\delta$, convergence may still be achievable, but they do not lead to improved convergence performance.**
>
> In the first step, it is worth noting that the left-hand side of Equation (20) can still converge under a relaxed condition, specifically when $\displaystyle \lim_{T \rightarrow \infty} \sum_{t=1}^{T-1} \eta^t \rightarrow \infty$ and $\displaystyle \lim_{T \rightarrow \infty} \frac{\sum_{t=1}^{T-1} (\eta^t)^2}{\sum_{t=1}^{T-1} \eta^t} < \infty$.
>
> Looking into the case for $\delta \in (-1/2, 0)$, which also promises convergence through the newly highlighted condition, the convergence rate becomes $ \frac{\mathcal{O} (T^{-2\delta})}{\mathcal{O}(T^{1/2-\delta})} = \mathcal{O} (\frac{1}{T^{1/2+\delta}})$. It is straightforward to verify that the convergence rate $\mathcal{O} (\frac{1}{T^{1/2+\delta}})$ for $\delta \in (-1/2, 0)$ is the same as $\mathcal{O} (\frac{1}{T^{1/2-\delta}})$ for $\delta \in (0, 1/2)$. **Thus, the convergence condition $\displaystyle \lim_{T \rightarrow \infty} \sum_{t=1}^{T-1} \eta^t \rightarrow \infty$ and $\displaystyle \lim_{T \rightarrow \infty} \sum_{t=1}^{T-1} (\eta^t)^2 < \infty$ with $\delta \in (0, 1/2)$ suffices to cover this case.**
>
> We further analyze the three boundary cases: $\delta = -1/2$, $\delta = 0$, and $\delta = 1/2$. For $\delta = -1/2$, we have $\eta^t = \eta$, which implies $\displaystyle \lim_{T \rightarrow \infty} \frac{\sum_{t=1}^{T-1} (\eta^t)^2}{\sum_{t=1}^{T-1} \eta^t} = \eta$. In this case, the optimality gap expands to $\mathcal{O} (\eta + \sigma + \theta)$, which is larger than $\mathcal{O} (\sigma + \theta)$ observed in other cases. For $\delta = 0$, $\displaystyle \lim_{T \rightarrow \infty} \sum_{t=1}^{T-1} (\eta^t)^2 \rightarrow \infty$, and the convergence rate degrades to $\mathcal{O} (\frac{\log(T)}{T^{1/2}})$. For $\delta = 1/2$, while the convergence condition is satisfied, the convergence rate degrades to $\mathcal{O} (\frac{1}{\log(T)})$. Both of these rates are slower than our presented $\mathcal{O} (\frac{1}{T^{1/2-\delta}})$.
>
> And for $\delta<-1/2$, $\displaystyle \lim_{T \rightarrow \infty} \frac{\sum_{t=1}^{T-1} (\eta^t)^2}{\sum_{t=1}^{T-1} \eta^t} \rightarrow \infty$, while $\displaystyle \lim_{T \rightarrow \infty} \sum_{t=1}^{T-1} \eta^t < \infty$ for $\delta>1/2$.
> None of the above two cases can promise the convergence of left-hand side of Equation (20).
>
> In light of the above analysis and comparison, we set $\delta \in (0, 1/2)$ to ensure the convergence condition is met and to express the result in the most concise manner.

---

> ### Comment · Reviewer_Zgfg · 2025-08-05
>
> Thank you for the explanation regarding $\delta$—I appreciate the effort.
>
> Regarding the motivation for using normalization, I recommend that the authors provide more discussion and justification in the paper. In particular, it remains unclear **why normalization is chosen as an improved technique under Byzantine threats**, especially in the absence of sufficient supporting discussion.
>
> Currently, the paper lacks a clear explanation of where normalization techniques have been applied in prior literature, what kinds of improvements they achieved in those contexts, and how those insights motivate their potential benefits in the Byzantine setting. Without such discussion, the use of normalization may seem lacking in principled motivation, even if it brings theoretical or empirical gains.
>
> To enhance clarity and justification, I suggest incorporating related works that showcase the effectiveness of normalization in other scenarios. Representative works include, but are not limited to:
>
> [1] Dai, Yan, Kwangjun Ahn, and Suvrit Sra. "The crucial role of normalization in sharpness-aware minimization." Advances in Neural Information Processing Systems 36 (2023): 67741-67770.
>
> [2] Jakovetić, Dus̆an, et al. "Nonlinear gradient mappings and stochastic optimization: A general framework with applications to heavy-tail noise." SIAM Journal on Optimization 33.2 (2023): 394-423.
>
> [3] Li, Xiaoxiao, et al. "Fedbn: Federated learning on non-iid features via local batch normalization." arXiv preprint arXiv:2102.07623 (2021).
>
> [4] Wang, Jianyu, et al. "Tackling the objective inconsistency problem in heterogeneous federated optimization." Advances in neural information processing systems 33 (2020): 7611-7623.
>
> Drawing such connections and clearly explaining how they inform the current application would help readers understand the rationale and novelty of using normalization under Byzantine threats.
>
> Given the importance of this clarification, I believe this issue warrants further revision and possibly another round of review. Thank you again for the thoughtful response. I appreciate the authors' efforts and hope these suggestions are helpful in further improving the clarity and impact of the paper.

---

> > ### Author Response · Authors · 2025-08-06
> >
> > Thank you for your comment. If we have the chance to publish this paper, we would like to incorporate the following supporting discussion into the motivation section, as highlighted in **Bold** below:
> >
> > "It is worth noting that most prior work has primarily concentrated on designing sophisticated aggregation schemes to counter Byzantine attacks, often overlooking the need to reduce the computational burden of aggregation. As summarized in Table 1, the most robust aggregation algorithms typically impose significant computational overhead on the central server, with complexities dependent on parameters mainly including the model dimension $p$, the number of clients $M$, and the error tolerance of aggregation algorithm $\epsilon$. A more detailed survey of related work is provided in Appendix D.
> > In this paper, we aim to fill this research gap, i.e., to reduce the cost of computation complexity for aggregation while preserving generality to data heterogeneity. **One potential approach for achieving our goal is to normalize the uploaded vectors before aggregation. Using normalization to handle gradients is widely adopted in machine learning [1] as it helps stabilize training by ensuring that the loss remains low even under slight perturbations to model parameters [2], and by mitigating the impact of heavy-tailed noise during training [3]. In FL, normalization has also been used to address data heterogeneity, as demonstrated in [4] and [5]. While existing literature has not applied normalization specifically to counter Byzantine attacks, its ability to tackle heavy-tailed noise suggests potential for mitigating the influence of outliers, which are commonly observed under Byzantine  attacks. Even in the absence of adversarial behavior, the operation of normalization can still alleviate data heterogeneity and perform well in a FL system. Last but not least, as shown in Table 1, normalization incurs minimal computational overhead, making it highly attractive option provided it can effectively mitigate the impact of Byzantine attacks. Motivated by above factors,** we investigate the realization of Federated learning employing the Normalized Gradients Algorithm on non-IID datasets and propose a novel robust aggregation algorithm named Fed-NGA."
> >
> > Here, we use [1], [2], [3], [4], and [5] to present our modifications in a more intuitive way. We will revise the corresponding citation formats in the final version.
> >
> > [1] Cutkosky, Ashok, and Harsh Mehta. "Momentum improves normalized sgd." International conference on machine learning. PMLR, 2020.
> >
> > [2] Dai, Yan, Kwangjun Ahn, and Suvrit Sra. "The crucial role of normalization in sharpness-aware minimization." Advances in Neural Information Processing Systems 36 (2023): 67741-67770.
> >
> > [3] Jakovetić, Dus̆an, et al. "Nonlinear gradient mappings and stochastic optimization: A general framework with applications to heavy-tail noise." SIAM Journal on Optimization 33.2 (2023): 394-423.
> >
> > [4] Li, Xiaoxiao, et al. "Fedbn: Federated learning on non-iid features via local batch normalization." arXiv preprint arXiv:2102.07623 (2021).
> >
> > [5] Wang, Jianyu, et al. "Tackling the objective inconsistency problem in heterogeneous federated optimization." Advances in neural information processing systems 33 (2020): 7611-7623.

---

> > > ### Comment · Reviewer_Zgfg · 2025-08-08
> > >
> > > Thank you again for your thoughtful response and for the effort you have put into addressing the concerns raised during the discussion process.
> > >
> > > As mentioned earlier, normalization is not a novel technique. Therefore, applying it in the context of Byzantine attacks calls for a deeper and more principled discussion to justify its use. For instance, in [4], the authors dedicate a full paragraph (page 2) to explaining how batch normalization helps align local training error surfaces, which in turn makes model averaging more effective. Their related work in section 2 also thoroughly discusses the broader benefits of normalization across machine learning settings. In contrast, this submission does not offer a comparable level of analysis. Even during the discussion phase, the authors responded with brief comments without conducting additional literature review or providing deeper insight. Moreover, in the response to Area Chair HRPS’s second question regarding the intuition behind the improvements over FedAvg, the authors also referred to related work on normalization. This further highlights that such motivation and context were missing in the original submission and had to be introduced only during the discussion process.
> > >
> > > To be clear, I am not dismissing the overall contributions of this paper. I sincerely appreciate the progress it makes in both theoretical and empirical aspects. However, I consider motivation to be the backbone of impactful research. When one identifies a gap in a specific setting and proposes to adapt an existing technique, it is important to engage in a thoughtful process of studying, understanding, and explaining that technique. This process builds a compelling bridge between the method and the scenario, and it is this bridge that I find currently lacking. Such a discussion could extend to questions like: Does the use of normalization face any unique challenges under Byzantine attacks in FL, beyond the standard FL discussed in [4]? Without such discussions around motivation, the use of normalization risks appearing as a simple integration rather than an informed design choice.
> > >
> > > I offer these comments with sincere respect for the authors’ efforts, and without any bias. I hope they are received constructively. At present, however, the lack of in-depth discussion regarding the motivation for applying normalization in this specific context leaves me unconvinced that the paper is ready for acceptance.

---

> > > > ### Author Response · Authors · 2025-08-09
> > > >
> > > > Here we present our perspectives on the motivation section.
> > > > We argue that our writing style for motivation does not hinder readers' understanding of the paper's most significant contribution: The advantage of Fed-NGA over other baselines in terms of aggregation time complexity.
> > > >
> > > > Regarding the effectiveness of normalization against Byzantine attacks,
> > > > we emphasize that the motivation section does not aim to convince readers that normalization is **certainly** effective.
> > > > This is because such use of normalization has not appeared in the Byzantine-resilience literature, and claims based solely on concept transfer from other domain or toy examples as suggested in [4] would lack rigor.
> > > > Only theoretical proofs and experimental results, which is provided here for the first time, can substantiate this claim.
> > > >
> > > > Conversely, we aim to establish that normalization offers computational complexity benefits and **may potentially** counter Byzantine attacks.
> > > > So it would be interesting to explore the how does normalization really perform against Byzantine attack, which can be only theoretically and experimentally verified as we did in the following sections.
> > > > This reasoning is coherent and well-founded.
> > > >
> > > > Since we can only conceptually argue that normalization may help counter Byzantine attacks, drawing analogies from other domains or using toy examples as suggested in [4] makes little difference, as neither approach can conclusively demonstrate its effectiveness in this setting.
> > > > Moreover, it is common in the literature to use analogies from other domains rather than toy examples in motivation sections.
> > > > For example: CClip [1] and the paper https://openreview.net/pdf?id=G8aS48B9bm mentioned by Area Chair HRPS. Clip itself is not a novel technique, but CClip was the first to use clipping for defending against Byzantine attacks, while the paper in https://openreview.net/pdf?id=G8aS48B9bm was the first to utilize clipping to assist in achieving both Byzantine robustness and partial participation simultaneously. Neither paper relies on toy examples to clarify their motivations.
> > > >
> > > > Regarding the unique challenge normalization faces under Byzantine attacks beyond standard FL, no definitive obstacle has been identified due to the current uncertainty about normalization’s effectiveness. If such a challenge exists, it arises because Byzantine clients can upload arbitrary vectors that induce arbitrary gradient directions during aggregation, even though normalization may flatten their amplitudes. This point naturally follows from the discussion of Byzantine client behavior highlighted in the Introduction.
> > > >
> > > > Finally, we sincerely appreciate the recommendation of related works on normalization in other domains, and we will retain them in the motivation section to illustrate the idea that normalization may help defend against Byzantine attacks. We also thank the reviewer for recognizing our theoretical and empirical contributions, which we believe form the core of our paper.
> > > >
> > > > [1] Karimireddy, Sai Praneeth, Lie He, and Martin Jaggi. "Learning from history for byzantine robust optimization." International conference on machine learning. PMLR, 2021.

---

### Official Review · Reviewer_TKaS · 2025-07-02

**Clarity:** 3
**Significance:** 3
**Originality:** 3
**Rating:** 5
**Confidence:** 3

**Summary:**

Fed-NGA introduces a approach for secure federated learning, effectively addressing dual challenges of Byzantine attacks and data heterogeneity through an elegantly simple yet powerful mechanism. By aggregating normalized client gradients via weighted averaging, the algorithm achieves unprecedented computational efficiency—reducing time complexity to optimal linear scaling while maintaining strong robustness guarantees. Crucially, Fed-NGA demonstrates rigorous convergence to stationary points even under non-convex loss landscapes, with theoretical analysis confirming resilience when malicious clients comprise up to half the network. Experimental validation across diverse datasets shows significant speed advantages over existing methods while maintaining competitive accuracy, though performance gradually decreases under extreme data heterogeneity.

**Questions:**

See Weakness.

**Ethical Concerns:**

["NO or VERY MINOR ethics concerns only"]

**Final Justification:**

After the rebuttal and discussion process, I am happy to change my decision to accept this paper.

**Limitations:**

Yes, the proposed method, while promising, reveals certain limitations in the Experiments section.

**Paper Formatting Concerns:**

This paper has no major formatting issues.

**Quality:**

3

**Strengths And Weaknesses:**

Strengths:
1.Fed-NGA excels in algorithmic simplicity and conceptual clarity, making it highly accessible for both theoretical analysis and practical implementation. The core innovation lies in its streamlined aggregation process: instead of relying on complex statistical filters or iterative optimizations, Fed-NGA merely normalizes each client's gradient vector to unit length and computes their weighted mean.
2.Beyond efficiency, Fed-NGA demonstrates remarkable resilience against security threats. It maintains robust performance even when malicious clients comprise 40% of the network, effectively neutralizing diverse attack strategies including Gaussian noise injection, gradient flipping, and sophisticated evasion techniques. The normalization process inherently nullifies magnitude-based attacks while preserving the directional integrity of honest updates.
3.The algorithm also addresses the persistent challenge of data heterogeneity. Unlike alternatives that require specialized handling of non-IID data distributions, Fed-NGA's normalized gradient aggregation naturally balances biased local updates. This provides reliable performance across varying degrees of data skew without additional computational cost.

Weaknesses:
1.The algorithm exhibits concerning performance volatility when confronting Byzantine client ratios exceeding 30%, revealing a critical vulnerability threshold in its security architecture. Beyond this inflection point, coordinated attacks (particularly Same-Value and Sign-Flip strategies) trigger precipitous accuracy declines that manifest as erratic fluctuations rather than graceful degradation.
2.Fed-NGA displays heightened sensitivity to data heterogeneity that compounds operational challenges in practical environments. As client data distributions become increasingly skewed—quantified by decreasing β values in the Dirichlet allocation framework, the algorithm suffers progressive performance erosion that persists despite its superiority over conventional baselines.

---

> ### Author Rebuttal · Authors · 2025-07-29
>
> 1. **Regarding Weakness 1**
>
> With regard to Weakness 1, we acknowledge that the performance degradation of Fed-NGA is not entirely uniform. However, as shown in Figure 7 (Page 15) and Figure 9 (Page 16), nearly all baselines exhibit an inflection point where the degradation pattern shifts from a gradual to a markedly faster decline in test accuracy. For instance, Figure 7(b) and Figure 9(b) demonstrate that MCA and CClip experience a modest decline when $\bar{C_{\alpha}} \leq 0.1$, followed by a steep drop under the Same-value attack, reaching their lowest performance at $\bar{C_{\alpha}} = 0.2$. Similarly, Figure 7(e) and Figure 9(e) show that Median, Krum, and GM also undergo slow degradation up to their respective inflection points, after which the performance decreases rapidly under the FoE attack ($\bar{C_{\alpha}} = 0.3$ for Median and Krum, and $\bar{C_{\alpha}} = 0.2$ for GM).
>
> These observations suggest that such abrupt performance degradation beyond certain Byzantine ratios is a common vulnerability in existing approaches. In contrast, Fed-NGA maintains a more stable performance trajectory, with test accuracy degrading slowly over a wider range of Byzantine ratios. The onset of sharp degradation is postponed to significantly higher Byzantine ratios, particularly under the Same-value and Sign-flip attacks. Moreover, Figures 7 and 9 show that Fed-NGA outperforms the baseline methods across the evaluated attack scenarios in most cases.
>
> 2. **Regarding Weakness 2**
>
> We acknowledge that the performance of Fed-NGA degrades as $\beta$ decreases. However, as shown in Table 4 (Page 15), Table 5 (Page 17), and Table 6 (Page 18), the performance of nearly all baselines decreases as $\beta$ decreases. And the performance fluctuations under different $\beta$ settings are summarized in the table below (We omit FedAvg from the comparison here, as the algorithm barely converges under attacks.).
>
> Table: Fluctuation of accuracy under different $\beta$ settings with $\bar{C_{\alpha}}=0.2$ over five attack types. - indicates that the algorithm does not converge under this setting.
> |         |          |            |  Table 4  |          |          |          |            |  Table 5  |         |          |          |            |  Table 6  |           |          |
> |:-------:|:--------:|:----------:|:---------:|:--------:|:--------:|:--------:|:----------:|:---------:|:-------:|:--------:|:--------:|:----------:|:---------:|:---------:|:--------:|
> |         | Gaussian | Same-value | Sign-flip |    LIE   |    FOE   | Gaussian | Same-value | Sign-flip |   LIE   |    FOE   | Gaussian | Same-value | Sign-flip |    LIE    |    FOE   |
> | Fed-NGA |   0.62   |    1.91    |  **0.45** |   0.48   | **0.45** |   1.64   |    2.04    |    0.47   |   1.33  | **0.47** |   0.92   |    2.19    |    2.93   |    1.03   | **1.69** |
> |  Median |   1.76   |    8.92    |    2.19   |   1.88   |   2.83   |   0.36   |    3.71    |    1.68   |   0.66  |    0.7   |   8.18   |    1.14    |   10.63   |    6.23   |   8.41   |
> |   Krum  |    1.8   |    9.17    |    2.28   |   1.56   |   7.12   |   0.73   |    1.12    |    2.42   |   2.74  |   0.51   |   4.64   |  **0.29**  |    3.62   |    5.32   |     -    |
> |    GM   |   0.78   |  **0.96**  |    1.05   |   0.39   |     -    |   0.24   |  **0.81**  |  **0.16** |   0.62  |     -    | **0.21** |    1.37    |  **2.47** |   -0.27   |   3.27   |
> |   MCA   | **0.07** |      -     |     -     |   3.09   |     -    | **0.15** |      -     |     -     | **0.3** |     -    |   0.23   |      -     |     -     | **-0.41** |     -    |
> |  CClip  |   0.23   |      -     |     -     | **0.23** |     -    |   3.22   |      -     |     -     |   1.72  |     -    |   2.12   |    0.34    |    7.24   |   -0.21   |   7.28   |
>
> From this table, the results indicate that Fed-NGA's (achieved 4 best performance) adaptability to data heterogeneity is comparable to that of GM (achieved 5 best performance) and MCA (achieved 4 best performance), and better than other robust algorithms. Additionally, even in suboptimal scenarios, Fed-NGA performs comparably to its optimal-case performance.
>
> The above analysis confirms that Fed-NGA exhibits top-tier adaptability to data heterogeneity compared to all baselines, without displaying heightened sensitivity to it.

---

> > ### Comment · Reviewer_TKaS · 2025-08-01
> >
> > Thanks for the discussion and it addressed some of my previous concerns.

---

> ### Author Response · Authors · 2025-08-06
>
> We would like to express our sincere gratitude for your diligent review of our paper. Your constructive comments have been instrumental instrumental in enhancing the quality of our paper, and we deeply value the insights you’ve shared. We are also thankful for your positive assessment of our theory, which serves as great encouragement for us to continue refining our work. Thank you once more for your time and invaluable support.

---

### Official Review · Reviewer_8n5C · 2025-07-03

**Clarity:** 3
**Significance:** 3
**Originality:** 3
**Rating:** 4
**Confidence:** 3

**Summary:**

The paper presents Fed-NGA, a federated learning (FL) algorithm designed to enhance robustness against Byzantine attacks and data heterogeneity. Fed-NGA uses normalized gradients and computes their weighted mean for aggregation, significantly reducing computational overhead compared to existing robust aggregation methods. Experimental evaluations on standard datasets (MNIST, CIFAR-10, TinyImageNet) demonstrate the efficiency and robustness of Fed-NGA compared to baselines such as FedAvg, Median, Krum, GM, CClip, and MCA.

**Questions:**

- How does Fed-NGA handle partial client participation or dropped clients?
- How does the method ensure fairness among clients with varying data quality, quantity, or training effort?
- Have you evaluated the communication cost of your method, such as the number of transmitted bytes, communication rounds, or total bandwidth usage?

**Ethical Concerns:**

["NO or VERY MINOR ethics concerns only"]

**Final Justification:**

After the rebuttal and discussion phase, the authors address my concerns regarding communication overhead and partial participation. Therefore, I decided to keep the positive score for this paper.

**Limitations:**

Yes. The paper acknowledges some limitations, but critical limitations regarding robustness to adaptive attackers, unrealistic theoretical assumptions, and neglected gradient magnitude information remain inadequately addressed.

**Quality:**

3

**Strengths And Weaknesses:**

**Strengths:**

* Overall, the structure of the paper is well-organized and easy to follow.
* The proposed method has good potential for real-world deployment with many clients, as it significantly reduces computational overhead with O(pM) aggregation complexity.
* The paper provides theoretical analysis supporting robustness against Byzantine attacks and convergence guarantees.
* The experiments provide comparative analysis across multiple attacks and scenarios.

**Weaknesses:**
- My main concern is the lack of analysis of communication cost in the proposed method, which is a critical aspect of FL systems. Communication overhead can dominate computation in real-world FL deployments and making it a more critical bottleneck than the efficiency of the aggregation method itself. E.g., it is unclear if normalization incurs additional costs during message preparation or transmission.
- There is a fairness issue related to the client’s actual contribution to the model’s improvement, i.e., normalizing gradients and aggregating them without accounting for factors such as data quality, quantity, or training effort. In real-world FL settings, clients with more useful data may not want to participate since their extra contribution is not recognized or rewarded.
- This study may overestimate model performance and convergence speed since it assumes full client participation. It is important to consider partial participation, which is a more common and realistic form of the FL setting.

---

> ### Author Rebuttal · Authors · 2025-07-29
>
> 1. **Regarding the communication cost issue**
>
> Regarding the communication cost issue, as detailed in Algorithm 1 on Page 5, Fed-NGA normalizes the uploaded vectors at the central server, requiring each participating client to upload its gradient (real or fake depending on whether it is malicious). This design yields exactly the same communication cost as all the robust  baselines, and thus does not introduce any additional communication overhead.
>
> 2. **Regarding clients contributing higher data quality, quantity, or training effort could deserve greater weight**
>
> We acknowledge that clients contributing higher data quality, quantity, or training effort could deserve greater weight. Nonetheless, under the threat of uncertain Byzantine behavior, relying on such trust becomes precarious. To be specific, Byzantine clients may gradually build credibility over multiple rounds before launching attacks, making over-reliance on them risky. On the other hand, this type of time-varying behavior highlights the need for robust methods that can adapt to dynamic adversaries. Although our analysis assumes fixed Byzantine clients, the proposed method actually remains effective even when the set of Byzantine clients changes over time, as long as the per-round Byzantine ratio $\bar{C}_{\alpha} < 0.5$ is maintained.
>
> 3. **Regarding partial client participation and dropped clients**
>
> With regard to the case of partial participation, to the best of our knowledge, none of the existing robust aggregation methods, including ours, explicitly consider this setting. This is primarily due to a fundamental limitation. Byzantine clients, whose objective is to disrupt the training process, may consistently participate in each training round. If some honest clients are inactive due to partial participation, the instantaneous Byzantine ratio $\bar{C}_{\alpha}$ in a given round can exceed 0.5. This challenges the core assumption required by all Byzantine-resilient algorithms, namely that honest clients form the majority in each round, and poses a fundamental limitation on defending against Byzantine attacks without leveraging external trusted information.
> Even in scenarios where the central server randomly drops clients, regardless of whether they are Byzantine or honest, the resulting Byzantine ratio after dropping cannot be guaranteed to stay below 0.5, either.
>
> On the other hand, partial participation or client dropout can be accommodated in our framework by treating inactive or dropped clients as Byzantine, since Byzantine clients are allowed to upload arbitrary gradients to the central server, including the null vector. Under this modeling, our theoretical results still guarantee convergence as long as the effective Byzantine ratio $\bar{C}_{\alpha} < 0.5$ in each training round, where inactive or dropped clients are also counted as Byzantine.
>
> It is also worth noting that the set of tolerable inactive or dropped clients and Byzantine clients can vary across training rounds. Although our theoretical analysis assumes a fixed set of Byzantine clients, the proposed Fed-NGA actually remains effective even when the Byzantine set changes over time, demonstrating robustness under more dynamic and realistic conditions.

---

> > ### Comment · Reviewer_8n5C · 2025-08-05
> >
> > Thank you for the responses and clarifications. I have some concerns as follows:
> >
> > - You claim Fed-NGA has no extra communication overhead, but can you provide empirical measurements (e.g., bandwidth per round) comparing it to baselines like Krum or Median?
> >
> > - Your rebuttal suggests Fed-NGA can handle partial participation if the effective Byzantine ratio stays <0.5. How often is this realistic? Can you simulate scenarios where dropout rates push the ratio beyond 0.5?"*

---

> ### Author Response · Authors · 2025-08-06
>
> Thanks for the discussion.
>
> 1. **Regarding the concern 1**
>
> In practice, nearly all robust aggregation algorithms are implemented on the central server. They operate by taking the gradient vectors uploaded by clients as input and producing an updated gradient as output. This updated gradient is then used to generate the updated model parameters, which are broadcasted to all clients for the next training round.
> In both the uploading stage from clients to the server and the broadcasting stage from the server to clients, the aggregation algorithm does not alter the content or dimensionality of the transmitted vectors. Therefore, the communication cost between the server and each client is solely determined by the number of parameters in the vector and the storage format (e.g., float32 by default in PyTorch).
>
> The experiments below demonstrate, for different aggregation algorithms, the number of parameters transmitted from each client to the central server per round or received by each client, the total parameter storage size, and **the communication bandwidth required to upload these parameters within 1 second**. These results are illustrated using the LeNet model on the MNIST and CIFAR10 datasets.
>
> | Dataset |            |     MNIST    |           |            |    CIFAR10   |           |
> |:-------:|:----------:|:------------:|:---------:|:----------:|:------------:|:---------:|
> |         | Parameters | Storage size | Bandwidth | Parameters | Storage size | Bandwidth |
> | Fed-NGA |   41.282K  |   0.161 MB   | 1.29 Mbps |  797.962K  |    3.19MB    |  25.5Mbps |
> |  FedAvg |   41.282K  |   0.161 MB   | 1.29 Mbps |  797.962K  |    3.19MB    |  25.5Mbps |
> |  Median |   41.282K  |   0.161 MB   | 1.29 Mbps |  797.962K  |    3.19MB    |  25.5Mbps |
> |   Krum  |   41.282K  |   0.161 MB   | 1.29 Mbps |  797.962K  |    3.19MB    |  25.5Mbps |
> |    GM   |   41.282K  |   0.161 MB   | 1.29 Mbps |  797.962K  |    3.19MB    |  25.5Mbps |
> |   MCA   |   41.282K  |   0.161 MB   | 1.29 Mbps |  797.962K  |    3.19MB    |  25.5Mbps |
> |  CClip  |   41.282K  |   0.161 MB   | 1.29 Mbps |  797.962K  |    3.19MB    |  25.5Mbps |
>
> As shown in the table above, all algorithms, including Fed-NGA and baselines, require the same upload bandwidth. This is because aggregation algorithms do not alter the size of the parameters before upload or download. Compared with baseline algorithms, Fed-NGA is not an exception and consequently does not introduce any additional communication overhead.

---

> ### Author Response · Authors · 2025-08-06
>
> 2. **Regarding the concern 2**
>
> Regarding the scenario where dropped clients might increase the effective Byzantine ratio beyond 0.5, we have revisited the problem.
> Instead of treating dropped clients as Byzantine, our approach can in fact tolerate a higher number of Byzantine and dropped clients simultaneously.
>
> To be specific, by excluding dropped clients from the aggregation and assuming the worst-case scenario where Byzantine clients never drop out, the proportion of Byzantine clients in the aggregation remains lower than that of honest clients as long as the condition **1 - dropout ratio $>$ 2 $\times$ Byzantine ratio** holds.
> Several representative configurations characterizing this bound include:
> (1) Byzantine ratio = 0.1, dropout ratio = 0.8;
> (2) Byzantine ratio = 0.2, dropout ratio = 0.6;
> (3) Byzantine ratio = 0.3, dropout ratio = 0.4;
> (4) Byzantine ratio = 0.4, dropout ratio = 0.2.
> As long as the Byzantine and dropout ratios fall within these bounds, convergence is still guaranteed for our Fed-NGA.
> **This refined analysis provides a tighter characterization of the feasible region, allowing for a higher tolerance of dropped clients, which better reflects practical scenarios.**
>
> To validate this, we conducted experiments under the two settings, evaluating final accuracy (\%) using the LeNet model on the CIFAR10 dataset under the Sign-flip attack with the Byzantine ratio = 0.2 and $\beta = 0.6$.
> In the table, **Bold** indicates that the algorithm achieves SOTA performance, while _Italic_ indicates that the algorithm failed to converge.
>
> | Dropout ratio |  Fed-NGA  |  FedAvg |  Median |   Krum  |   GM   |   MCA   |  CClip  |
> |:-------------:|:---------:|:-------:|:-------:|:-------:|:------:|:-------:|:-------:|
> |      0.5      | **46.63** | _10.00_ |  30.39  |  33.72  |  44.97 | _10.00_ | _10.00_ |
> |      0.7      |  _10.80_  | _10.00_ | _10.19_ | _10.00_ | _9.98_ | _10.00_ | _10.00_ |
>
> From the first row of the table, where the Byzantine ratio is 0.2 and the dropout ratio is 0.5, which does not violate the newly established bound, Fed-NGA achieves SOTA performance. This is because, after removing 50\% of the clients and keeping the 20\% Byzantine clients active, the effective Byzantine ratio among the remaining clients becomes 0.2 / 0.5 = 0.4, which is within the tolerable range derived in our analysis.
> In contrast, the second row corresponds to a Byzantine ratio of 0.2 and a dropout ratio of 0.7, which exceeds the derived bound. In this case, no algorithm is able to maintain convergence under such a high dropout rate in the presence of Byzantine clients. **This highlights the existence of a critical dropout ratio threshold beyond which convergence cannot be guaranteed for any algorithm under Byzantine attacks.**

---

> > ### Comment · Reviewer_8n5C · 2025-08-06
> >
> > Thank the authors for the clarifications, and I will maintain my posiitive socre.

---

> > > ### Author Response · Authors · 2025-08-06
> > >
> > > Thank you very much for your careful and thorough review of our paper. We deeply appreciate your constructive comments, which have positively contributed to improving the quality of our paper. We are also grateful for your positive evaluation of our paper, as it encourages us to further refine and advance our research. Thank you again for your valuable time and support.

---

### Comment · Area_Chair_HRPS · 2025-08-06
**Some Questions**

Dear Authors,

I had several questions:

Could you briefly compare your work to https://openreview.net/pdf?id=G8aS48B9bm

Can you provide an intuition why the method would improve on FedAvg in the No Attack setting in your results?

How does your method impact the learning rate tuning?

Can you provide for a single dataset/model(e.g. cifar) an idea of the relative runtime (it can be based on a simple FLOP analysis or wall time) compared to the total runtime for a round? Relative to the time of a round it is not completely clear if a significant amount of time is saved

---

> ### Author Response · Authors · 2025-08-07
>
> Thanks for the discussion.
>
> 1. **Regarding question 1**
>
> In fact, our work has clear distinctions from the referenced paper in both core objectives and technical approaches. The referenced paper focuses on achieving partial participation with Byzantine robustness simultaneously, employing gradient difference clipping between two adjacent local training steps to facilitate partial participation, while relying on existing Byzantine-resilient aggregation methods for robustness.
> In contrast, our work introduces a novel Byzantine-resilient aggregation method, Fed-NGA, which is specifically designed to defend against Byzantine attacks. This method demonstrates robustness and features low aggregation time complexity, with its core contribution lying in the innovation of the aggregation mechanism itself rather than addressing partial participation.
>
> 2. **Regarding question 2**
>
> We attribute this phenomenon to the role of normalization, which helps mitigate the impact of data heterogeneity [1], [2]. All our experiments are conducted in a data heterogeneous environment, where data heterogeneity can be viewed as a form of perturbation. **Fed-NGA is robust to Byzantine attacks, and as such, it naturally exhibits resilience to the perturbations induced by data heterogeneity.** Consequently, based on the arguments outlined above, this indicates that Fed-NGA may outperform FedAvg in the no attack scenario.
> FedAvg is a classic FL algorithm, yet this does not imply it is well-suited to data heterogeneity.
> For instance, our experimental results show that MCA and GM sometimes achieve slightly higher accuracy than FedAvg, particularly under high data heterogeneity. This can be attributed to the fact that robust algorithms exhibit resilience to slight perturbations, such as those induced by data heterogeneity.
>
> [1] Li, Xiaoxiao, et al. "Fedbn: Federated learning on non-iid features via local batch normalization." arXiv preprint arXiv:2102.07623 (2021).
>
> [2] Wang, Jianyu, et al. "Tackling the objective inconsistency problem in heterogeneous federated optimization." Advances in neural information processing systems 33 (2020): 7611-7623.
>
> 3. **Regarding question 3**
>
> The Fed-NGA algorithm we propose involves normalizing the vectors uploaded by clients, a process that alters the modulus of the aggregated vector $g^t$ and consequently impacts the tuning of the learning rate. This issue is also elaborated on in Line 490.
>
> For the training model employed in our experiments, we observed that the vector modulus derived from local training is generally greater than 1. To address this, relative to the baselines, we increased the initial learning rate of Fed-NGA and reduced its learning rate decay rate to ensure convergence (Line 483). Additionally, we noted that baselines fail to perform adequately when using the learning rate parameters optimized for our algorithm (Line 491).
>
> Furthermore, the normalization stabilizes the modulus of the aggregated gradient vector across rounds. This, in turn, reduces the sensitivity of convergence to the learning rate and does not increase the burden of learning rate tuning.
>
> 4. **Regarding question 4**
>
> Under the conditions you specified, our simulations show that the average wall time required for one round of local training of the LeNet model on the CIFAR10 dataset is **42.3 ms** on clients. Below, we take a Byzantine ratio of 0.3 as an example to present the per-round wall time considering only the aggregation time (ms) on the central server.
>
> |         | No Attack | Gaussian | Same-value | Sign-flip |   LIE   |   FoE  |
> |:-------:|:---------:|:--------:|:----------:|:---------:|:-------:|:------:|
> | Fed-NGA |   0.1116  |  0.0982  |   0.0908   |   0.1114  |  0.0844 | 0.1114 |
> |  Median |   44.326  |  62.126  |   64.392   |   60.832  |  62.122 | 60.714 |
> |   Krum  |   43.546  |  62.158  |   65.032   |   62.074  |  62.104 | 61.846 |
> |    GM   |  15.7824  |  30.344  |   52.504   |   37.988  |  28.28  | 73.868 |
> |   MCA   |  12.7318  |  14.6904 |   50.196   |   2738.8  | 14.6402 | 2738.8 |
> |  CClip  |  11.2018  |  9.3134  |   32.032   |   2013.4  |  9.2546 | 2013.4 |
>
> Specifically, the aggregation time of Fed-NGA accounts for only 0.2\%-0.26\% of the total per-round time (local training + aggregation). In contrast, Krum’s aggregation time exceeds 50\% of the total round time across all attack scenarios, and GM’s aggregation time ranges from 27\% (No Attack) to 63\% (FoE) of the total time. This stark contrast demonstrates that Fed-NGA offers significant advantages in terms of runtime efficiency. The results clearly validate that the reduction in per-round FL time time achieved by Fed-NGA is practically meaningful.

---

### Decision · Program_Chairs · 2025-09-17

**Decision:**

Accept (poster)

**Comment:**

This paper introduces a simple byzantine resilient aggregation technique that simply normalizes the client gradients before averaging. The reviewers generally agree that the work is technically sound, with convergence analysis, proofs of Byzantine robustness, and  experiments across diverse datasets and attack types. A key strength is the scalability compared to classic methods like Krum.  The empirical results convincingly show both runtime and accuracy advantages compared to established baselines (Median, Krum, GM, MCA, CClip). One reviewer raised concerns about the motivation since normalization is not an inherently novel idea. Another concern (from the meta-reviewer) is the need to tune new HP for this method compared to the base FL method used. Overall, the paper makes a solid contribution to the area of Byzantine-robust FL, with both an empirical evaluation of a simple practical method and providing the associated theoretical basis.